# UniRestorer: Universal Image Restoration via Adaptively Estimating Image Degradation at Proper Granularity

**Jingbo Lin[1], Zhilu Zhang[1], Wenbo Li[2], Hang Xu[2], Hongzhi Zhang[1], Wangmeng Zuo[1]** [*]
[1] Harbin Institute of Technology, [2] Huawei Noah's Ark Lab
`{jblincs1996@gmail.com}`

## Abstract

Recently, considerable progress has been made in all-in-one image restoration. Generally, existing methods can be degradation-agnostic or degradation-aware. However, the former are limited in leveraging degradation estimation-based priors, and the latter suffer from the inevitable error in degradation estimation. Consequently, the performance of existing methods has a large gap compared to specific single-task models. In this work, we make a step forward in this topic, and present our UniRestorer with improved restoration performance. Specifically, we perform hierarchical clustering on degradation space and train a multi-granularity mixture-of-experts (MoE) restoration model. Then, UniRestorer adopts both degradation and granularity estimation to adaptively select an appropriate expert for image restoration. In contrast to existing degradation-agnostic and -aware methods, UniRestorer can leverage degradation estimation to benefit degradation-specific restoration and use granularity estimation to make the model robust to degradation estimation error. Experimental results show that our UniRestorer outperforms state-of-the-art all-in-one methods by a large margin, and is promising in closing the performance gap to specific single-task models. The code and pre-trained models are released at https://github.com/mrluin/UniRestorer.

## 1 Introduction

Image restoration is a fundamental task in computer vision, it aims to restore high-quality (HQ) images from corresponding low-quality (LQ) counterparts, including denoising (Zhang et al., 2021b; Abdelhamed et al., 2018), deblurring (Zhang et al., 2022c; Nah et al., 2017a; Rim et al., 2020), de-weathering (Liu et al., 2024; 2018a; Fu et al., 2017), low-light enhancement (Zhang et al., 2024b; Wei et al., 2018; Xu et al., 2022), etc. In the past decade, advanced deep neural architectures (He et al., 2016; Vaswani et al., 2017; Liu et al., 2021; Zhang et al., 2023b; Radford et al., 2021; Li et al., 2024; ?; Kong et al., 2025) have driven image restoration methods to evolve from task-specific (Zhang et al., 2017; Dong et al., 2014; Kupyn et al., 2019; Ren et al., 2019; Song et al., 2023; Liu et al., 2018b; Zhang et al., 2021c; 2022b; 2024a; 2025) to task-agnostic (Zamir et al., 2021; Liang et al., 2021; Chen et al., 2022; Zamir et al., 2022; Lin et al., 2024) backbones. Recently, influenced by the success of foundation models in natural language processing and high-level vision, there has been increasing interest in addressing multiple image restoration tasks within a single framework, known as all-in-one image restoration (Jiang et al., 2024).

A straightforward approach is to utilize data from all tasks to train a degradation-agnostic model (see Fig. 1 (a)), but it easily yields unsatisfactory results due to the conflicts between certain tasks (*e.g.*, denoising and deblurring). Inspired by parameter-efficient fine-tuning (PEFT) methods (Brian et al., 2021; Houlsby et al., 2019), learnable prompts (Li et al., 2022a; Zhang et al., 2023a; Potlapalli et al., 2023; Valanarasu et al., 2022; Guo et al., 2024; Xiangtao et al., 2024) and adapters (Ai et al., 2024b) are integrated into the restoration model to modulate intermediate features (see Fig. 1 (b)), allowing the model to process different tasks more specifically. Furthermore, some methods introduce mixture-of-experts (MoE) (Yang et al., 2023; Chen et al., 2024; Cao et al., 2024b; Wenlong et al.,

---
[*]Corresponding author.

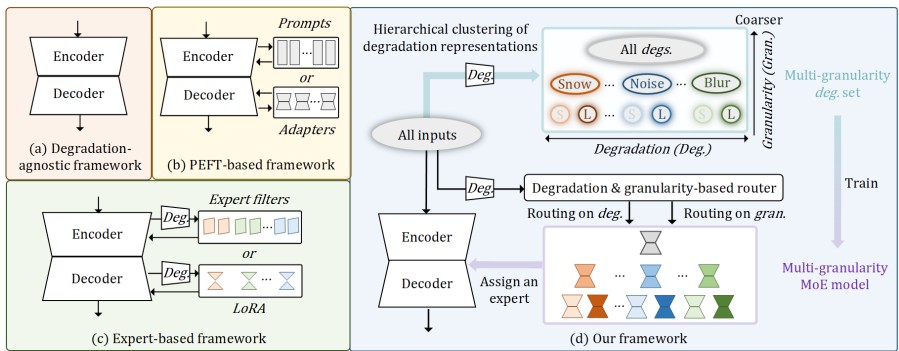

Figure 1: **Illustration of representative all-in-one image restoration frameworks**. **(a)** Degradation-agnostic methods (Wang et al., 2021; Zhang et al., 2021a) train a shared backbone using data from all tasks and are limited in leveraging degradation priors. **(b)** PEFT-based methods apply learnable prompts or adapters to the backbone to adapt various tasks. **(c)** Expert-based methods train specific MoE, LoRA, or filters for specific tasks. However, both (b) and (c) suffer from the inevitable error in degradation estimation. **(d)** We construct a multi-granularity degradation set and a multi-granularity MoE restoration model. By jointly leveraging the degradation- and granularity-based router, our UniRestorer can be effective in leveraging degradation-specific restoration while being robust to degradation estimation error.

2023; Li et al., 2022c) (MoE-like using filters (Park et al., 2023; Cao et al., 2024a; Conde et al., 2024; Liang et al., 2022; Xu et al., 2024; Zhu et al., 2023) and low-rank adaptation (LoRA) (Tian et al., 2024; Ai et al., 2024a; Zamfir et al., 2025)) modules, routing the current feature to specific experts based on deep features and estimated degradation (see Fig. 1 (c)).

Despite rapid progress, these methods show limited performance and fall significantly behind specific single-task models. Degradation-agnostic models cannot leverage the benefits from degradation estimation-based priors, limiting their effectiveness in all-in-one image restoration. PEFT-based and MoE-based methods provide degradation-aware ways for processing different corruptions by conditioning on degradation estimation. However, their effectiveness heavily relies on accurate degradation estimation, which is inherently challenging due to the vast and diverse degradation space of all-in-one image restoration. Consequently, degradation-aware models are inevitably vulnerable to estimation errors, which can lead to inappropriate prompt or expert routing, ultimately degrading restoration quality.

From the above analysis, one plausible solution to improve all-in-one restoration is to estimate and leverage image degradation while being robust to degradation estimation error. To this end, we propose to represent image degradation at multiple levels of granularities rather than a single-level, and construct a multi-granularity degradation representation (DR) set by hierarchizing the degradation space. We then build a corresponding multi-granularity MoE restoration model, where fine-grained experts specialize in specific degradations to enhance restoration accuracy, while coarse-grained experts generalize across broader degradation spaces. To enable more robust expert routing for a degraded image, beyond vanilla degradation estimation, we introduce granularity estimation to indicate the degree of degradation estimation error. For example, in Fig. 1 (d), given an image suffering from large snow degradation, if the degradation estimation is ambiguous, the granularity estimation will assign a coarser-grained expert, *e.g.*, an expert for general snow, to alleviate the effect of estimation error. Conversely, when the degradation estimation is confident, granularity estimation will assign a finer-grained expert to maximize the restoration quality, *e.g.*, an expert for large snow. By jointly leveraging both degradation and granularity estimation, our approach can mitigate the adverse effects of inaccurate degradation estimation, achieving high restoration and generalization performance simultaneously.

Specifically, we take the following steps to implement our UniRestorer. First, we optimize a degradation extractor that is aware of finer-grained image degradation. Second, all low-quality images in the training set are fed into the degradation extractor, and their features are then hierarchically clustered into multiple DR groups at different granularities, respectively, which forms the multi-granularity degradation set. Third, we train a multi-granularity MoE restoration model based on the constructed multi-granularity degradation set. Fourth, given images with unknown degradations, we learn routers based on both degradation and granularity estimation results. The router identifies which

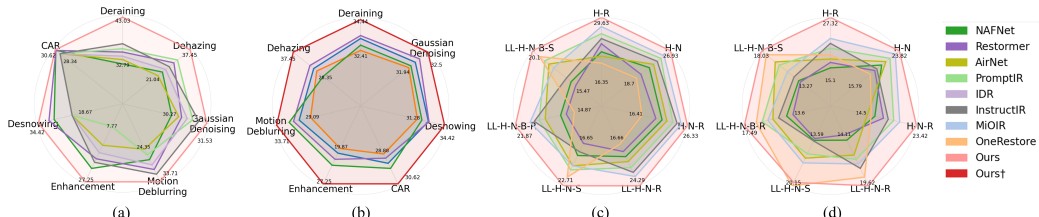

Figure 2: **Comparisons with task-agnostic methods, all-in-one methods, and single-task models.** Ours and Ours†denote *auto* and *instruction* modes, which are respectively used for a fair comparison with all-in-one and specific single-task models. **(a)** Comparisons on single-degradation all-in-one setting. **(b)** Comparisons with specific single-task models. **(c)** Comparisons on mixed-degradation all-in-one (in-of-distribution) setting. **(d)** Comparisons on mixed-degradation all-in-one (out-of-distribution) setting.

DR group the degradation and granularity best match in the degradation set, and then allocates the corresponding expert for inference.

To demonstrate the effectiveness of the proposed method, we conduct experiments on both single-degradation and mixed-degradation scenarios. For the single-degradation scenario, we unify seven widely studied image restoration tasks. For the mixed-degradation scenario, we evaluate on the CDD-11 (Guo et al., 2024) benchmark dataset. In addition, we conduct a comprehensive comparative evaluation. The results indicate that our method not only significantly outperforms existing all-in-one methods but also achieves performance that is competitive with, or even superior to, specific single-task models (see Fig. 2).

The main contributions can be summarized as follows:

- We propose UniRestorer, a novel all-in-one image restoration framework that can estimate and leverage degradation representation to adaptively handle diverse image corruptions.

- We propose a multi-granularity MoE restoration model, where finer-experts specialize in specific degradations to enhance restoration accuracy, while coarser-experts generalize on broader degradation spaces to ensure generalization ability.

- To alleviate the impact of degradation estimation error on expert routing, we propose to estimate a proper granularity for image degradation. By jointly leveraging degradation and granularity estimation, our method can adaptively allocate the most appropriate expert.

- Extensive experiments are conducted on single-degradation and mixed-degradation scenarios, showing that our method significantly outperforms existing all-in-one methods and effectively closes the performance gap with specific single-task models.

## 2 RELATED WORK

**Image Restoration for Specific Tasks** Pioneer works propose tailored frameworks for specific degradations, *e.g.*, DnCNN (Zhang et al., 2017) for denoising, SRCNN (Dong et al., 2014) for superresolution, DeblurGANv2 (Kupyn et al., 2019) for deblurring, PReNet (Ren et al., 2019) for draining, Dehazeformer (Song et al., 2023) for dehazing, and DesnowNet (Liu et al., 2018b) for desnowing. Benefits from advanced deep neural networks, some works tend to develop task-agnostic models, such as SwinIR (Liang et al., 2021), NAFNet (Chen et al., 2022), and Restormer (Zamir et al., 2022), which become strong baselines in various image restoration tasks.

**PEFT-based All-in-One Image Restoration** To adapt to various corruptions, pioneer all-in-one image restoration methods (Potlapalli et al., 2023; Zhang et al., 2023a; Valanarasu et al., 2022; Li et al., 2022a; Xiangtao et al., 2024; Ai et al., 2024b) integrate PEFT-based techniques (Brian et al., 2021; Houlsby et al., 2019) and feature modulation (Zhang et al., 2018) into the restoration backbone. For example, TransWeather (Valanarasu et al., 2022) and AirNet (Li et al., 2022a) learn degradation-related embeddings as prompts to modulate deep features based on different corrupted inputs. Similarly, PromptIR (Potlapalli et al., 2023) and IDR (Zhang et al., 2023a) employ prompt learning in a more powerful backbone (Zamir et al., 2022) and achieve state-of-the-art performance. MiOIR (Xiangtao et al., 2024) further exploits the benefits of incorporating explicit prompt and adaptive prompt learning. OneRestore (Guo et al., 2024) learns text-aligned prompts and achieves

good results in restoring mixed-degradation. However, prompt learning-based techniques are still limited in performance and task range.

**Expert-based All-in-One Image Restoration** Mixture-of-Experts (MoE) has proven to be a solution to multi-task learning due to its conditional processing capability (Chen et al., 2023b;a; Zhu et al., 2022). Recent MoE-based all-in-one image restoration methods can be categorized into implicit MoE (Yang et al., 2023; Conde et al., 2024; Xu et al., 2024; Cao et al., 2024a) and explicit MoE (Zhu et al., 2023; Park et al., 2023; Zamfir et al., 2024; Cao et al., 2024b; Tian et al., 2024; Chen et al., 2024; Ai et al., 2024a; Zamfir et al., 2025). The experts in the former methods are implicitly learned and do not correspond to a specific task. The explicit MoE methods learn experts for specific tasks, unleashing the power of additionally increased parameters. Based on a shared backbone, WGWSNet (Zhu et al., 2023) and ADMS (Park et al., 2023) learn specific filters for specific tasks. To be more efficient, DaAIR (Zamfir et al., 2024), InstructIPT (Tian et al., 2024), and LoRA-IR (Ai et al., 2024a) learn individual LoRA weights or adapters for each task. MoCE-IR (Zamfir et al., 2025) learns complexity-aware experts for different tasks. By combining knowledge of the shared backbone and task-specific modules, these methods can adapt to different tasks effectively. In addition, GRIDS (Cao et al., 2024b) and RestoreAgent (Chen et al., 2024) divide the complete degradation space into subspaces and learn a non-shared full-parameter model for each subspace to alleviate potential conflicts between different degradations.

However, these methods mostly learn DRs at a single, coarse granularity and rely solely on degradation estimation as the routing condition, making them vulnerable to estimation errors. In contrast, our approach explores DRs across multiple granularities, jointly utilizing coarse levels (the same as task-agnostic settings) and finer levels beyond those used in existing methods. The integration of granularity and degradation estimation in the routing process enables more robust expert selection.

## 3 PROPOSED METHOD

In this section, we introduce our all-in-one image restoration framework, UniRestorer. We first optimize one degradation representation (DR) extractor that learns DRs at a fine-grained level. Based on DRs extracted from all low-quality training data, we conduct hierarchical data clustering that separates DRs into multiple DR groups at different granularities (see Fig. 3), leading to a multi-granularity degradation set and MoE restoration model. To alleviate the effect of degradation estimation error on routing, we adopt both degradation and granularity estimation to allocate the most appropriate expert to given unknown corrupted inputs.

### 3.1 PRELIMINARIES

**Mixture-of-Experts (MoE)** MoE is a promising way for scaling up and deploying large or gigantic models due to its competitive computational bounds and latency compared to the vanilla single model. Generally, the MoE architecture consists of two key components: a set of experts $\mathbb{F}$ and a routing network $\mathcal{G}$. Specifically, the expert set $\mathbb{F} = \{\mathcal{F}_1, \cdots, \mathcal{F}_n\}$, contains $n$ parallel modules (*e.g.*, convolution layers or feed-forward layers). Given input $\boldsymbol{x}$, the routing $\mathcal{G}$ is used to calculate the contribution $\mathcal{G}^i(\boldsymbol{x})$ of each expert $\mathcal{F}_i(\boldsymbol{x})$ to the MoE output $\boldsymbol{y}$,

$$\boldsymbol{y} = \sum_{i=0}^{n} \mathcal{G}^i(\boldsymbol{h}) \, \mathcal{F}_i(\boldsymbol{x}), \tag{1}$$

where $\boldsymbol{h}$ is one representation of $\boldsymbol{x}$. $\mathcal{G}$ is typically a noisy top-$k$ network parameterized with $\mathbf{W}_g$ and $\mathbf{W}_n$, which models $P(\mathcal{F}_i|\boldsymbol{x})$ as the probability of using the $i$-th expert $\mathcal{F}_i$ and selects experts with top-$k$ probability to contribute to the output. The noisy routing process can be described as follows,

$$\mathcal{G}(\boldsymbol{h}) = \text{Top-K}\big(\text{Softmax}\big(\boldsymbol{h}\mathbf{W}_g + \epsilon \, \text{Softplus}(\boldsymbol{h}\mathbf{W}_n)\big)\big), \quad \epsilon \sim \mathcal{N}(0, 1). \tag{2}$$

where $\text{Top-K}(\cdot)$ sets all elements in the vector to zero except the elements with the largest $k$ values, $\text{Softplus}$ is the smooth approximation to the function ReLU.

### 3.2 MULTI-GRANULARITY DEGRADATION AND EXPERT

**Fine-Grained DR Extractor** Considering the convenience of using text to represent the type and degree of degradation, we adopt a CLIP-based method, DA-CLIP (Luo et al., 2024), as our basic

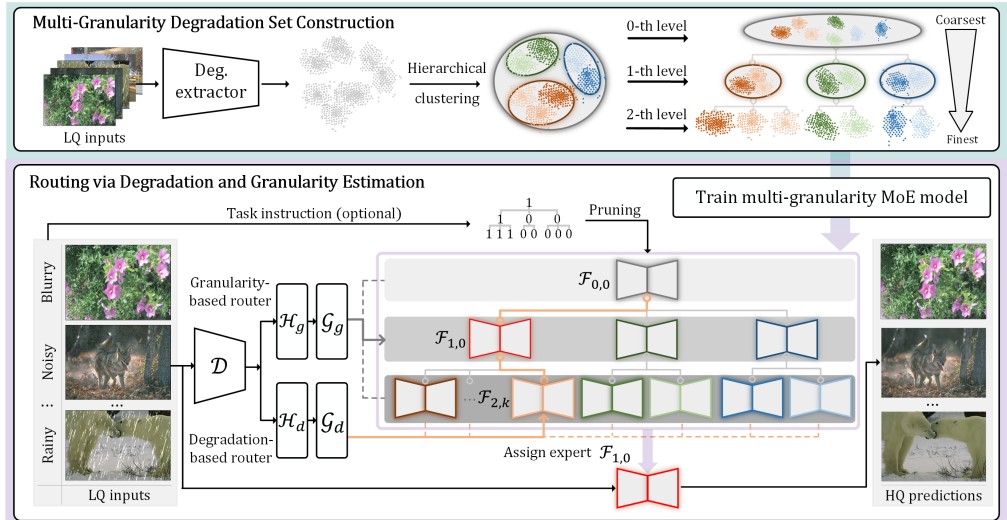

Figure 3: **Illustration of our proposed UniRestorer**. We develop a multi-granularity degradation set by hierarchical clustering on extracted DRs at different granularities. Based on the multi-granularity degradation set, we train a multi-granularity MoE restoration model. Besides vanilla degradation estimation, we introduce granularity estimation to indicate the degree of degradation estimation error. Based on both degradation and granularity estimation, the router can adaptively allocate an appropriate expert for unknown corrupted input.

backbone of the DR extractor. Using contrastive learning, DA-CLIP trains an image controller to distinguish different corruptions. In this work, we aim to enable the DR extractor to be aware of finer-grained DRs. To this end, we construct fine-grained degradation captions for low-quality images, refining the degradation type into the degradation degree captions. Take Gaussian noise as an example, we divide it with level in range of $[0, 50]$ into $[0, 15)$, $[15, 35)$, and $[35, 50]$, described as 'small noise', 'medium noise', and 'large noise', respectively. We then use BLIP (Li et al., 2022b) to generate captions for clean images. By combining low-quality images, clean images, as well as their content and degradation captions, we can train a fine-grained DR extractor. Given low-quality input images $\mathbf{X}$, the fine-grained DR $e \in \mathbb{R}^p$ can be obtained by the DR extractor $\mathcal{D}$, *i.e.*,

$$e = \mathcal{D}(\mathbf{X}). \qquad (3)$$

When training is finished, we employ $\mathcal{D}$ to obtain the DR set $\{e_j\}_{j=0}^N$ from $N$ training samples.

**Hierarchical Degradation Clustering** We adopt an unsupervised data clustering approach (Mitchell, 1997) to construct the multi-granularity degradation set. The training samples are hierarchically clustered from top to down, thus constructing a hierarchical degradation set with $l$ levels of granularities ($l = 3$). The coarsest level is 0-th level and the finest level is $(l-1)$-th level. Each level of granularity consists of non-overlapped DR groups, and samples drawn from a DR group have similar DRs, as shown in Fig. 3. Specifically, 1-th degradation set can be obtained from the coarsest-grained (0-th) level set. First, we cluster the 0-th level set, *i.e.*,

$$\{u_{1,i}\}_{i=0}^m \leftarrow \text{K-means}(\{e_j\}_{j=0}^N), \qquad (4)$$

where $\{u_{1,i}\}_{i=0}^m$ is cluster centroids. Then, we use function $\text{argmin}_{u \in \{u_{1,i}\}_{i=0}^m}(||e_j - u||_2^2)$ to decide which cluster the sample $e_j$ belong to, and assign the sample to corresponding cluster. All samples in the same cluster form a DR group. By analogy, we conduct the same process based on newly introduced DR groups until the finest-grained level.

**Multi-Granularity MoE Restoration Model** We train a multi-granularity MoE restoration model based on the multi-granularity degradation set, where each expert corresponds to a specific DR group. Expert networks are trained with $\ell_1$ loss as the reconstruction loss between predicted results $\hat{\mathbf{Y}}$ and target $\mathbf{Y}$, *i.e.*,

$$\ell_1 = ||\hat{\mathbf{Y}} - \mathbf{Y}||_1. \qquad (5)$$

Experts trained at coarser-grained levels can generalize to a wider range of degradations, and experts trained in finer-grained levels can highly improve the restoration performance on degradations that

they are good at. Therefore, given corrupted inputs, we can estimate their DRs and assign them to the most appropriate expert. In implementation, on the one hand, we employ a full-parameter network for each expert and train it from scratch, which follow prior works such as GRIDS (Cao et al., 2024b) and RestoreAgent (Chen et al., 2024). On the other hand, we train a LoRA (Hu et al., 2022) version derived from the coarsest-grained expert as the restoration model of other experts, which aims to reduce the redundancy of parameters. More details can be found in Sec. C of *Appendix*.

## 3.3 ROUTING VIA DEGRADATION AND GRANULARITY ESTIMATION

It makes sense to use estimated DRs as condition to assign experts to corrupted inputs. However, degradation estimation is error-prone, especially at the fine-grained level, which may lead to inaccurate routing and thus impede restoration performance. To this end, based on the constructed multi-granularity degradation set, we estimate image degradation at the proper granularity level, and perform routing based on both degradation and granularity estimation, which makes our method robust to degradation estimation error.

Specifically, we train two separate branches (*i.e.*, $\mathcal{H}_d$ and $\mathcal{H}_g$) upon $\mathcal{D}$ for generating degradation estimation vector $\boldsymbol{e}_d \in \mathbb{R}^p$ at the finest-grained level and granularity estimation vector $\boldsymbol{e}_g \in \mathbb{R}^p$. Then, we train two routers (*i.e.*, $\mathcal{G}_d$ and $\mathcal{G}_g$) as Eq. (2) based on degradation and granularity estimation, respectively. To clarify the routing process, we denote $\mathcal{F}_{i,j}$ be the expert trained on $j$-th set of the $i$-th level, $\mathcal{P}^{(k)}$ be the path from the $k$-th leaf to the root. Thus, $\mathcal{P}^{(k)}$ can cover experts $\{\mathcal{F}_{i,j_i^k}\}_{i=0}^{l-1}$ from different granularities, where $j_i^k$ is the index of the selected expert on $i$-th level in path $\mathcal{P}^{(k)}$. $\mathcal{G}_d$, the degradation estimation-based routing, is conducted at the finest-grained level. It is used to find which fine-grained group the current DR possibly belongs to. Based on $\mathcal{G}_d$, we can obtain a set of experts in $\mathcal{P}^{(k)}$ (as the orange solid line in Fig. 3), which can described as,

$$\{\mathcal{F}_{0,j_0^k}, \mathcal{F}_{1,j_1^k}, \ldots, \mathcal{F}_{l-1,j_{l-1}^k}\} = \mathrm{argmax}(\mathcal{G}_d(\boldsymbol{e}_d)|\mathbb{F}). \tag{6}$$

Further, based on $\mathcal{G}_g$, we can obtain a set of experts at $t$-th level (as the gray solid line in Fig. 3), which can described as,

$$\{\mathcal{F}_{t,0}, \mathcal{F}_{t,1}, \ldots, \mathcal{F}_{t,m_t}\} = \mathrm{argmax}(\mathcal{G}_g(\boldsymbol{e}_g)|\mathbb{F}), \tag{7}$$

where $m_t + 1$ is the number of all experts in $t$-th level. The routing result is the intersection of the above two routing processing,

$$\mathcal{F} = \{\mathcal{F}_{0,j_0^k}, \mathcal{F}_{1,j_1^k}, \ldots, \mathcal{F}_{l-1,j_{l-1}^k}\} \cap \{\mathcal{F}_{t,0}, \mathcal{F}_{t,1}, \ldots, \mathcal{F}_{t,m_t}\} \tag{8}$$

Inspired by data uncertainty learning (Chang et al., 2020), we learn $\boldsymbol{e}_d$ and $\boldsymbol{e}_g$ by introducing loss as,

$$\mathcal{L}_{dg}(\boldsymbol{e}_d, \boldsymbol{e}_g) = \mathbb{E}\left[\frac{1}{2\,\boldsymbol{e}_g}\big\|\boldsymbol{u} - \boldsymbol{e}_d\big\|^2 + \frac{1}{2}\log\boldsymbol{e}_g\right], \tag{9}$$

where $\boldsymbol{u} \in \mathbb{R}^p$ is the cluster centroids in the finest-grained level. The loss builds the relationships of $\boldsymbol{e}_d$ and $\boldsymbol{e}_g$. When the distance between current degradation estimation and corresponding weight center is high, a larger $\boldsymbol{e}_g$ is used to ensure that $\mathcal{L}_{dg}$ is not too high. Therefore, the granularity can reflect the error of current degradation estimation. For stable training, we clip the value of $\boldsymbol{e}_g$ to a minimum value of 0.01. Further, we adopt load balancing loss (Shazeer et al., 2017) $\mathcal{L}_{aux}$ as auxiliary loss. Finally, we train our routers by

$$\mathcal{L}_{total} = \ell_1 + \alpha\,\mathcal{L}_{dg} + \beta\,\mathcal{L}_{aux}, \tag{10}$$

where $\alpha$ and $\beta$ are two hyper-parameters to trade-off the effect of $\mathcal{L}_{dg}$ and $\mathcal{L}_{aux}$. The detailed theoretical analysis of $\mathcal{L}_{dg}$ can e seen in Sec. A of *Appendix*.

**Instruction Mode** Our framework can automatically conduct the processing pipeline of extraction-recognition-routing-restoration (*i.e.*, *auto* mode). When users are convinced about the type of degradation, they can also provide the type of degradation to instruct the routing process (*i.e.*, *instruction* mode), as shown in Fig. 3. The instruction plays the role of pruning, it utilizes a set of masks to prune the corresponding DR groups from the multi-granularity degradation set, leading to a more efficient and accurate routing.

Table 1: **All-in-one image restoration (single-degradation) results**. PSNR (dB, ↑) and SSIM (↑) are reported in RGB color spaces. The **best** performance is highlighted. '$n$T' means the average performance of $n$ tasks (3T: Derain, Dehaze, Denoise; 5T: Derain, Dehaze, Denoise, Deblur, Lowlight; 7T: all).

| Methods | Derain Rain100L | | Dehaze SOTS | | Denoise BSD68 | | Deblur GoPro | | Lowlight LOLv1 | | Desnow Snow100K | | CAR LIVE1 | | Average 3T | 5T | 7T |
|---|---|---|---|---|---|---|---|---|---|---|---|---|---|---|---|---|---|
| MPRNet (Zamir et al., 2021) | 25.56 | .903 | 16.37 | .806 | 23.92 | .702 | 25.40 | .858 | 7.75 | .124 | 19.33 | .740 | 27.77 | .889 | 18.51 | 22.68 | 19.66 |
| SwinIR (Liang et al., 2021) | 21.20 | .837 | 15.94 | .775 | 24.12 | .780 | 16.86 | .735 | 13.80 | .409 | 18.60 | .718 | 15.61 | .746 | 17.55 | 17.09 | 18.45 |
| NAFNet (Chen et al., 2022) | 31.32 | .967 | 22.43 | .907 | 30.27 | .918 | 26.53 | .877 | 23.09 | .796 | 26.33 | .889 | 31.26 | .946 | 24.55 | 25.76 | 26.28 |
| Restormer (Zamir et al., 2022) | 34.25 | .981 | 29.64 | .971 | 31.00 | .929 | 27.50 | .896 | 22.80 | .821 | 28.24 | .914 | 31.21 | .946 | 30.46 | 28.56 | 28.27 |
| DL-3T (Fan et al., 2019) | 32.62 | .931 | 26.92 | .391 | 30.12 | .838 | – | – | – | – | – | – | – | – | 28.09 | – | – |
| AirNet-3T (Li et al., 2022a) | 34.90 | .967 | 27.94 | .962 | 31.06 | .872 | – | – | – | – | – | – | – | – | 29.29 | – | – |
| PromptIR-3T (Potlapalli et al., 2023) | 36.37 | .972 | 30.58 | .974 | 31.12 | .873 | – | – | – | – | – | – | – | – | 31.50 | – | – |
| InstructIR-3T (Conde et al., 2024) | 37.98 | .978 | 30.22 | .959 | 31.32 | .875 | – | – | – | – | – | – | – | – | 31.49 | – | – |
| DaAIR-3T (Zamfir et al., 2024) | 37.10 | .978 | 32.30 | .981 | 31.06 | .868 | – | – | – | – | – | – | – | – | 32.51 | – | – |
| PromptIR-TUR-3T (Wu et al., 2025) | 38.57 | .984 | 31.17 | .978 | 31.19 | .872 | – | – | – | – | – | – | – | – | 32.67 | – | – |
| MoCEIR-3T (Zamfir et al., 2025) | 38.57 | .984 | 31.34 | .979 | 31.24 | .873 | – | – | – | – | – | – | – | – | 32.73 | – | – |
| DL-5T (Fan et al., 2019) | 21.96 | .762 | 20.54 | .826 | 23.09 | .745 | 19.86 | .672 | 19.83 | .712 | – | – | – | – | 21.02 | 20.28 | – |
| Transweather-5T (Valanarasu et al., 2022) | 29.43 | .905 | 21.32 | .885 | 29.00 | .841 | 25.12 | .757 | 21.21 | .792 | – | – | – | – | 23.31 | 24.41 | – |
| TAPE-5T (Liu et al., 2022a) | 29.67 | .904 | 22.16 | .861 | 30.18 | .855 | 24.47 | .763 | 18.97 | .621 | – | – | – | – | 24.10 | 24.28 | – |
| AirNet-5T (Li et al., 2022a) | 32.98 | .951 | 21.04 | .884 | 30.91 | .882 | 24.35 | .781 | 18.18 | .735 | – | – | – | – | 23.82 | 24.10 | – |
| IDR-5T (Zhang et al., 2023a) | 35.65 | .965 | 25.20 | .938 | 31.09 | .883 | 26.65 | .810 | 20.70 | .820 | – | – | – | – | 27.36 | 26.86 | – |
| InstructIR-5T (Conde et al., 2024) | 36.84 | .973 | 27.10 | .956 | 31.40 | .887 | 29.40 | .886 | 23.00 | .836 | – | – | – | – | 28.99 | 29.19 | – |
| DaAIR-5T (Zamfir et al., 2024) | 36.28 | .975 | 31.97 | .980 | 31.07 | .878 | 29.51 | .890 | 22.38 | .825 | – | – | – | – | 32.20 | 30.24 | – |
| Transweather-TUR-5T (Wu et al., 2025) | 33.09 | .952 | 29.68 | .966 | 30.40 | .869 | 26.63 | .815 | 23.02 | .838 | – | – | – | – | 29.96 | 30.31 | – |
| DCPT-PromptIR-5T (JiaKui et al., 2025) | 37.32 | .978 | 30.72 | .977 | 31.32 | .885 | 28.84 | .877 | 23.35 | .840 | – | – | – | – | 31.46 | 30.31 | – |
| MoCEIR-5T (Zamfir et al., 2025) | 38.04 | .982 | 30.48 | .974 | 31.34 | .887 | 30.05 | .899 | 23.00 | .852 | – | – | – | – | 31.39 | 30.58 | – |
| UniRestorer (Ours) | **41.68** | **.996** | **36.44** | **.984** | 31.43 | .936 | **31.48** | **.948** | 26.67 | **.863** | 31.12 | **.940** | 30.57 | .926 | **36.71** | **33.38** | **32.77** |
| UniRestorer-LoRA (Ours) | 40.22 | .991 | 34.94 | .971 | 31.39 | **.937** | 30.86 | .933 | 24.42 | .827 | 30.53 | .925 | 30.31 | .922 | 35.51 | 32.36 | 31.81 |

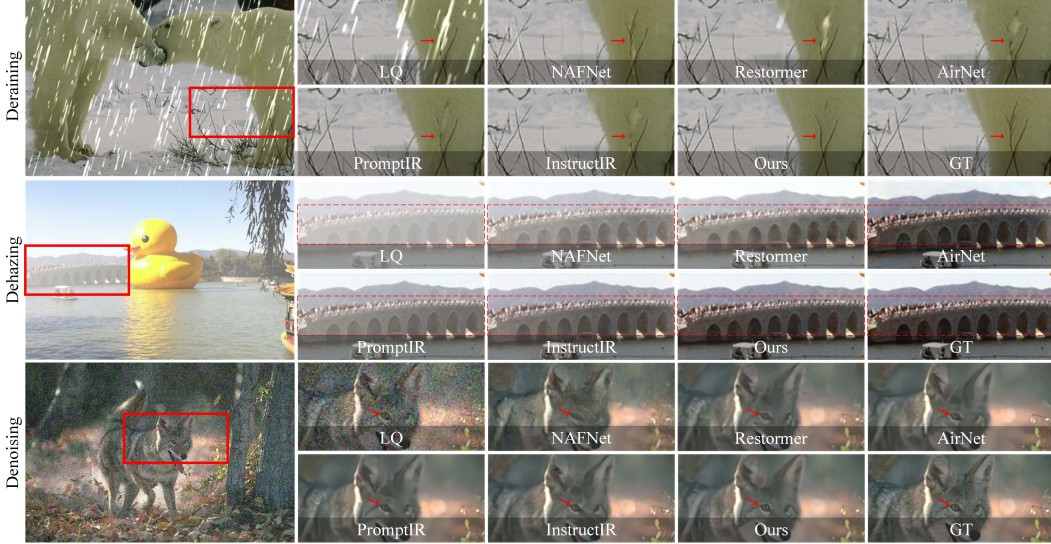

Figure 4: **Visual comparison of results on all-in-one image restoration (single-degradation)**. Our method can effectively remove the degradations (*i.e.*, rainstreak, haze, and noise), restore clearer texture (*i.e.*, 'branches' and 'eyes' pointed by red arrows) and closer color (*i.e.*, 'bridge'). More results can be seen in *Appendix*.

# 4 EXPERIMENTS

## 4.1 EXPERIMENTAL SETUP

We train and evaluate our framework on single-degradation and mixed-degradation scenarios. The usage of datasets can be checked in Tab. A of *Appendix*. In comparison, we only evaluate on tasks of the methods support. More details of datasets, degradation pipeline, experimental setups, and training details can be seen in Secs. B and C of *Appendix*.

## 4.2 COMPARISON WITH STATE-OF-THE-ART METHODS

**Comparison with All-in-One Models** As shown in Tab. 1 and Fig. 4, our method can significantly outperform task-agnostic and all-in-one methods by a large margin in single-degradation. In mixed-degradation (Tab. 2), our method continuously achieves higher performance than other methods from

Table 2: **All-in-one image restoration (mixed-degradation) results on CDD-11 dataset**. PSNR (dB, ↑) and SSIM (↑) are reported in RGB color spaces. The **best** performance is highlighted.

| Methods | L | | H | | R | | S | | L-H | | L-R | | L-S | | H-R | | H-S | | L-H-R | | L-H-S | | Average |
|---|---|---|---|---|---|---|---|---|---|---|---|---|---|---|---|---|---|---|---|---|---|---|---|
| AirNet (Li et al., 2022a) | 24.83 | .778 | 24.21 | .951 | 26.55 | .891 | 26.79 | .919 | 23.23 | .779 | 22.82 | .710 | 23.29 | .723 | 22.21 | .868 | 23.29 | .901 | 21.80 | .708 | 22.24 | .725 | 23.75 |
| PromptIR (Potlapalli et al., 2023) | 26.32 | .805 | 26.10 | .969 | 31.56 | .946 | 31.53 | .960 | 24.49 | .789 | 25.05 | .771 | 24.51 | .761 | 24.54 | .924 | 23.70 | .925 | 23.74 | .752 | 23.33 | .747 | 25.90 |
| WGWSNet (Zhu et al., 2023) | 24.39 | .774 | 27.90 | .982 | 33.15 | .964 | 34.43 | .973 | 24.27 | .800 | 25.06 | .772 | 24.60 | .765 | 27.23 | .955 | 27.65 | .960 | 23.90 | .772 | 23.97 | .771 | 26.96 |
| WeatherDiff (Özdenizci & Legenstein, 2023) | 23.58 | .763 | 21.99 | .904 | 24.85 | .885 | 24.80 | .888 | 21.83 | .756 | 22.69 | .730 | 22.12 | .707 | 21.25 | .868 | 21.99 | .868 | 21.23 | .716 | 21.04 | .698 | 22.49 |
| OneRestore (Guo et al., 2024) | 26.48 | .826 | 32.52 | .990 | 33.40 | .964 | 34.31 | .973 | 25.79 | .822 | 25.58 | .799 | 25.19 | .789 | 29.99 | .957 | 30.21 | .964 | 24.78 | .788 | 24.90 | .791 | 28.47 |
| MoCE-IR-S (Zamfir et al., 2025) | 27.26 | .824 | 32.66 | .990 | 34.31 | .970 | 35.91 | .980 | 26.24 | .817 | 26.25 | .800 | 26.04 | .793 | 29.93 | .964 | 30.19 | .970 | 25.41 | .789 | 25.39 | .790 | 29.05 |
| UniRestorer (Ours) | **28.28** | **.905** | 35.29 | **.992** | 37.29 | **.979** | 38.02 | **.985** | 27.74 | **.900** | 27.87 | **.902** | 27.56 | **.899** | 33.14 | **.945** | 31.85 | **.934** | 26.36 | **.889** | 26.49 | **.890** | **30.90** |
| UniRestorer-LoRA (Ours) | 27.71 | .900 | 34.53 | .991 | 36.07 | .972 | 36.84 | .983 | 26.88 | .893 | 26.93 | .894 | 26.73 | .892 | 32.02 | .936 | 30.82 | .976 | 25.78 | .884 | 25.87 | .885 | 30.02 |

Table 3: **Generalization performance on real-world datasets**. All methods are evaluated without additional training. PSNR (dB, ↑) is reported in RGB-channels.

| Methods | Deraining | Low-light | Desnowing | Average |
|---|---|---|---|---|
| MPRNet (Zamir et al., 2021) | 29.16 | 9.67 | 25.82 | 27.10 |
| SwinIR (Liang et al., 2021) | 21.74 | 18.67 | 18.45 | 20.91 |
| NAFNet (Chen et al., 2022) | 26.04 | 26.46 | 27.51 | 26.33 |
| Restormer (Zamir et al., 2022) | 27.23 | 28.92 | 26.73 | 27.26 |
| PromptIR (Potlapalli et al., 2023) | 25.68 | - | - | - |
| MiOIR (Xiangtao et al., 2024) | 28.69 | 10.62 | - | - |
| InstructIR (Conde et al., 2024) | 28.93 | 30.31 | - | - |
| OneRestore (Guo et al., 2024) | 24.55 | 17.76 | 20.12 | 23.24 |
| UniRestorer (Ours) | **30.04** | **30.40** | **28.04** | **29.70** |
| UniRestorer-LoRA (Ours) | 29.88 | 30.35 | 27.73 | 28.65 |

Table 4: **Generalization performance on unseen corruptions**. All methods are evaluated without training on corresponding datasets. PSNR (dB, ↑) is reported in RGB-channels.

| Methods | Raindrop | UDC | Underwater |
|---|---|---|---|
| NAFNet (Chen et al., 2022) | 22.89 | 26.89 | 15.54 |
| Restormer (Zamir et al., 2022) | 23.34 | 28.44 | 15.80 |
| PromptIR (Potlapalli et al., 2023) | 22.98 | 25.02 | 16.12 |
| MiOIR (Xiangtao et al., 2024) | 23.59 | 20.11 | 16.26 |
| OneRestore (Guo et al., 2024) | 22.02 | 15.16 | 15.87 |
| UniRestorer (Ours) | **24.91** | **29.64** | **18.07** |
| UniRestorer-LoRA (Ours) | 24.43 | 28.92 | 17.55 |

Table 5: **Compare to single-task image restoration models**. PSNR (dB, ↑) are reported in RGB color spaces. The **best** performance is highlighted. The task-agnostic methods are trained from scratch on each task. † denotes *instruction* mode.

| Methods | Derain Rain100L | Denoise BSD68 | | | Desnow Snow100K | | | CAR. LIVE1 | | | | Lowlight LOLv1 | Deblur GoPro | Dehaze SOTS |
|---|---|---|---|---|---|---|---|---|---|---|---|---|---|---|
| | | $\sigma=15$ | $\sigma=25$ | $\sigma=50$ | S | M | L | $q=10$ | $q=20$ | $q=30$ | $q=40$ | | | |
| MPRNet (Zamir et al., 2021) | 35.21 | 34.91 | 32.67 | 29.62 | 36.22 | 34.49 | 30.63 | 27.87 | 30.23 | 31.57 | 32.50 | 23.51 | 32.66 | 31.07 |
| SwinIR (Liang et al., 2021) | 37.70 | 34.81 | 32.51 | 29.29 | 33.45 | 32.00 | 28.40 | 26.16 | 28.62 | 29.93 | 30.81 | 19.87 | 29.09 | 28.35 |
| Restormer (Zamir et al., 2022) | 39.60 | 34.92 | 32.70 | 29.70 | 36.50 | 34.77 | 30.86 | 27.89 | 30.21 | 31.56 | 32.48 | 23.66 | 32.92 | 33.18 |
| NAFNet (Chen et al., 2022) | 38.36 | 34.80 | 32.58 | 29.50 | 35.94 | 33.95 | 29.50 | **27.95** | 30.24 | 31.56 | 32.47 | 23.75 | 33.69 | 22.70 |
| UniRestorer†(Ours) | **41.68** | **35.00** | **32.75** | **29.75** | **36.94** | **35.18** | **31.14** | 27.94 | **30.31** | **31.66** | **32.58** | **27.25** | **33.71** | **37.45** |
| UniRestorer-LoRA†(Ours) | 40.22 | 34.92 | 32.73 | 29.70 | 36.65 | 34.85 | 30.80 | 27.84 | 30.20 | 31.58 | 32.46 | 25.00 | 33.54 | 35.32 |

simple to complex scenarios. In addition, comparisons on *real-world* image restoration datasets (Tab. 3) and *unseen* degradations (Tab. 4) reveal that our method also guarantees better generalization performance than others. We found that it becomes challenging for all-in-one models to achieve significant performance gains over SOTA task-agnostic methods as the number of restoration tasks increases, suggesting that the use of tailored prompts or expert modules may have limited impact during training. This can be attributed to the used trivial degradation representations, which may fail to effectively guide methods to learn different restoration process. More comparisons on real-world and unseen degradations can be seen in Tab. E of the *Appendix*.

**Comparison with Single-Task Models** For a fair comparison, we evaluate the performance of our method in *instruction* mode (Ours†). The results in Tab. 5 show that our method achieved competitive or better performance compared to single-task models across various image restoration tasks. The main reason lies in the fact that when single-task models are required to handle a broader range of degradation patterns (*e.g.*, all levels of snow), their performance on specific degradation (*e.g.*, large snow) tends to be not guaranteed. Our method can learn image degradations in finer-grained levels, allowing each expert to focus on specific degradation (*e.g.*, different levels of snow). As a result, our approach can achieve competitive or better performance compared to single-task models.

**Overhead Comparisons** We calculate FLOPs and latency with $512 \times 512 \times 3$ input, on a single NVIDIA A6000 GPU. As shown in Tab. 6, thanks to mixture-of-experts, the overhead of our method is slightly higher than the used image restoration backbone, Restormer (Zamir et al., 2022).

Since our method is flexible, we can replace restoration network backbone with a more efficient one for fitting computational cost requirements. We replace the used Restormer with lightweight NAFNet (LiteNAF.) and lightweight Restormer (LiteRes.), the performance and training cost comparison is reported in 3-task all-in-one image restoration task. As shown in Table 7, Ours-LiteNAF. achieves fast inference speed and lower training cost than PromptIR, and its restoration performance outperforms

Table 6: Overhead comparisons to SOTA methods.

| Methods | #Params. (M) | #FLOPs (G) | Latency (s) |
|---|---|---|---|
| Restormer (Zamir et al., 2022) | 26 | 1128.9 | 0.368 |
| NAFNet (Chen et al., 2022) | 17 | 505.5 | 0.186 |
| PromptIR (Potlapalli et al., 2023) | 36 | 1266.2 | 0.355 |
| AirNet (Li et al., 2022a) | 9 | 2414 | 0.548 |
| DCPT-PromptIR (JiaKui et al., 2025) | 36 | 1266.2 | 0.355 |
| MoCE-IR (Zamfir et al., 2025) | 25 | 716.16 | 0.347 |
| UniRestorer (Ours) | 627 | 1155.8 | 0.484 |
| UniRestorer-LoRA (Ours) | 171 | 1155.8 | 0.484 |

Table 7: Performance and overhead comparison to PromptIR with our lightweight variants in three-task all-in-one image restoration setups.

| Methods | Derain Rain100L | Dehaze SOTS | Denoise BSD68 | #FLOPs (G) | Latency (s) | Params.(M) Inf./Train | Inf. Mem. (M) | Time(h) Train |
|---|---|---|---|---|---|---|---|---|
| PromptIR (Potlapalli et al., 2023) | 36.37 | 30.58 | 31.12 | 1266.2 | 0.355 | 39.6/39.6 | 3552 | 201 |
| Ours | 41.68 | 36.44 | 31.43 | 1155.8 | 0.484 | 26.1/339.3 | 4258 | 578 |
| Ours-LiteNAF. | 38.98 | 31.69 | 31.28 | 154.7 | 0.156 | 17.1/222.4 | 1818 | 169 |
| Ours-LiteRes. | 41.41 | 35.38 | 31.35 | 672.7 | 0.395 | 14.8/192.5 | 3526 | 442 |

PromptIR in the 3-task. This demonstrates that the variant of our method not only improves efficiency but also maintains superior restoration performance, highlighting that we can change the restoration backbone to fit the requirement of training cost in practice.

## 4.3 ABLATION STUDY

Since the effectiveness of our method reflects on two aspects: in-distribution (*in-dist.*) and out-of-distribution (*out-dist.*) performance, we train and evaluate our method on synthesized low-quality data, where degradation parameters can be adjusted to determine whether the input samples fall into the *in-dist.* or *out-dist.* Specifically, we train our method on DIV2K (Agustsson & Timofte, 2017) and Flickr2K (Lim et al., 2017) datasets with hand-crafted degradation pipeline, the details can be found in Sec. B. We conduct comparisons on DIV2K-valid and Urban100 (Nah et al., 2017b) datasets, the results are in Tab. C and Tab. D of *Appendix*, respectively. For quick evaluation, we only evaluate the performance on Urban100 dataset in ablation studies.

**Effect of DR Fineness** By utilizing $\mathcal{G}_d$ and clustering input samples into different numbers of DR groups, we investigate the effect of DR fineness. As shown in the upper part of Tab. 8, the performance on *in-dist.* consistently improves as grouping into finer-grained DR groups. However, due to the increased difficulty of routing, the performance gain plateaus when #DRs exceeds 8. For performance on *out-dist.*, clustering DR groups in proper coarse-grained level performs better than finer-grained levels, suggesting that it is essential to combine multiple granularities to keep high performance on both *in-dist.* and *out-dist.*

**Effect of Multiple Granularities** Utilizing both $\mathcal{G}_d$ and $\mathcal{G}_g$, we investigate the effect of different combinations of granularities. As shown in the lower part of Tab. 8, incorporating both fine-grained levels and coarse-grained levels can maintain much better restoration performance and generalization ability than employing single granularity. Taking performance, training cost, and learning difficulty into account, we finally adopt $\{1, 4, 8\}$ in our mixed-degradation experiments.

**Effect of Granularity Estimation-based Routing** By comparing with 'Vanilla' method, which only routes in the finest level, we investigate the effect of granularity estimation-based routing. Specifically, we normalize degradation parameters in range from *in-dist.* to *out-dist.* into the interval $[0, 1]$. Then, we uniformly sample 15 points in this interval. At each sampled point, we add corresponding degradations to images in Urban100. We compute the average granularity level and average PSNR for each point, and we conduct a comparison to the 'Vanilla'. The visualization of the results is shown in Fig. 5. The results show that, as the degradation shifts from *in-dist.* to *out-dist.*, our method can adaptively select experts in a coarser-grained level, while the 'Vanilla' baseline only uses experts in

Table 8: Ablation study on the effect of DR fineness and the number of granularties.

| Routers | # Levels | { # DRs } | MiO (Average) In-dist. | MiO (Average) Out-dist. |
|---|---|---|---|---|
| $\mathcal{G}_d$ | 1 | { 1 } | 22.06 | 17.23 |
| | 1 | { 2 } | 22.42 | 17.51 |
| | 1 | { 4 } | 23.75 | **18.49** |
| | 1 | { 8 } | 24.22 | 18.35 |
| | 1 | { 16 } | **24.30** | 18.44 |
| $\mathcal{G}_d, \mathcal{G}_g$ | 2 | { 1, 8 } | 24.27 | 18.86 |
| | 2 | { 4, 8 } | 24.44 | 19.06 |
| | 3 | { 1, 4, 8 } | **24.46** | 19.45 |
| | 4 | { 1, 4, 8, 16} | 24.41 | **19.47** |

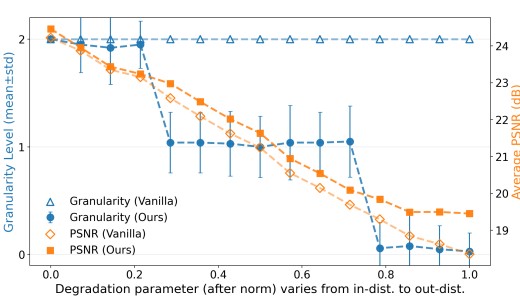

Figure 5: Visualization of routing results on granulairty and PSNR.

the finest-grained level. Benefiting from coarser-grained experts, our method can generalize to a wider range of degradations, achieving better generalization performance in *out-dist.* while maintaining restoration performance in *in-dist.*.

**Effect of Loss Functions**  We conduct three experiments to validate the effectiveness of the used loss functions. By using only $\ell_1$ loss, our method achieves 24.14 dB on *in-dist.* and 18.37 dB on *out-dist.*. Benefits from $\mathcal{L}_{dg}$, our method improves the performance on *in-dist.* to 24.35 dB and improves performance on *out-dist* to 18.76 dB. Adopting $\mathcal{L}_{load}$, the method further has 0.11 dB and 0.29 dB improvement on *in-dist.* and *out-dist.*, respectively. The results show that $\mathcal{L}_{dg}$ and $\mathcal{L}_{load}$ can help to avoid the routing collapse to the finest-grained level, particularly improving performance on *out-dist.* The details can be found in Tab. H of *Appendix*.

**Others**  We provide quantitative analysis of the clustering step in Tab. B. For a fairer comparison, we also train scaling-up SOTA models, and the results are in Tab. F. We provide more analysis about our method, including the effect of DR extractor ( Tab. G), the used loss functions (Tab. H), the effect of input resolution (Tabs. J and I), discussion on clustering method and the hybrid usage (Sec. E). Finally, more visual results can be seen in Sec. H.

### 4.4 ANALYSIS ON ROUTING

We investigate the finest-level clustering results of CDD-11 dataset. The dominant degradation types of the eight clusters are: (1) {Haze+Rain (H+R)}, (2) {Rain (R)}, (3) {Lowlight+Rain (L+R), Lowlight+Haze+Rain (L+H+R)}, (4) {Lowlight+Haze (L+H)}, (5) {Haze+Snow (H+S), Snow (S)}, (6) {Haze (H)}, (7) {Lowlight+Haze+Snow (L+H+S), Lowlight+Snow (L+S)}, (8) {Lowlight (L)}. We further visualize the routing results in CDD-11 testset. The routing results in Fig. D of the *Appendix* show that when degradation is ambiguous, the accurate degradation estimation becomes difficult, our method prefers coarser-grained granularity level to guarantee the expert network can handle the input; when degradation estimation is accurate, our method mostly selects the finest-grained experts for superior performance. More results on routing can be seen in Sec. F of the *Appendix*.

### 5 CONCLUSION

In this paper, we introduced our UniRestorer, a novel universal image restoration framework designed to leverage degradation priors for improved restoration performance while alleviating the inevitable error in degradation estimation. To this end, we learn all-in-one image restoration in multiple granularities and propose to estimate image degradation at proper granularity. Specifically, we perform hierarchical clustering on degradation space and develop a multi-granularity degradation set as well as a MoE restoration model. Besides vanilla degradation estimation, granularity estimation is further introduced to indicate the degree of degradation estimation error. By jointly utilizing degradation and granularity estimation, we train routers to adaptively allocate unknown corrupted inputs to the most appropriate expert. Experimental results demonstrate our superior performance and promising potential in closing the performance gap to specific single-task models.

**Acknowledgements.** This work was partially supported by the National Key RD Program of China under Grant No. 2022YFA1004100, China Postdoctoral Science Foundation under Grant No. 2025M784371, and the National Natural Science Foundation of China (NSFC) under Grant No. 62371164.

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

# APPENDIX

The content of the appendix involves:

- Theoretical analysis on granularity estimation is in Sec. A.

- Datasets, degradation pipeline, and degradation parameters are in Sec. B.

- Implementation and training details are in Sec. C.

- More comparison results are in Sec. D.

- More ablation studies are in Sec. E.

- More analysis on routing are in Sec. F.

- Limitation and future works are in Sec. G.

- More visual comparisons are in Sec. H.

## A  THEORETICAL ANALYSIS

Existing degradation-aware routing methods typically generate a deterministic degradation representation $\mathbf{e}_i \in \mathbb{R}^D$ for the $i$-th sample and use it to determine the expert assigned to that sample. However, in practice, it is challenging to model an accurate degradation representation, $\mathbf{e}_i$. Inaccurate representations of $\mathbf{e}_i$ can lead to the selection of the wrong expert, which severely impacts the restoration performance. Motivated by this issue, in this paper, we utilize a distribution $\mathcal{N}(\mu_i, \sigma_i^2\mathbf{I})$ instead of a deterministic representation to model the degradation, where $\mu_i = \phi_\mu(\mathbf{e}_i) \in \mathbb{R}^D$ and $\sigma_i^2 = \phi_\sigma(\mathbf{e}_i) \in \mathbb{R}^D$ represent the estimated degradation and uncertainty, respectively, and $\mathbf{I}$ is the identity matrix. In our implementation, we learn branch $\mathcal{H}_d$ to estimate $\phi_\mu$ and learn $\mathcal{H}_g$ to estimate $\phi_\sigma$. Given the distribution $\mathcal{N}(\mu_i, \sigma_i^2\mathbf{I})$ and $k$ degradation cluster centers $\mathbf{u} = \{\mathbf{u}_1, \cdots, \mathbf{u}_j, \cdots, \mathbf{u}_k\}$, we can calculate the probability of $i$-th sample belonging to $j$-th cluster as follows,

$$p(\mathbf{u}_j|\mathcal{N}(\mu_i, \sigma_i^2\mathbf{I})) = \frac{1}{\sqrt{2\pi\sigma_i^2}} \exp\left(-\frac{(\mathbf{u}_j - \mu_i)^2}{2\sigma_i^2}\right). \tag{A}$$

Then the log likelihood is as follows,

$$\ln p(\mathbf{u}_j|\mathcal{N}(\mu_i, \sigma_i^2\mathbf{I})) = -\frac{1}{2\sigma_i^2}(\mathbf{u}_j - \mu_i)^2 - \frac{1}{2}\ln\sigma_i^2 - \frac{1}{2}\ln 2\pi. \tag{B}$$

Taking the negative of Eq. (B) and ignoring the constant term, we obtain the used loss function as follows,

$$\mathcal{L}_{\text{dg}} = \frac{1}{2\sigma_i^2}(\mathbf{u}_j - \mu_i)^2 + \frac{1}{2}\ln\sigma_i^2 \tag{C}$$

In Eq. (C), the first term encourages the degradation estimation $\mu_i$ to be close to its corresponding degradation cluster center $\mathbf{u}_j$. The second term serves as the regularization, preventing the estimation uncertainty from becoming too large.

Thus, we can conclude that, for samples with easily estimable degradation, $(\mathbf{u}_j - \mu_i)^2$ is usually small, and $\sigma_i^2$ could also be small; for samples with complex or ambiguous degradations, $(\mathbf{u}_j - \mu_i)^2$ is large, leading to a larger $\sigma_i^2$ being learned to balance the loss value. Based on the degradation estimation $\mu_i$ and the estimated uncertainty $\sigma_i^2$, we construct degradation-based routing and granularity-based routing. Specifically, when the model produces a degradation estimation $\mu_i$ with a small uncertainty $\sigma_i^2$, our model tends to utilize the finer-grained expert to deal with this sample, ensuring restoration accuracy. In contrast, when the model produces a degradation estimation $\mu_i$ with a large uncertainty $\sigma_i^2$, it chooses the coarser-grained expert to deal with this sample, promoting restoration generalization. This design effectively mitigates the issue of restoration failure caused by wrong routing in the existing degradation-aware routing methods.

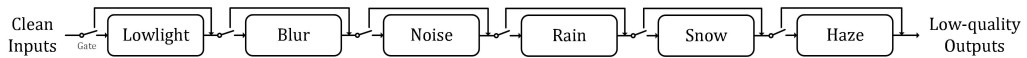

Figure A: Degradation Pipeline.

# B DATA

## B.1 DATASET

For a quick lookup, we list the experimental setups for all reported results in Tab. A. The detailed information of experiments on single-degradation, mixed-degradation, single-task comparison, and ablation studies is as follows: (1) Single-degradation: For training, we adopt Rain100L (Yang et al., 2017) dataset for deraining, RESIDE (Li et al., 2019) dataset for dehazing, DFBW (DIV2K (Agustsson & Timofte, 2017), Flickr2K (Lim et al., 2017), BSD (Martin et al., 2001), and WED (Ma et al., 2017)) datasets for Gaussian color image denoising and compression artifacts removal, GoPro (Nah et al., 2017a) dataset for motion deblurring, LOLv1 (Wei et al., 2018) dataset for low-light enhancement, Snow100K (Liu et al., 2018a) dataset for desnowing. Following (Potlapalli et al., 2023; Zhang et al., 2023a), we evaluate our method on Rain100L (Yang et al., 2017) for deraining, SOTS (Li et al., 2019) for dehazing, BSD68 (Martin et al., 2001) for Gaussian color image denoising, Go-Pro (Nah et al., 2017a) for motion deblurring, LOLv1 (Wei et al., 2018) for low-light enhancement, Snow100K-L (Liu et al., 2018a) for desnowing, and LIVE1 (Sheikh et al.) for compression artifacts removal. (2) Mixed-degradation: We adopt CDD-11 (Guo et al., 2024) as training and evaluation data. For more complex scenarios, we synthesize mixed degradation datasets from clean images in DF2K (DIV2K (Agustsson & Timofte, 2017) and Flickr2K (Lim et al., 2017)) datasets for training, we evaluate the performance on DIV2K-valid and Urban100 datasets . The degradation pipeline is illustrated in Fig. A. (3) Single-task comparison: We use the same datasets as those used in the single-degradation setup.

## B.2 DEGRADATION PIPELINE AND PARAMETERS

We follow the degradation pipeline of (Zhang et al., 2022a) to construct synthetic low-quality datasets for the training of the degradation extractor and all-in-one image restoration (mixed-degradation setup). As shown in Fig. A, given high-quality clean inputs $\mathbf{Y}$, we orderly add synthetic lowlight, blur, noise, rain, snow, and haze to synthesize corresponding low-quality outputs $\mathbf{X}$. We adopt gates between different degradations to gap one of them probably, the probability of all gates is generally set to $0.5$. The way of each degradation is introduced as follows:

**Blur** Following (Zhang et al., 2018), we synthesize blurring degradation as,

$$\mathbf{X} = \mathbf{Y} * \mathbf{K}, \tag{D}$$

where $\mathbf{K}$ is the blur kernel. In our implementation, the type of blur kernel includes isotropic and anisotropic, each of which is employed with the probability of $0.5$. The kernel size is sampled in the range $[7, 23]$. The standard deviation of the Gaussian kernel is set in range $[1, 3]$ for *in-dist.* and $[3, 4]$ for *out-dist.*

**Noise** Gaussian noise is synthesized as,

$$\mathbf{X} = \mathbf{Y} + \mathbf{N}, \tag{E}$$

where $\mathbf{N} \sim \mathcal{N}(0, \sigma^2)$ denotes Gaussian noise. $\sigma$ is set in range $[15, 35]$ for *in-dist.* and $[35, 50]$ for *out-dist.*

**Rain** Raining degradation is synthesized as,

$$\mathbf{X} = \mathbf{Y} + \mathbf{R}, \tag{F}$$

where $\mathbf{R}$ is rain streaks. Following (Xiangtao et al., 2024), $\mathbf{R}$ is synthesized with random noise and Gaussian blur, where noise is related to the strength of the rain, and blur kernel is related to the rain direction, length, and width. We adopt the number of rain streaks range $[50, 100)$ in *in-dist.* and $[100, 150]$ for *out-dist.* Rain direction is generally adopted in range $[-45°, 45°]$, rain length is set in range $[20, 40]$, and rain width is uniformly sampled from $\{3, 5, 7, 9, 11\}$.

Table A: **Setups of all experiments**. We list model, training data, evaluation data, degradation pipeline, and metrics for all reported tables. '–' in training data means directly evaluation without training.

| Table ID | Training data | Evaluation data | Degradation Pipeline | Remark |
|---|---|---|---|---|
| *Single-degradation* | | | | |
| Tab. 1, Tab. 5, Tab. F | Rain100L, RE-SIDE, DIV2K, Flickr2K, WED, BSD, Go-Pro, LOLv1, Snow100K | Rain100L, SOTS, CBSD68, Go-Pro, LOLv1, Snow100K-L, LIVE1 | – | Tab. 1: evaluation of all-in-one image restoration performance in single-degradation setup; Tab. 5: comparing our performance to single-task models; Tab. F: comparing to SOTA methods with parameter scaling up. |
| Tab. 3, Tab. E | Using model same as Tab. 1 | LHP, LOLv2, RealSnow, SIDD, DND, ISP artifact, RealBlur | – | This table is for evaluating the generalization performance on real-world datasets. |
| Tab 4. | Using model same as Tab. 1 | RainDrop, TOLED, UIEB | – | This table is for evaluating the generalization performance on unseen degradations. |
| Tab. J, Tab. I | Using model same as Tab. 1 | DIV2K, 4K-Rain, 4K-Haze, RVDS | Gaussian noise, jpeg compression, lowlight, motionblur | These two table are for validating the resolution effect of input images on the extracted DR. |
| Tab. G | Rain200L, Rain200H, DID, DDN | Rain200L, Rain200H, DID, DDN | – | This table is for validating the effectiveness of our DR extractor. To increase the diversity and complexity of rain degradation, we expand deraining datasets. |
| *Mixed-degradation* | | | | |
| Tab. 2 | CDD-11 | CDD-11 | – | This table is for evaluating the performance of all-in-one image restoration in mixed-degradation setup. |
| Tab. D | DIV2K, Flickr2K | Urban100 | Using pipeline as Fig. A | This table is for evaluating the performance of all-in-one image restoration in mixed-degradation setup. |
| Tab. C | Using model same as Tab. D | DIV2K-valid | Using pipeline as Fig. A | This table is for evaluating the performance of all-in-one image restoration in mixed-degradation setup. |
| Tab 8, Tab. H | Using model same as Tab. D | Urban100 | Using pipeline as Fig. A | We validate the effectiveness of our method on two aspects: in-distribution and out-of-distribution. Tab. 8: validating the effect of DR finess and #granulairty; Tab. H: validating the effect of the used loss functions; |

**Haze** Following (Li et al., 2019), hazing degradation is synthesized as,

$$\mathbf{X} = \mathcal{T}(\mathbf{Y})\,\mathbf{Y} + a\left(1 - \mathcal{T}(\mathbf{Y})\right),$$
$$\mathcal{T}(\mathbf{Y}) = \exp\left(-\beta\,\mathcal{D}(\mathbf{Y})\right), \tag{G}$$

where $a$ is the global atmospheric light, $\mathcal{T}(\mathbf{Y})$ is the transition map, $\mathcal{D}(\mathbf{Y})$ is the depth map, and $\beta$ is the scattering coefficient of the atmosphere. In haze degradation synthesizing, the value of $\beta$ largely

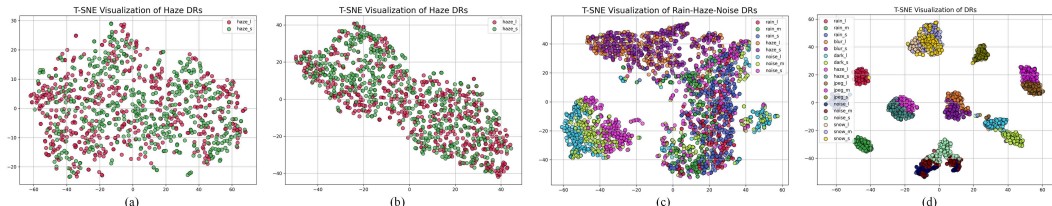

Figure B: T-SNE visualization of deep features (Restormer Zamir et al. (2022), PromptIR Potlapalli et al. (2023)) and degradation representations (Ours). (a) and (b): Restormer and PromptIR are not aware of fine-grained degradation representations. (c) While PromptIR demonstrates a certain capability to distinguish between different types of degradations, it still exhibits misclassification across various cases. (d) Our method distinguish degradation representations well in both coarse-grained and fine-grained levels.

decides the thickness of haze. We adopt one publicly released project $\mathrm{MegaDepth}$[1] to estimate the depth image. In our experiments, the global atmospheric light is set in range $[0.8, 1]$, the $\beta$ is set in range $[0.5, 1.5]$ for *in-dist.* and $[1.5, 2]$ for *out-dist.*

**Snow** Snowing degradation is synthesized as,

$$\mathbf{X} = \mathbf{Y} \odot (\mathbf{1} - \mathbf{M}) + \mathbf{M}, \tag{H}$$

where $\mathbf{M}$ is the snow mask randomly sampled from Snow100K (Liu et al., 2018a) dataset.

**Lowlight** Following (Guo et al., 2024), lowlight degradation is synthesized as,

$$\mathbf{X} = \frac{\mathbf{Y}}{\mathcal{I}(\mathbf{Y})} \odot \mathcal{I}(\mathbf{Y})^{\gamma} \tag{I}$$

where $\mathcal{I}(\mathbf{Y})$ is the illusion map obtained from RetinexFormer (Cai et al., 2023) and $\gamma$ is the darkening coefficient. Since we explicitly include noise degradation in our degradation pipeline, we do not additionally consider noise in this step. We set $\gamma$ in range $[1, 2]$ for *in-dist.* and $[2, 2.5]$ for *out-dist.*

Besides, **JPEG compression** is synthesized by employing the publicly released project DiffJPEG[2].

## C  IMPLEMENTATION AND TRAINING DETAILS

**Degradation Extractor** Our degradation extractor is based on previous work, DA-CLIP (Luo et al., 2024). DA-CLIP uses a lightweight image controller, a copy of the CLIP image encoder augmented with zero-initialized residual connections, to guide the image encoder output toward producing both content-related features and degradation-related embeddings. Using contrastive learning, the content embedding is aligned with clean text descriptions, and degradation embedding indicates the type of degradation, aligned with degradation-related text prompts. For more details, please refer to the original paper. To demonstrate the effectiveness of our degradation extractor, we visualize the T-SNE results in Fig. B (a). For example, in the single dehazing task, both Restormer and PromptIR fail to capture fine-grained DRs, as shown in Fig. B (a) and Fig. B (b). In contrast, our method learns DRs at finer levels of granularity and allows restoration on more specific degradations, thus achieving competitive or even better performance to single-task models. While PromptIR can separate different types of degradations, it still exhibits misidentification across various cases. Such cases may render the dedicated prompts less effective, which are originally as guidance for specific tasks. As a result, PromptIR may offer limited performance improvements over task-agnostic methods in some of the restoration tasks.

**Hierarchical Clustering** We follow SEAL (Wenlong et al., 2024) using silhouette score (Rousseeuw, 1987) to quantitatively evaluate the quality of the clustering, which is defined as follows:

$$S = \frac{1}{N} \sum_{i=1}^{N} s_i, \quad s_i = \frac{b_i - a_i}{\max(a_i, b_i)} \in [-1, 1], \tag{J}$$

---

[1]https://github.com/zhengqili/MegaDepth
[2]https://github.com/mlomnitz/DiffJPEG

Table B: Silhouette score of K-Means clustering.

| {# DRs } | Silhouette Score | | |
|---|---|---|---|
| | 1st-level | 2nd-level | 3rd-level |
| {1, 2} | 0.3103 | – | – |
| {1, 4} | 0.4926 | – | – |
| {1, 8} | 0.6605 | – | – |
| {1, 16} | 0.5932 | – | – |
| {1, 4, 8} | 0.4926 | 0.6686 | – |
| {1, 4, 8, 16} | 0.4926 | 0.6686 | 0.5525 |

where $a_i$ denotes the average intra-cluster distance of $i$-th sample, and $b_i$ denotes the minimum average distance between $i$-th sample and all points in the nearest neighboring cluster. As indicated by Eq. (J), a higher silhouette score reflects a more compact and better-separated clustering structure. Specifically, we calculate the silhouette score in mixed-degradation setup. As shown in Table B, when the number of DR groups increases from 2 to 8, the silhouette score improves. This is because a finer partition allows degradation patterns in each group to be more consistent. However, further increasing the number of DR groups from 8 to 16 does not yield additional improvement. This suggests that overly fine-grained DR groups may introduce redundant or noisy partitions, leading to less meaningful separation and consequently limiting the clustering quality. Besides, the setting of $\{1, 4, 8\}$ produces a higher silhouette score than that of $\{1, 8\}$. This indicates that introducing an intermediate degradation level helps the model better capture the continuous transition among degradation patterns, leading to more compact intra-group structures and clearer inter-group separations. In addition to the silhouette score, the ablation study in Tab. 8 of the main paper also indicates that $\{1, 4, 8\}$ is a better trade-off between performance and training cost.

**Multi-Granularity MoE Restoration Model**  After training the degradation extractor, we construct the DR set for all training samples. Then, we hierarchically cluster the obtained DR set into multiple groups in different granularities, constructing multi-granularity degradation sets. In our full-model version, considering the task-agnostic capability, we generally adopt Restormer (Zamir et al., 2022) as the expert model across most tasks. However, we found Restormer cannot perform well in low-light enhancement. We thus observe clusters predominantly composed of low-light degradations and select RetinexFormer (Cai et al., 2023) as the expert model for those clusters. In our LoRA version, we adopt Restormer as the base model and generally train LoRA weight with rank of 8 as (Ai et al., 2024a). Referring to the total number of degradation types and degrees in training DR extractor, we adopt 3 levels and empirically set 1, 7, and 19 clusters for the 0-th level to the 2-th level in single-degradation all-in-one setting. For mixed-degradation setup, we generally adopt 1, 4, and 8 clusters, which performs well in both CDD-11 benchmark dataset and our introduced more complex scenarios, indicating a good generalization ability.

**Structure of $\mathcal{H}$ and $\mathcal{G}$**  Without dedicated design, we adopt 5-layer MLP networks for both $\mathcal{H}_d$ and $\mathcal{H}_g$, in which the number of input and output channels is 512 (the same as the dimension of DR $\mathbf{e}$). The structures of $\mathcal{G}_d$ and $\mathcal{G}_g$ are one-layer MLP networks with input channels of 512 and the number of output channels equal to the number of experts.

**Instruction Mode, Ours†**  The *instruction* mode of our method allows inference based on given task instructions, *i.e.*, the task name. In our implementation, the task instructions can convert to specific binary masks, which represent pruning or not pruning on the corresponding expert model in our MoE restoration model. According to the masks, we first prune out the corresponding expert models, then we conduct degradation- and granularity-based routing to find the most appropriate expert model for corrupted inputs. In this way, the error in degradation estimation is alleviated, the routing accuracy is highly improved, and thus the restoration performance is further improved.

**Training of Degradation Extractor**  To train our DR extractor, we adopt the ways in Sec. B to synthesize various types of low-quality data. According to the parameter that decides the degradation degree, we separate synthetic data into multiple groups. For deraining degradation, strength in range [0, 50) is small rain, [50, 100) is medium rain, and [100, 150) is large rain; In denoising, $\sigma$ of Gaussian noise in range [0, 15) is small noise, [15, 35) is medium noise, and [35, 50] is large noise; In dehazing, scattering coefficient $\beta$ larger than 0.1 is thick haze, $\beta$ smaller than 0.1 is normal haze; In deblurring, kernel size of motion blur kernel larger than 15 is large blur, kernel

size smaller than 15 is small blur; In desnowing, we use snow masks from Snow100K dataset and adopt the official separation. In compression artifacts removal, image quality in range $[0, 15)$ is large compression, $[15, 35)$ is medium compression, $[35, 50]$ is small compression; In lowlight enhancement, the darkening coefficient in range $[1, 1.5)$ is light corruption and $[1.5, 3)$ is heavy corruption. The CLIP image encoder typically pre-processes the input to resolution of $224 \times 224$, it has little effect on the accuracy of semantic awareness in high-level vision tasks and task-level degradation classification in original DA-CLIP. However, we found that this large-ratio distortion can lead to the loss of fine-grained degradation information, *e.g.*, noise can be heavily alleviated by downsampling in CLIP processor. To address this issue, we crop high-resolution clean images into $224 \times 224$ patches and add synthetic degradations with different types and degrees to construct the dataset with fine-grained degradation, which can well maintain fine-grained degradation information. And typically, we adopt center-cropped $224 \times 224$ images as input in inference.

**Training of Multi-Granularity MoE Restoration Model**  Most works in high-level vision and NLP tasks train experts and routers jointly. To well deal with the conflicts between restoring different degradations, MoE system in image restoration works or all-in-one image restoration works prefer separate training (Zhu et al., 2023; Park et al., 2023; Cao et al., 2024b; Tian et al., 2024; Chen et al., 2024; Ai et al., 2024a), *i.e.*, sequentially training experts and routers. Similarly to the mentioned works, our experts are explicitly determined to solve degradations in each degradation set, experts should be learned in advance and are not affected by the routers to guarantee the stability of the whole model. In the training of experts, we first conduct hierarchical clustering based on collected datasets, after that, all degraded training images are divided into multiple groups in different granularities. In each granularity, we record for each group the data samples that is allocated to it. Then, we use the recorded data to train expert restoration network for each group. After training of experts, the weights of experts are frozen, and we train routers based on degradation- and granularity-estimation. We further tried two manners to train our method: training experts and routers from scratch simultaneously, further finetuning after sequentially train experts and routers. In the former, we found that the trainnig is instable and the capacity of experts quickly collapses. In the latter, it costs additional training time and GPU memory while the improvement is marginal.

**Training Details**  For the training of experts, we generally use the AdamW optimizer with batch-size of 8 and patch-size of 128, we take the CosineAnnealing learning rate scheduler with the initial learning rate of $3e^{-4}$. In full-model version, experts are seperatly trained on each DR set; In LoRA version, we train model on the coarest DR set as the base model, based on it, we then train LoRA weights on other finer DR set with rank of 8 as (Ai et al., 2024a). After training experts, the weights of experts are frozen, and we train degradation and granularity estimation based routers. We adopt Adam optimizer with batch-size of 8 and patch-size of 256, we adopt a fixed learning rate of $1e^{-3}$, and we set $\alpha$ to 0.1 and $\beta$ to 0.01 in Eq. (11), respectively. All experiments are trained on 8 NVIDIA A6000 GPUs.

# D  MORE COMPARISON RESULTS

**Results on More Complex Mixed-degradation**  In Tab. C and Tab. D, we further evaluate the performance of all-in-one image restoration in mixed-degradation setup. The evaluation results show that our method outperforms previous task-agnostic methods and all-in-one methods from simple degradations to complex degradations, and it also generalizes well in *out-dist.* degradations.

**More Comparisons on Real-world Datasets and Unseen Degradations**  More comparisons on real-world denoising, real-world motion-deblurring, and ISP artifacts removal can be seen in Table E. It shows that our method demonstrates better generalization ability compared to existing all-in-one image restoration methods.

**Comparisons to Scaling-up SOTA Methods**  According to the degradation- and granularity- estimation, the routers select the most appropriate expert for each input low-quality image, leading to a sparse inference strategy (see Tab. 6). Though our method has comparable inference #parameters, FLOPs, and latency to methods with similar backbones, one may know the effect of training parameters. To mitigate the gap of training #parameters, we scale up the training #parameters of Restormer (Zamir et al., 2022) (by scaling up the width to 128), PromptIR (Potlapalli et al., 2023) (by scaling up the width to 128), and MoCE-IR (Zamfir et al., 2025) (by scaling up the width to 128 and increase the number of experts tp 7) The experiment conducts on all-in-one image restoration

Table C: **All-in-One image restoration (mixed-degradation) results**. PSNR (dB, ↑) and SSIM (↑) are reported in RGB color spaces. The **best** performance is highlighted. The evaluation is on DIV2K-valid with *in-dist.* and *out-dist.* degradation parameters. '*n*T' means the average performance of *n* tasks (3T: H-R, H-N, H-N-R; 5T: H-R, H-N, H-N-R, LL-H-N-R, LL-H-N-B-R; 7T: all).

| Methods | H-R | | H-N | | H-N-R | | LL-H-N-R | | LL-H-N-S | | LL-H-N-B-R | | LL-H-N-B-R | | 3T | 5T | 7T |
|---|---|---|---|---|---|---|---|---|---|---|---|---|---|---|---|---|---|
| *In-distribution* | | | | | | | | | | | | | | | | | |
| MPRNet (Zamir et al., 2021) | 17.22 | .691 | 19.02 | .717 | 17.18 | .639 | 17.25 | .549 | 17.18 | .583 | 16.99 | .466 | 16.80 | .506 | 17.80 | 17.57 | 17.37 |
| SwinIR (Liang et al., 2021) | 16.35 | .620 | 18.74 | .693 | 16.48 | .585 | 16.78 | .513 | 16.73 | .555 | 16.54 | .434 | 16.10 | .482 | 17.19 | 17.01 | 16.81 |
| NAFNet (Chen et al., 2022) | 19.14 | .775 | 22.71 | .843 | 18.56 | .723 | 17.41 | .603 | 17.09 | .631 | 16.37 | .505 | 15.83 | .530 | 20.13 | 18.98 | 18.15 |
| Restormer (Zamir et al., 2022) | 19.85 | .795 | 20.44 | .820 | 17.64 | .719 | 16.86 | .588 | 16.56 | .623 | 15.92 | .483 | 15.47 | .520 | 19.31 | 18.27 | 17.53 |
| AirNet (Li et al., 2022a) | 18.67 | .752 | 23.34 | .855 | 19.03 | .726 | – | – | – | – | – | – | – | – | 20.34 | – | – |
| PromptIR (Valanarasu et al., 2022) | 22.20 | .825 | 24.23 | **.886** | 20.86 | .772 | – | – | – | – | – | – | – | – | 22.43 | – | – |
| MiOIR (Xiangtao et al., 2024) | 24.58 | .892 | 24.46 | .879 | 22.42 | .818 | 19.46 | .661 | – | – | 18.12 | .561 | – | – | 23.82 | 21.80 | – |
| InstructIR (Conde et al., 2024) | 21.57 | .827 | 23.80 | .838 | 21.14 | .775 | 19.21 | .666 | – | – | 18.06 | .564 | – | – | 22.17 | 20.75 | – |
| OneRestore (Guo et al., 2024) | 17.35 | .711 | 18.70 | .669 | 16.41 | .617 | 16.66 | .621 | 20.24 | .769 | – | – | – | – | 17.48 | – | – |
| UniRestorer (Ours) | **29.63** | **.950** | 26.93 | .880 | 26.33 | .875 | 24.29 | .830 | 22.71 | .790 | 21.87 | .735 | 20.10 | .729 | 27.63 | 25.81 | 24.55 |
| UniRestorer-LoRA (Ours) | 29.28 | .944 | 26.61 | .873 | 26.03 | .869 | 24.01 | .824 | 22.45 | .784 | 21.63 | .729 | 19.89 | .724 | 27.30 | 25.67 | 24.28 |
| *Out-of-distribution* | | | | | | | | | | | | | | | | | |
| MPRNet (Zamir et al., 2021) | 15.71 | .636 | 16.38 | .598 | 15.57 | .560 | 14.84 | .345 | 14.59 | .355 | 14.58 | .286 | 14.33 | .299 | 15.88 | 15.41 | 15.14 |
| SwinIR (Liang et al., 2021) | 15.10 | .556 | 15.79 | .553 | 14.50 | .486 | 14.72 | .341 | 14.32 | .353 | 14.29 | .282 | 13.95 | .298 | 15.13 | 14.88 | 14.66 |
| NAFNet (Chen et al., 2022) | 16.77 | .713 | 19.68 | .751 | 15.11 | .595 | 14.40 | .385 | 14.05 | .390 | 13.89 | .317 | 13.27 | .324 | 17.18 | 16.00 | 15.31 |
| Restormer (Zamir et al., 2022) | 17.44 | .723 | 17.31 | .738 | 17.44 | .723 | 14.11 | .378 | 13.59 | .388 | 13.60 | .314 | 12.83 | .318 | 17.39 | 15.97 | 15.18 |
| AirNet (Li et al., 2022a) | 17.50 | .732 | 20.16 | .777 | 15.07 | .380 | – | – | – | – | – | – | – | – | 17.57 | – | – |
| PromptIR (Valanarasu et al., 2022) | 19.84 | .797 | 20.57 | .795 | 18.22 | .672 | – | – | – | – | – | – | – | – | 19.54 | – | – |
| MiOIR (Xiangtao et al., 2024) | 20.85 | .850 | 22.39 | .821 | 18.87 | .700 | 15.27 | .406 | – | – | 14.70 | .332 | – | – | 20.70 | 18.41 | – |
| InstructIR (Conde et al., 2024) | 20.33 | .825 | 19.02 | .636 | 17.74 | .597 | 15.34 | .386 | – | – | 14.52 | .296 | – | – | 19.03 | 17.39 | – |
| OneRestore (Guo et al., 2024) | 17.81 | .732 | 16.77 | .580 | 15.10 | .595 | 17.49 | .566 | **21.04** | **.682** | – | – | – | – | 16.56 | – | – |
| UniRestorer (Ours) | **27.32** | **.937** | 23.82 | **.811** | 23.42 | **.801** | 20.21 | **.661** | 20.15 | .631 | **17.49** | **.529** | 18.03 | **.543** | 24.85 | 22.45 | 21.49 |
| UniRestorer-LoRA (Ours) | 26.96 | .929 | 23.47 | .801 | 23.09 | .792 | 19.92 | .651 | 19.86 | .622 | 17.24 | .522 | 17.74 | .535 | 24.51 | 22.66 | 21.18 |

Table D: **All-in-One image restoration (mixed-degradation) results**. PSNR (dB, ↑) and SSIM (↑) are reported in RGB color spaces. The **best** performance is highlighted. The evaluation is on Urban100 with *in-dist.* and *out-dist.* degradation parameters. '*n*T' means the average performance of *n* tasks (3T: H-R, H-N, H-N-R; 5T: H-R, H-N, H-N-R, LL-H-N-R, LL-H-N-B-R; 7T: all).

| Methods | H-R | | H-N | | H-N-R | | LL-H-N-R | | LL-H-N-S | | LL-H-N-B-R | | LL-H-N-B-R | | 3T | 5T | 7T |
|---|---|---|---|---|---|---|---|---|---|---|---|---|---|---|---|---|---|
| *In-distribution* | | | | | | | | | | | | | | | | | |
| MPRNet (Zamir et al., 2021) | 17.30 | .784 | 19.78 | .812 | 18.55 | .749 | 17.00 | .589 | 15.02 | .556 | 16.17 | .469 | 16.64 | .580 | 18.54 | 17.53 | 17.20 |
| SwinIR (Liang et al., 2021) | 16.25 | .643 | 20.02 | .790 | 17.52 | .612 | 16.47 | .543 | 16.68 | .570 | 15.97 | .481 | 14.42 | .379 | 17.93 | 17.38 | 16.76 |
| NAFNet (Chen et al., 2022) | 19.79 | .838 | 20.08 | .805 | 18.08 | .777 | 17.04 | .569 | 17.18 | .648 | 14.87 | .457 | 15.97 | .409 | 19.31 | 18.43 | 17.57 |
| Restormer (Zamir et al., 2022) | 19.40 | .807 | 22.82 | .846 | 18.06 | .727 | 16.41 | .604 | 16.67 | .636 | 14.62 | .443 | 14.14 | .367 | 20.09 | 18.67 | 17.44 |
| AirNet (Li et al., 2022a) | 17.93 | .769 | 20.85 | .821 | 15.40 | .676 | – | – | – | – | – | – | – | – | 18.06 | – | – |
| PromptIR (Valanarasu et al., 2022) | 21.46 | .860 | 27.3 | .916 | 19.53 | .780 | – | – | – | – | – | – | – | – | 22.76 | – | – |
| MiOIR (Xiangtao et al., 2024) | **28.98** | **.945** | 27.34 | .906 | 23.68 | .840 | 19.43 | .711 | – | – | 16.49 | .525 | – | – | 26.66 | – | – |
| OneRestore (Guo et al., 2024) | 17.58 | .771 | 17.65 | .764 | 18.39 | .712 | 18.39 | .746 | 19.73 | .724 | – | – | – | – | 17.87 | 18.34 | – |
| MoCE-IR (Zamfir et al., 2025) | 19.27 | .824 | 17.81 | .819 | 14.63 | .713 | 22.58 | .848 | 14.88 | .547 | – | – | – | – | 17.23 | 17.83 | – |
| UniRestorer (Ours) | 27.68 | .943 | **28.66** | **.920** | 26.40 | **.896** | 25.54 | **.844** | 22.31 | **.800** | 20.61 | **.698** | 18.28 | **.607** | 27.58 | 26.11 | 24.21 |
| UniRestorer-LoRA (Ours) | 27.33 | .937 | 28.34 | .908 | 26.10 | .890 | 25.26 | .838 | 22.05 | .794 | 20.37 | .692 | 18.07 | .602 | 27.26 | 25.82 | 23.93 |
| *Out-of-distribution* | | | | | | | | | | | | | | | | | |
| MPRNet (Zamir et al., 2021) | 15.80 | .639 | 14.13 | .630 | 15.92 | .654 | 14.88 | .369 | 11.65 | .266 | 13.75 | .289 | 13.92 | .248 | 15.28 | 14.47 | 14.29 |
| SwinIR (Liang et al., 2021) | 15.63 | .610 | 13.96 | .581 | 13.58 | .455 | 13.86 | .377 | 12.63 | .272 | 12.81 | .217 | 12.20 | .264 | 14.39 | 13.93 | 13.52 |
| NAFNet (Chen et al., 2022) | 15.93 | .630 | 15.86 | .772 | 15.36 | .608 | 14.07 | .305 | 12.90 | .340 | 13.47 | .201 | 13.25 | .281 | 15.71 | 14.28 | 14.40 |
| Restormer (Zamir et al., 2022) | 16.45 | .716 | 14.64 | .682 | 16.92 | .681 | 14.88 | .364 | 12.90 | .313 | 12.92 | .222 | 12.59 | .233 | 16.00 | 15.15 | 14.47 |
| AirNet (Li et al., 2022a) | 16.60 | .735 | 17.56 | .717 | 13.75 | .572 | – | – | – | – | – | – | – | – | 15.97 | – | – |
| PromptIR (Valanarasu et al., 2022) | 20.93 | .852 | **21.85** | **.812** | 16.52 | .647 | – | – | – | – | – | – | – | – | 19.76 | – | – |
| MiOIR (Xiangtao et al., 2024) | **21.75** | **.882** | 19.82 | .736 | 19.76 | .712 | 14.12 | .453 | – | – | 12.56 | .287 | – | – | **20.44** | – | – |
| OneRestore (Guo et al., 2024) | 16.58 | .720 | 16.76 | .741 | 15.46 | .652 | 16.01 | .679 | 18.03 | .612 | – | – | – | – | 16.26 | 16.56 | – |
| MoCE-IR (Zamfir et al., 2025) | 18.72 | .826 | 14.76 | .705 | 11.32 | .585 | 17.93 | .592 | 19.01 | .641 | – | – | – | – | 14.93 | 16.34 | – |
| UniRestorer (Ours) | 20.65 | .856 | 20.10 | .763 | **20.48** | **.755** | 19.42 | **.585** | 18.43 | **.542** | 16.97 | **.419** | 17.09 | **.490** | 20.41 | **19.81** | **19.02** |
| UniRestorer-LoRA (Ours) | 20.29 | .848 | 19.75 | .753 | 20.15 | .746 | 19.13 | .575 | 18.14 | .533 | 16.72 | .412 | 16.80 | .482 | 20.06 | 19.49 | 18.71 |

single-degradation setups. As shown in Tab. F, even after scaling up training parameters of Restormer, PromptIR, and MoCE-IR same to ours, our method still achieves better performance across all tasks, it reveals that simply increasing the model size is not the essence of improving the performance, which is different from the scaling law in high-level vision and NLP tasks. From this perspective, our method provides an effective paradigm for scaling up low-level vision image restoration models, enabling both improved performance and efficient computation.

Table E: Comparisons to SOTA methods on more real-world degradations. PSNR (dB, ↑) is reported in RGB color spaces.

| Methods | SIDD | DND | ISP artifact | RealBlur-J | RealBlur-R |
|---|---|---|---|---|---|
| PromptIR (Potlapalli et al., 2023) | 24.32 | 25.19 | 12.81 | 25.48 | 33.34 |
| MiOIR (Xiangtao et al., 2024) | 22.87 | 23.67 | 13.79 | 24.51 | 16.79 |
| MoCE-IR-3T (Zamfir et al., 2025) | 23.91 | 24.04 | 12.97 | 25.60 | 31.39 |
| MoCE-IR-5T (Zamfir et al., 2025) | 15.90 | 21.16 | 12.88 | 14.30 | 12.20 |
| Ours | 25.10 | 26.47 | 16.26 | 26.10 | 34.06 |

Table F: Comparisons to SOTA methods by vanilla scaling up. PSNR (dB, ↑) and SSIM (↑) are reported in RGB color spaces. The **best** performance is highlighted.

| Methods | Params | Derain Rain100L | | Dehaze SOTS | | Denoise BSD68 | | Deblur GoPro | | Lowlight LOLv1 | | Desnow Snow100K | | CAR LIVE1 | | Average |
|---|---|---|---|---|---|---|---|---|---|---|---|---|---|---|---|---|
| Restormer (Zamir et al., 2022) | 183M | 34.46 | .981 | 27.83 | .958 | 31.10 | .936 | 27.78 | .891 | 22.03 | .788 | 26.98 | .923 | 26.88 | .865 | 28.15 |
| PromptIR (Potlapalli et al., 2023) | 202M | 34.94 | .984 | 27.59 | .952 | 31.18 | .937 | 27.77 | .888 | 21.44 | .810 | 27.08 | .923 | 26.85 | .863 | 28.12 |
| MoCE-IR (Zamfir et al., 2025) | 200M | 37.70 | .985 | 29.93 | .962 | 31.30 | .929 | 29.61 | .914 | 22.42 | .790 | 28.58 | .939 | 27.07 | .837 | 29.51 |
| UniRestorer-LoRA (Ours) | 171M | 40.22 | .991 | 34.94 | .971 | 31.39 | .937 | 30.86 | .933 | 24.42 | .827 | 30.53 | .925 | 30.31 | .922 | 31.81 |
| UniRestorer (Ours) | 627M | **41.68** | **.996** | **36.44** | **.984** | **31.43** | .936 | **31.48** | **.948** | **26.67** | **.863** | **31.12** | **.940** | **30.57** | **.926** | **32.77** |

Table G: Effect of DR extractors.

| Methods | Rain 200H | Rain 200L | Test 1200 | Test 2800 |
|---|---|---|---|---|
| VGG | 31.47 | 40.59 | 35.26 | 34.18 |
| DDR | 30.97 | 40.37 | 35.25 | 34.17 |
| DA-CLIP | 31.65 | 40.67 | 35.24 | 34.12 |
| Manual | 31.88 | 41.43 | 35.32 | 34.23 |
| Ours | **32.01** | **41.61** | **35.50** | **34.40** |

Table H: Effect of loss functions.

| $\ell_1$ | $\mathcal{L}_{dg}$ | $\mathcal{L}_{load}$ | MiO (Average) In-dist. | Out-dist. |
|---|---|---|---|---|
| ✓ | | | 24.14 | 18.37 |
| ✓ | ✓ | | 24.35 | 18.76 |
| ✓ | ✓ | ✓ | **24.46** | **19.45** |

Table I: Effect of high-resolution inputs on DR. Accuracy of degradation classification is reported. 'S', 'M', and 'L' mean small, medium, large degree of degradations.

| Methods | Derain S | M | L | Dehaze S | L | Denoise S | M | L | Deblur S | L | Lowlight S | L | Desnow S | M | L | CAR S | M | L | Average |
|---|---|---|---|---|---|---|---|---|---|---|---|---|---|---|---|---|---|---|---|
| Resize | 0.84 | 0.73 | 0.86 | 0.73 | 0.75 | 1.00 | 0.12 | 0.30 | 0.81 | 0.74 | 0.92 | 0.94 | 0.70 | 0.73 | 0.66 | 0.86 | 0.36 | 0.58 | 0.72 |
| Center-crop | 0.96 | 0.92 | 1.00 | 0.89 | 0.93 | 0.93 | 0.92 | 0.96 | 0.93 | 0.87 | 0.96 | 0.93 | 0.85 | 0.80 | 0.83 | 0.83 | 0.90 | 1.00 | 0.91 |
| Full-resolution | 0.93 | 0.93 | 0.96 | 0.94 | 0.93 | 0.96 | 0.91 | 0.93 | 0.95 | 0.90 | 0.91 | 0.96 | 0.92 | 0.86 | 0.93 | 0.89 | 0.96 | 1.00 | 0.93 |

Table J: Effect of high-resolution inputs on DR. PSNR is reported in RGB color spaces.

| Methods | Derain | Dehaze | Denoise | Deblur | Lowlight | Desnow | CAR | Average |
|---|---|---|---|---|---|---|---|---|
| Resize | 22.84 | 25.15 | 29.37 | 34.59 | 24.11 | 29.92 | 26.57 | 27.50 |
| Center-crop | 23.60 | 25.37 | 30.28 | 35.30 | 24.61 | 30.08 | 27.75 | 28.14 |
| Full-resolution | 23.55 | 25.43 | 30.28 | 35.48 | 24.67 | 30.19 | 27.76 | 28.19 |

## E MORE ABLATION STIDUES

**Effect of Degradation Extractor** As shown in Tab. G, we adopt pre-trained vgg16 (Simonyan & Zisserman, 2015), pre-trained ESRGAN (Liu et al., 2022b) on bicubic 4× super-resolution, and pre-trained DACLIP (Luo et al., 2024) as DR extractors. We construct multi-granularity degradation sets by leveraging DRs extracted from different DR extractors. Then we re-train expert networks based on each obtained degradation set. The experiments are conducted on deraining task. Since our degradation extractor is aware of finer-grained DRs, the clustered training data can be more consistent in DR, thus our method can achieve better performance.

**Effect of Loss Functions** As shown in Tab. H, both $\mathcal{L}_{dg}$ and $\mathcal{L}_{load}$ can benefit more accurate routing. In particular, $\mathcal{L}_{dg}$ and $\mathcal{L}_{load}$ can avoid the routing to collapse to the finest-grained level, improving performance on *out-dist.*

**Effect of Resolution of Input Images** As shown in Tab. I and Tab. J, we further compare two other manners for degradation extraction from low-quality images. (1) We resize the full image to $224 \times 224$ and feed the resized image into DR extractor, named 'Resize'. (2) We directly feed the full-resolution image into DR extractor, named 'Full'.

We can draw several observations. (1) 'Resize' can negatively affect degradation representation, since it destroys the degradation contents. For example, the noise level would be suppressed after resizing an image with large noise. This is why the classification accuracy in large noise is only $30\%$. (2) 'Full' is indeed better than our cropping strategy in restoration performance and classification accuracy of spatially variant degradation (e.g., snow). But the overall average improvement is insignificant, within 0.05dB PSNR and 2% accuracy. (3) Our strategy has less inference time than 'Full'. For example, when testing a $3840 \times 2160$ image using DR extractor, 'Full' costs 1740ms while we only require 30ms. In this case, we achieve 58 times faster.

In summary, when the low-quality input is very large, our strategy achieves comparable overall results with full-image inference in both restoration performance and classification accuracy, while offering faster inference speed. It shows our strategy can alleviate the influence of the high-resolution, and has a small overall effect on the accuracy of the DR extractor.

Table K: Silhouette score of Spectral clustering in different {# DRs}. Higher scores indicate better clustering.

| {# DRs } | Silhouette Score | | |
| --- | --- | --- | --- |
| | 1st-level | 2nd-level | 3rd-level |
| {1, 2} | 0.3824 | – | – |
| {1, 4} | 0.4630 | – | – |
| {1, 8} | 0.5895 | – | – |
| {1, 16} | 0.2779 | – | – |
| {1, 4, 8} | 0.4630 | 0.6668 | – |
| {1, 4, 8, 16} | 0.4630 | 0.6668 | 0.4889 |

**Effect of Alternative Clustering Method** As shown in Table K and Table L, we conduct additional experiments with a variant of our proposed model, denoted as $Ours_{spe}$, where the K-Means clustering is replaced with spectral clustering. We report both clustering quality (*i.e.*, silhouette score) and restoration performance (*i.e.*, PSNR) of the $Ours_{spe}$, as shown in Table L and Table K, respectively. For quick evaluation, the trained model only includes results for two hierarchical layers, *i.e.*, {#DRs}={1, 4}. The results demonstrate that $Ours_{spe}$ achieves similar performance to $Ours_{kms}$ in both the silhouette score and PSNR. This suggests our model is robust to alternative clustering methods.

**Effect of Top-$k$ Router** In our experiments, we adopt the top-1 routing strategy for the following reasons: (1) Our model with top-1 routing strategy achieves inference costs comparable to other methods. (2) Using additional routers, such as the top-2 routing strategy, would double the inference cost. (3) This sparse activation strategy has proven effective in current LLM/MLLM. For example, DeepSeek-R1 (Guo et al., 2025) has a total of 671B parameters, but only 37B parameters ($\sim 5.5\%$) are activated in a single inference, significantly reducing computational costs. We believe this strategy is also reasonable and acceptable in image restoration tasks.

**Discussion on Adaptive Clustering** We conduct experiments on adaptive clustering. Similar to GRIDS (Cao et al., 2024b), we offer a candidate list of [2, 3, 4, 5, 6] for cluster count $k$ and use the binary search strategy to search for the best $k$ in splitting each cluster. During splitting, we adopt the silhouette score as the metric to decide whether to split and to evaluate if the selected $k$ is appropriate for the current split. Additionally, we ensure that the minimum number of samples in each cluster is no lower than 200. This process results in a three-level tree with hyperparameter $[1, 6, 13]$. To evaluate the effectiveness of the results of adaptive clustering, we train our method in the newly formed degradation space and evaluate the performance on CDD-11 dataset. The results in Table M show that adaptive clustering indeed further improves the performance of our method.

**Discussion on Hybrid Usage** When users provide a perceived degradation type and degree of an image, they can also provide a confidence level (*e.g.*, 0, 1, 2, higher values indicate higher confidence) for the current degradation, which can serve as a granularity routing cue. We randomly synthesize

Table L: Performance comparison of spectral clustering on CDD-11 dataset. PSNR (dB, ↑) is reported in RGB color spaces.

| Methods | L | H | R | S | L-H | L-R | L-S | H-R | H-S | L-H-R | L-H-S | Avg. |
|---|---|---|---|---|---|---|---|---|---|---|---|---|
| $\text{Ours}_{spe}$ | 26.91 | 27.41 | 32.77 | 32.84 | 25.62 | 25.86 | 24.79 | 26.25 | 24.19 | 23.87 | 23.77 | 26.75 |
| $\text{Ours}_{kms}$ | 27.10 | 27.36 | 33.33 | 32.55 | 25.53 | 25.74 | 25.16 | 26.42 | 24.21 | 24.03 | 24.17 | 26.87 |

Table M: Performance (PSNR) comparison of adaptive clustering on CDD-11 dataset, PSNR is reported.

| Methods | L | H | R | S | L-H | L-R | L-S | H-R | H-S | L-H-R | L-H-S | Avg. |
|---|---|---|---|---|---|---|---|---|---|---|---|---|
| $\text{Ours}_{adaptive}$ | 27.76 | 34.71 | 35.76 | 37.35 | 27.32 | 27.05 | 26.90 | 31.86 | 31.02 | 26.14 | 26.38 | 30.20 |
| Ours | 27.71 | 34.53 | 36.07 | 36.84 | 26.88 | 26.93 | 26.73 | 32.02 | 30.82 | 25.78 | 25.87 | 30.02 |

Table N: Comparisons of different hybrid usage. PSNR (dB, ↑) SSIM (↑) are reported in RGB color spaces.

| Method | 'Deg.' | 'Deg. & Gran.' | 'Single-task' |
|---|---|---|---|
| Deraining | 32.18/.973 | 32.42/.975 | 32.62/.980 |

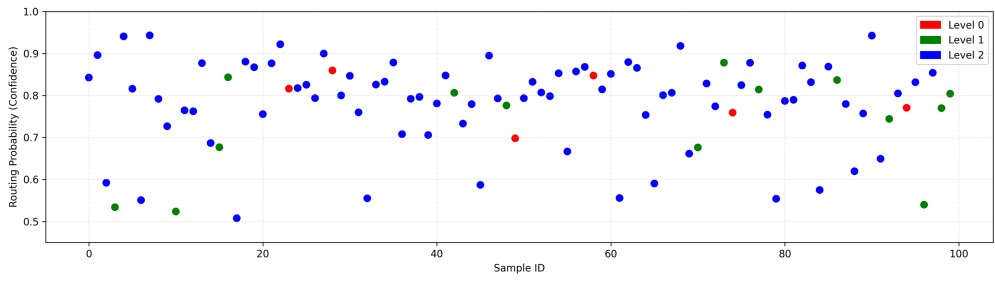

(a) Routing confidence of samples in urban100 with in-dist. degradation parameters

(b) Routing confidence of samples in urban100 with out-dist. degradation parameters

Figure C: Visualization of routing results and routing confidence on granularity. The samples are from Urban100 dataset with mixed degradations.

rain streaks of varying intensities on clean images from the Urban100 dataset. We then evaluate the performance when both degradation type, degree, and confidence are provided (referred to as 'Deg. & Gran.'). Specifically, we compare the 'Deg. & Gran.' with two other conditions: 'Deg.' (which only provides degradation type, as described in our main paper) and 'single-task' (Restormer trained on deraining task). Results in Table N show that it has potential in further improving the performance and closing the gap with single-task models when granularity cues are provided.

Table O: Effect of errors in granularity estimation. The PSNR and the number of samples allocated to each granularity are reported.

| Methods | In-dist. | | | | Out-dist. | | | |
|---|---|---|---|---|---|---|---|---|
| | 0st-level | 1nd-level | 2rd-level | PSNR | 0st-level | 1nd-level | 2rd-level | PSNR |
| Ours (actual) | 5 | 17 | 78 | 24.46 | 34 | 50 | 16 | 19.45 |
| Ours (GT) | 7 | 13 | 80 | 24.53 | 29 | 46 | 25 | 19.66 |

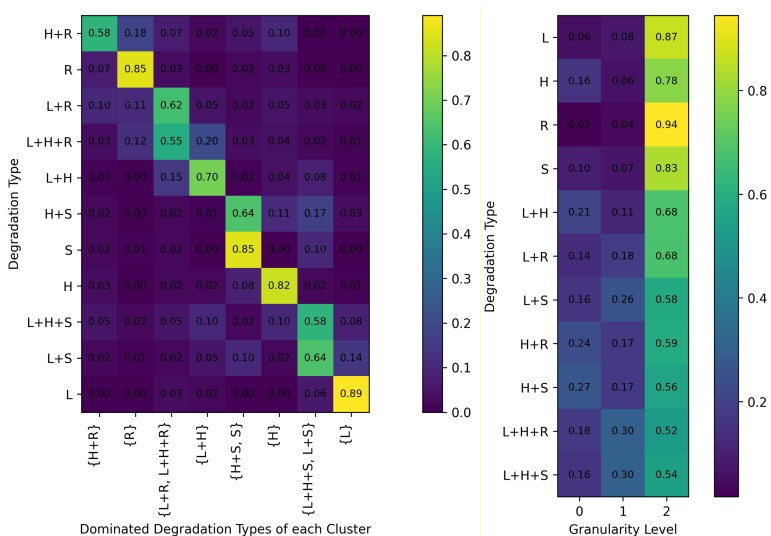

Figure D: Visualization of routing frequency of degradation estimation-based routing and granularity estimation-based routing. In the left figure, the horizontal axis represents the dominated degradation set of each finest-grained leaf node, and the vertical axis represents the set of samples with different degradation types in the CDD11 test dataset. In the right figure, the horizontal axis represents the granularity level, and the vertical axis represents the set of samples with different degradation types in the CDD11 test dataset.

## F  ANALYSIS ON ROUTING

**Effect of Errors in Granularity Estimation**  We first define the 'GT granularity' as the granularity level at which the model achieves the best restoration performance, compared to all other granularity levels. To assess the effect of routing noise and errors, we compare the restoration performance of our model using the 'GT granularity' and the estimated granularity. Table O illustrate the restoration performance on *in-dist.* and *out-dist.*, respectively. The restoration performance gap between the estimated granularity and 'GT granularity' is minimal. This suggests that granularity estimation has only minor errors, making a slight impact on restoration performance. The corresponding routing confidence of *in-dist.* and *out-dist.* can be seen in Fig. C.

## G  LIMITATIONS AND FUTURE WORKS

On the one hand, the training data mainly comes from publicly released datasets, as required for fair comparison with baseline methods. However, the scale of training data is essential, particularly in developing large-scale models. In image restoration works, the scale of training data is often limited, *e.g.* Rain100L (Yang et al., 2017) contains only 200 pairs of training data. Previous works (Wang et al., 2021) have exploited the benefits brought by scaling up the training data, we thus plan to collect more high-quality training data to bring further improvement to our method. On the other hand, we plan to expand and align our degradation space with real-world scenarios, aiming to restore real-world image degradations.

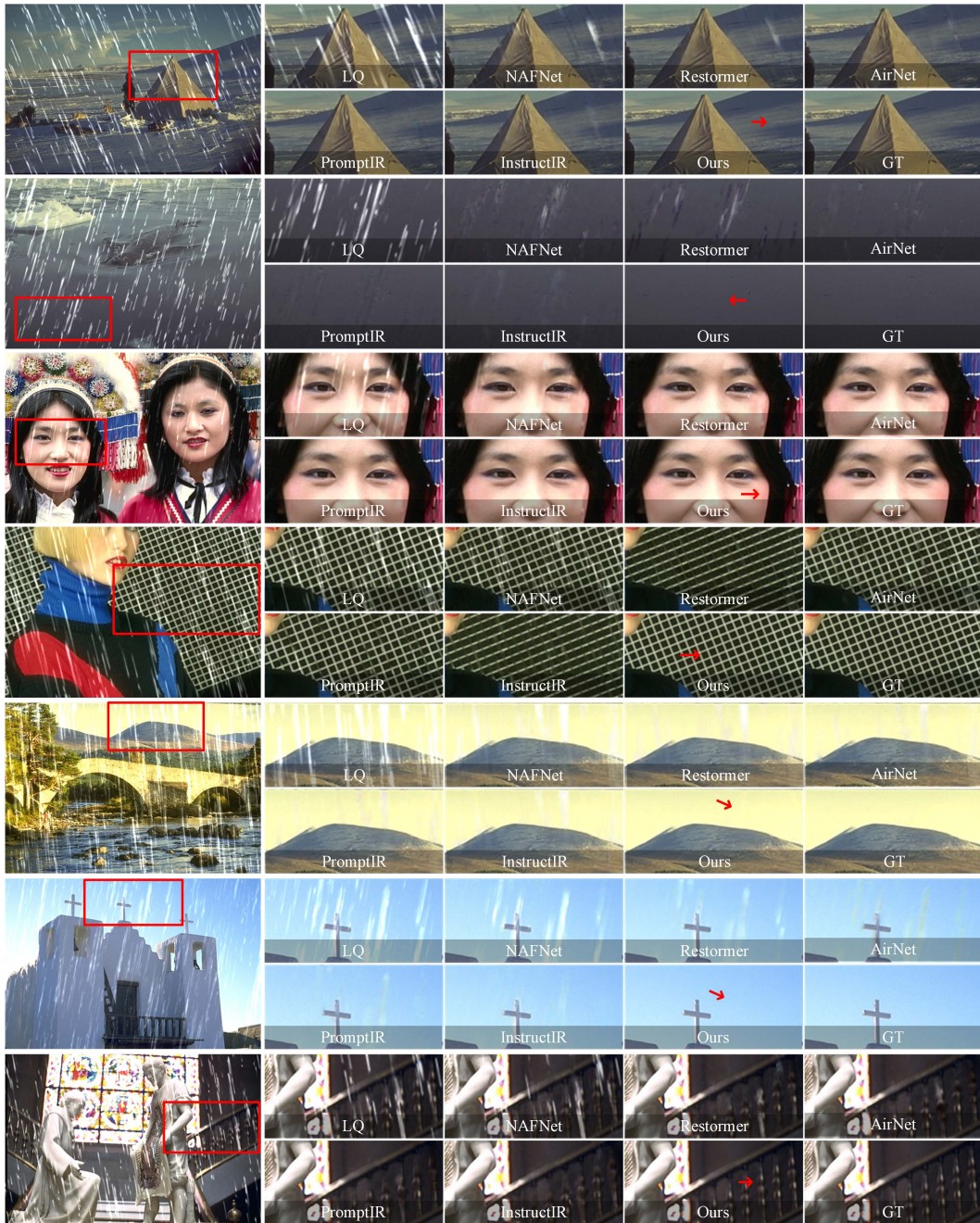

Figure E: Visual comparison of results on All-in-One image restoration (deraining).

## H    MORE RESULTS AND VISUAL COMPARISONS

We show more visual comparison results in this section. Compared to task-agnostic methods, all-in-one methods, and specific single-task models, our method can restore more clear results with detailed textures. The single-degradation results of all-in-one image restoration comparisons can be seen in Fig. E to Fig. K. The comparison results with specific single-task models can be seen in Fig. L to Fig. Q. In the evaluation of mixed degradation, since the resolution of images from DIV2K-valid is too high, limiting inference on the whole image for all methods. We first crop high-resolution images into overlapped $512 \times 512$ crops, after inference, we inverse the cropping process and recompose crops back to one whole image, though averaging the overlapped area, the border artifacts cannot be avoidable. Comparison result of *in-dist.* and *out-dist.* can be seen in Fig. R and Fig. S, respectively.

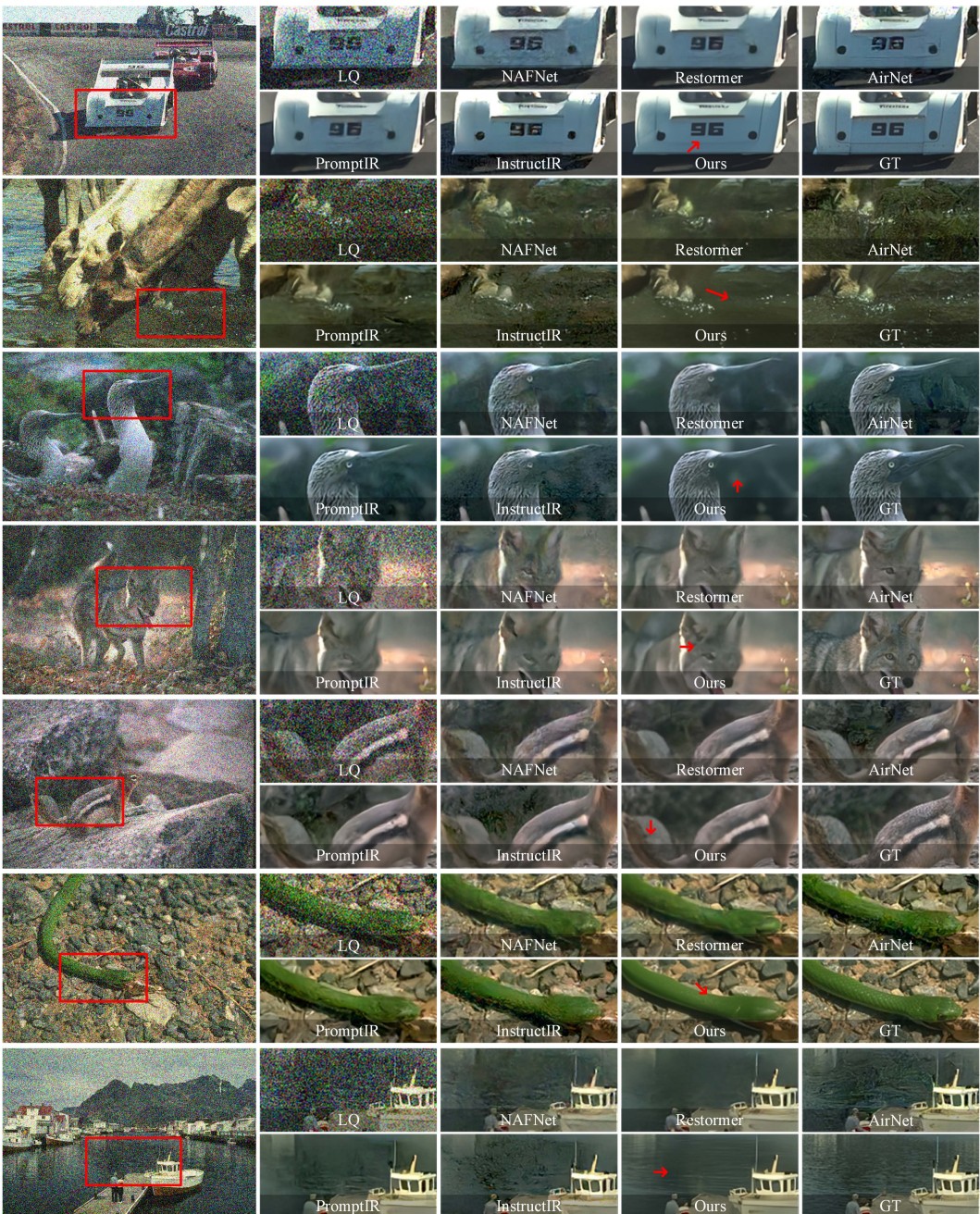

Figure F: Visual comparison of results on All-in-One image restoration (Gaussian-denoising).

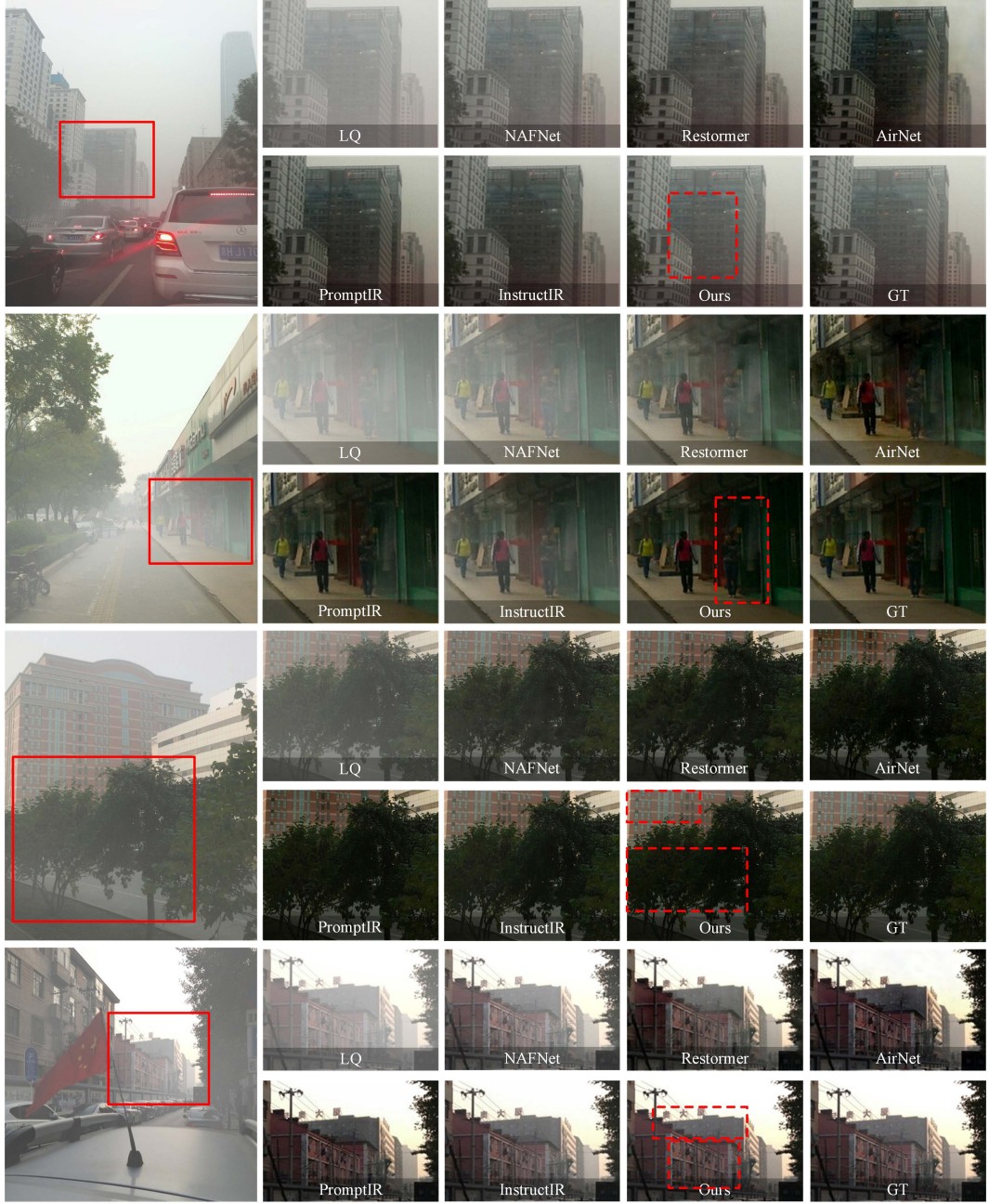

Figure G: Visual comparison of results on All-in-One image restoration (dehazing).

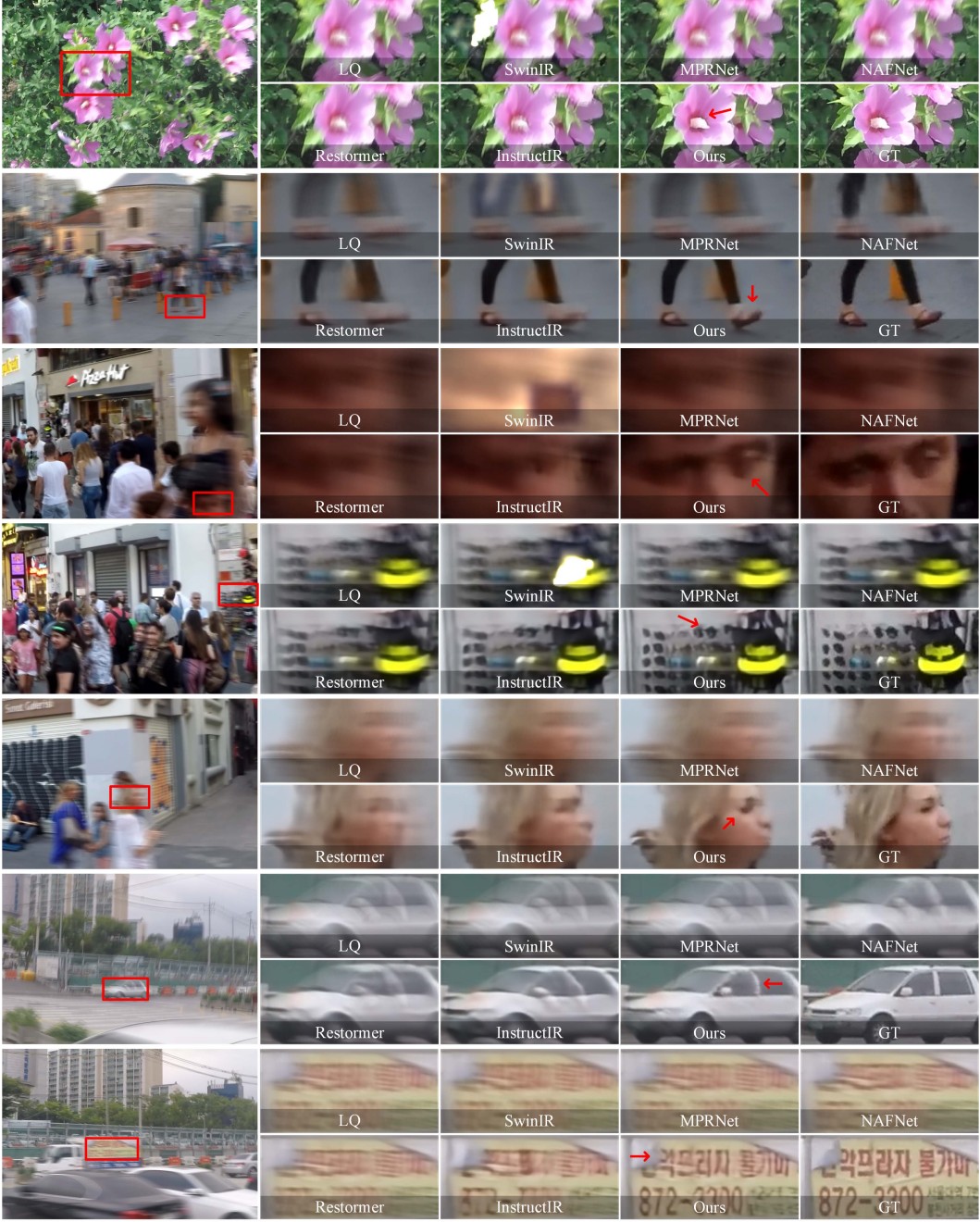

Figure H: Visual comparison of results on All-in-One image restoration (motion-deblurring).

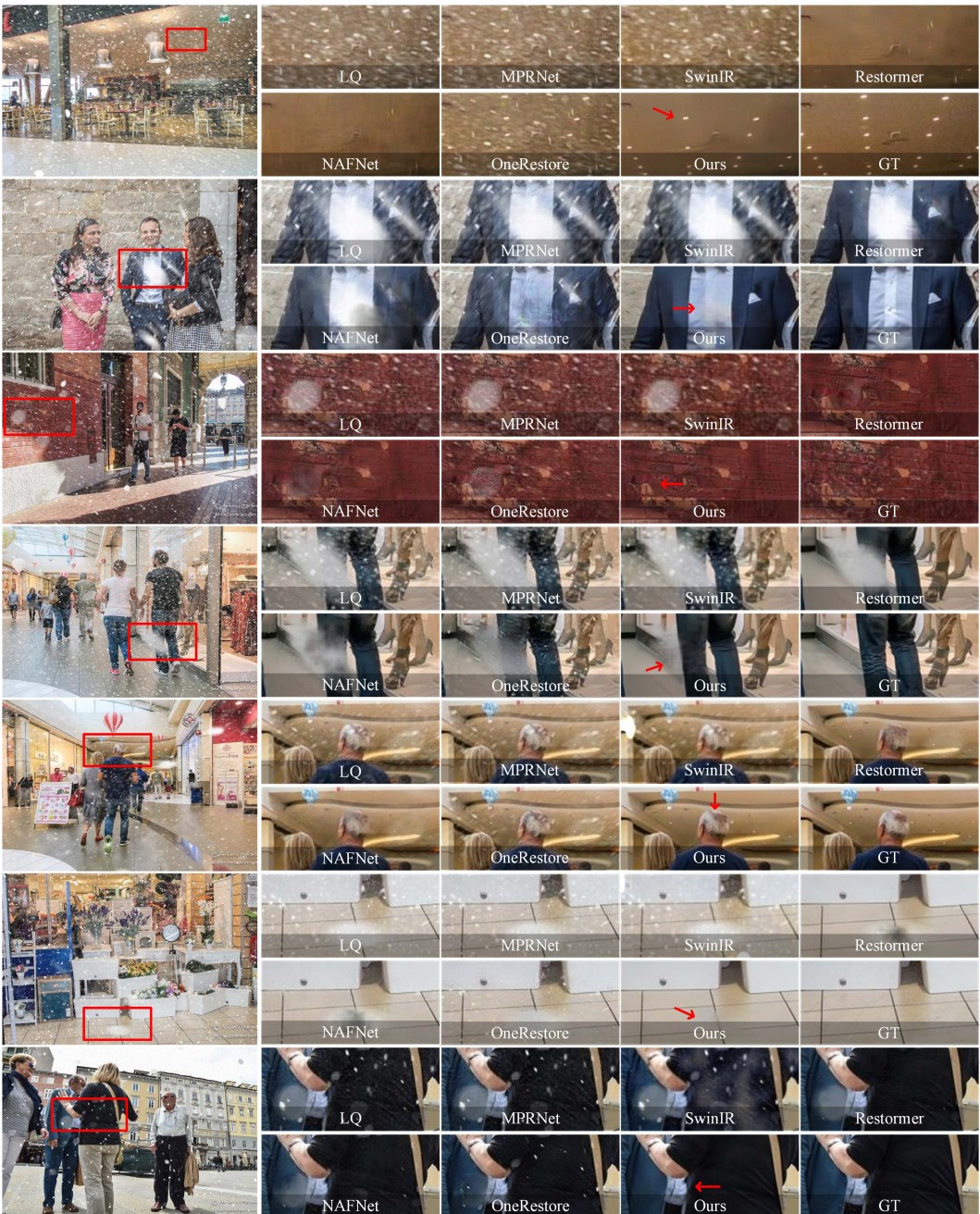

Figure I: Visual comparison of results on All-in-One image restoration (desnowing).

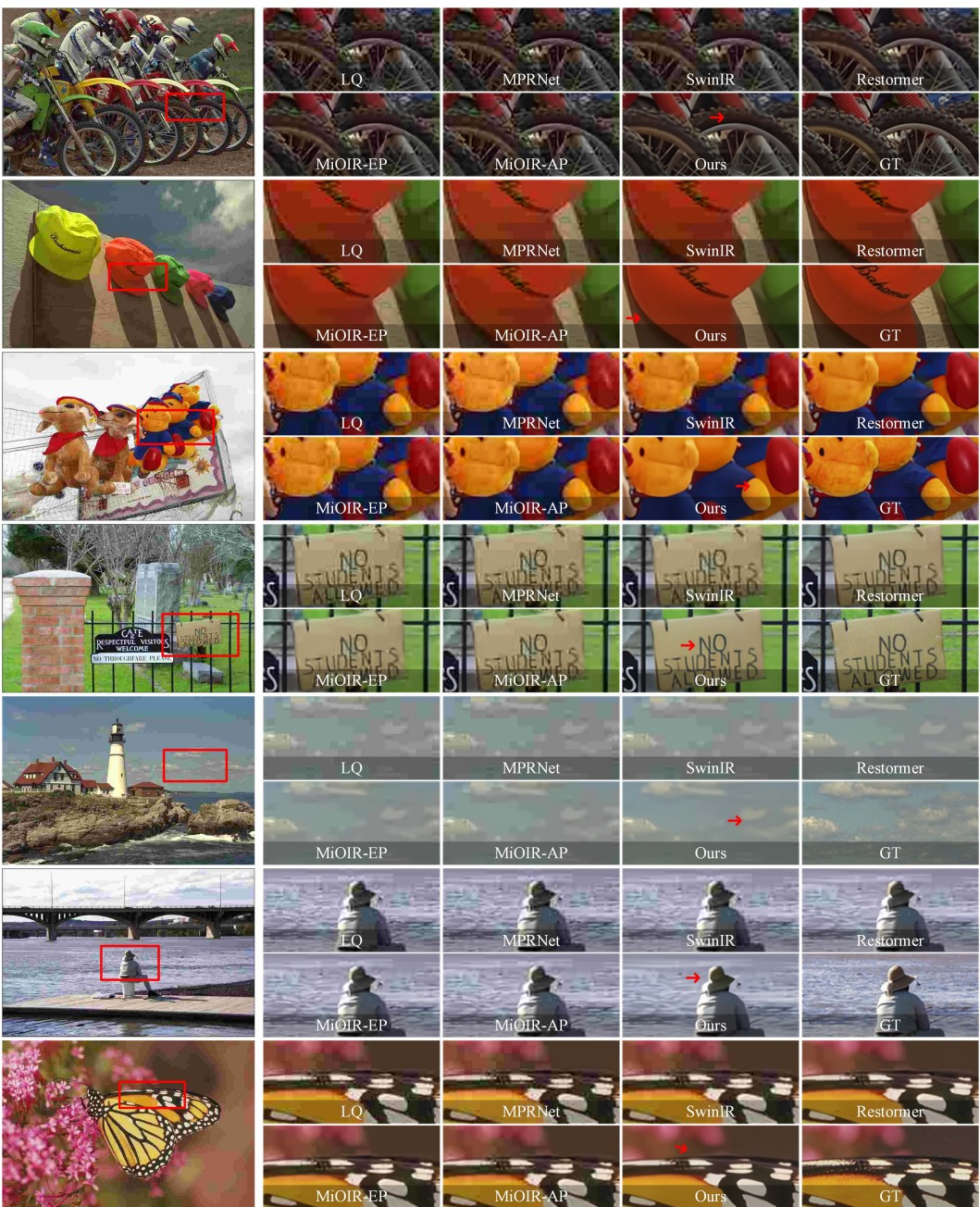

Figure J: Visual comparison of results on All-in-One image restoration (compression artifacts removal).

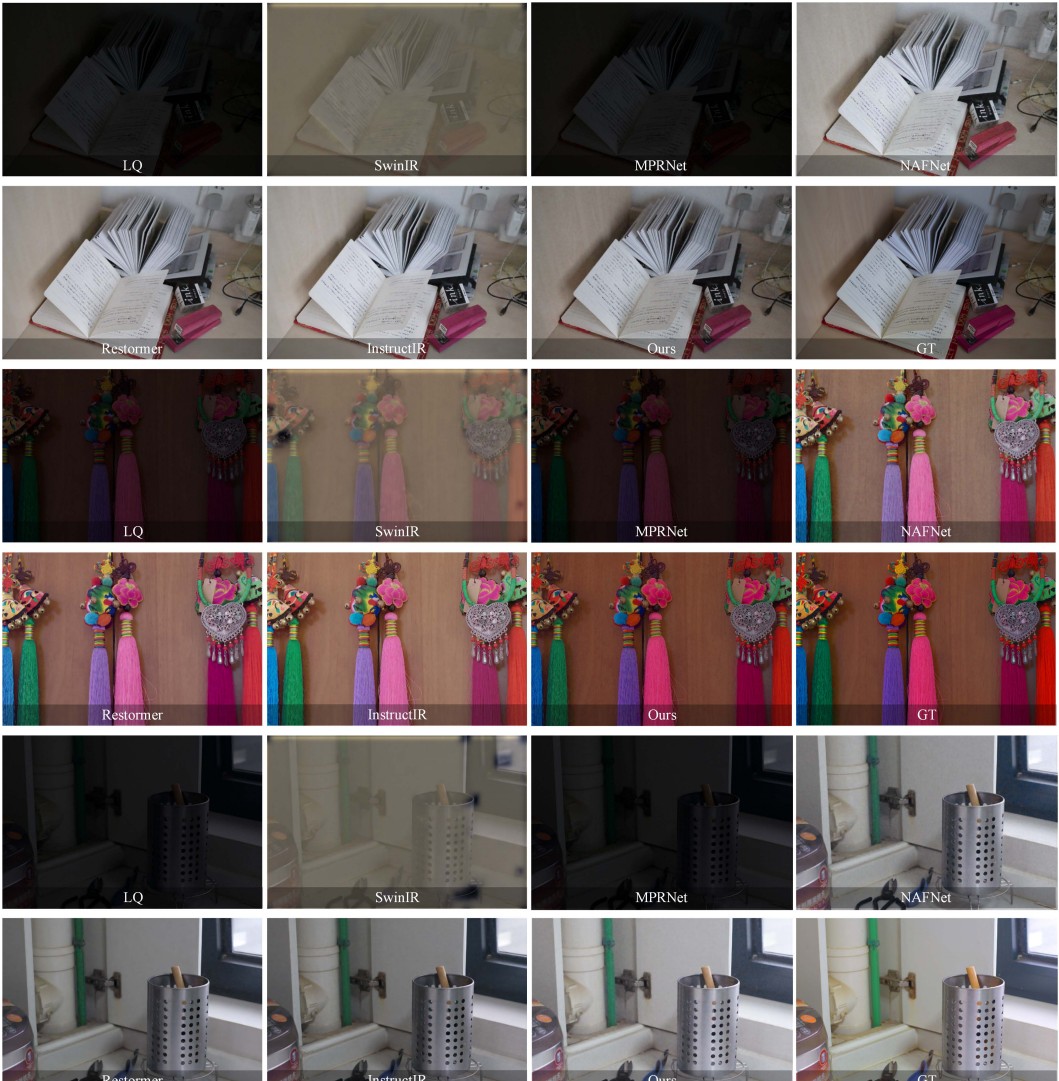

Figure K: Visual comparison of results on All-in-One image restoration (lowlight enhancement).

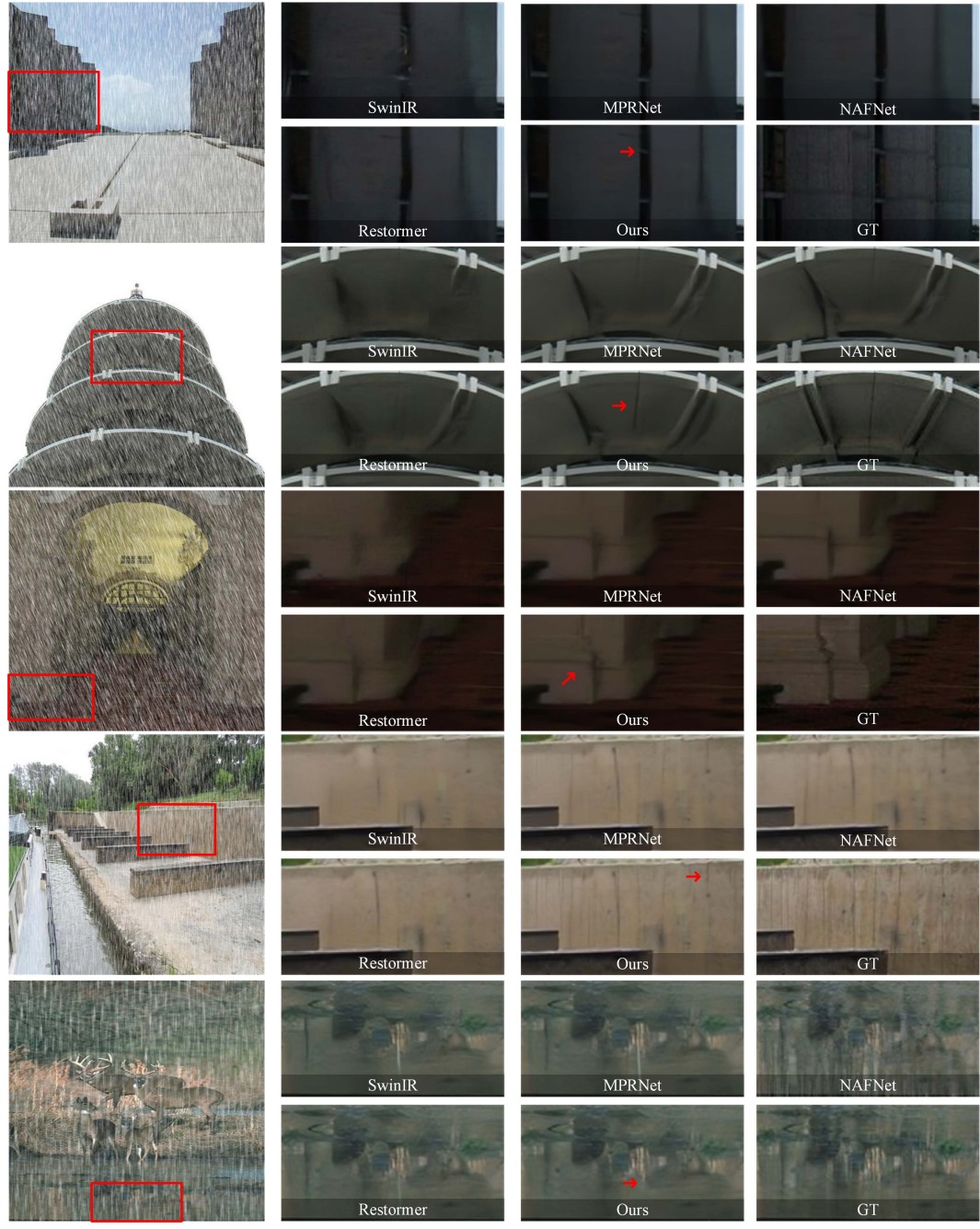

Figure L: Visual comparison of results with specific models (deraining).

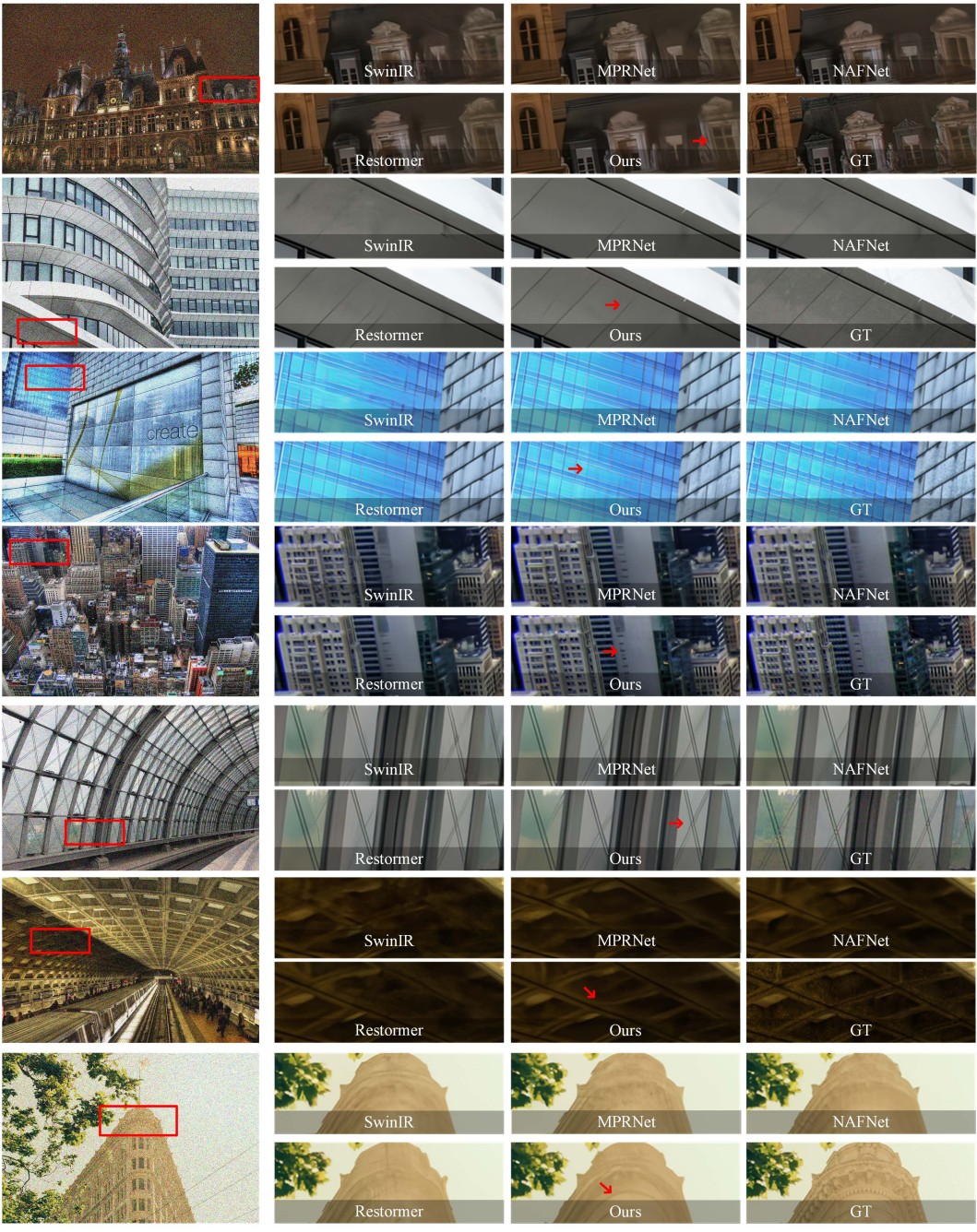

Figure M: Visual comparison of results with specific models (Gaussian-denoising).

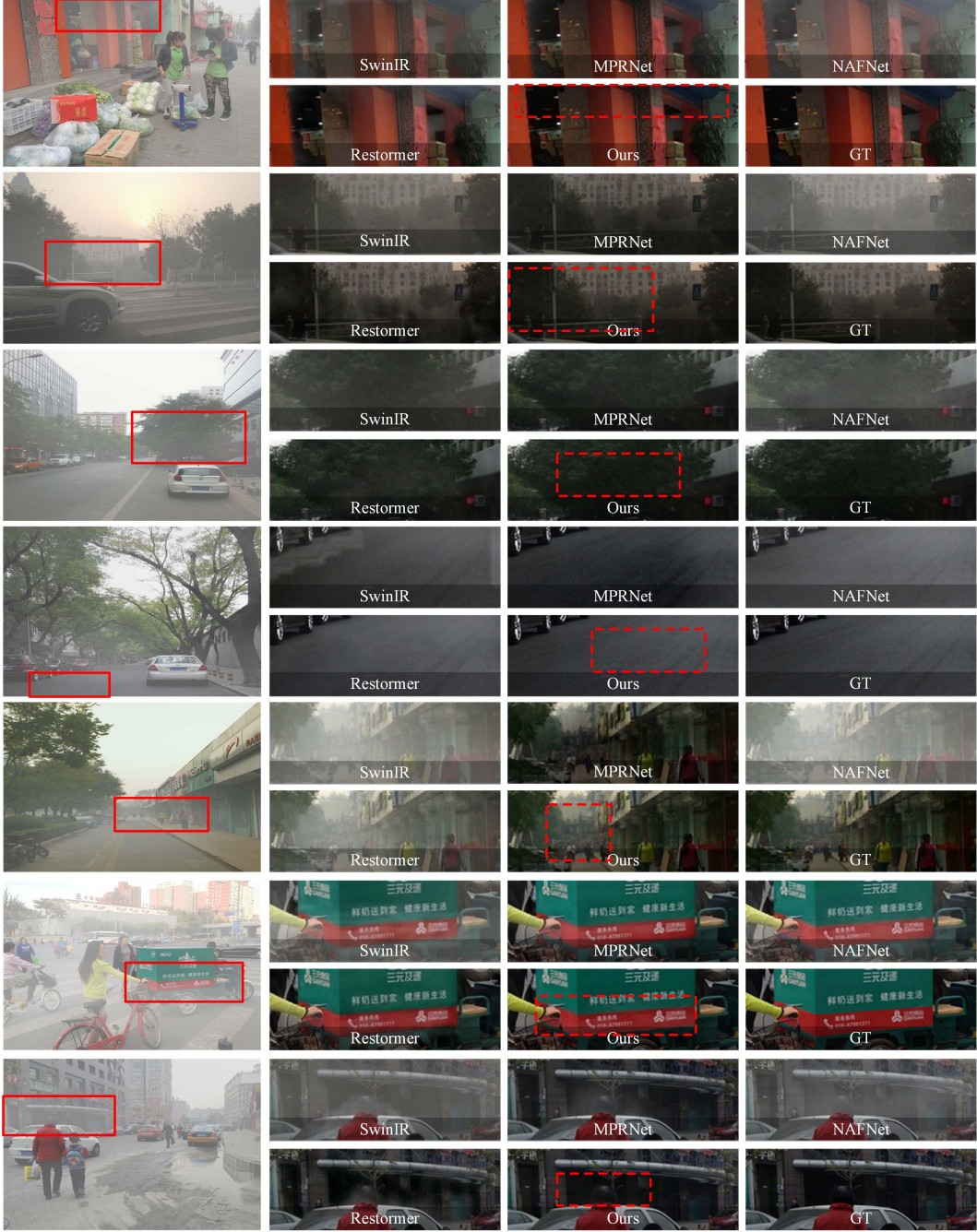

Figure N: Visual comparison of results with specific models (dehaze).

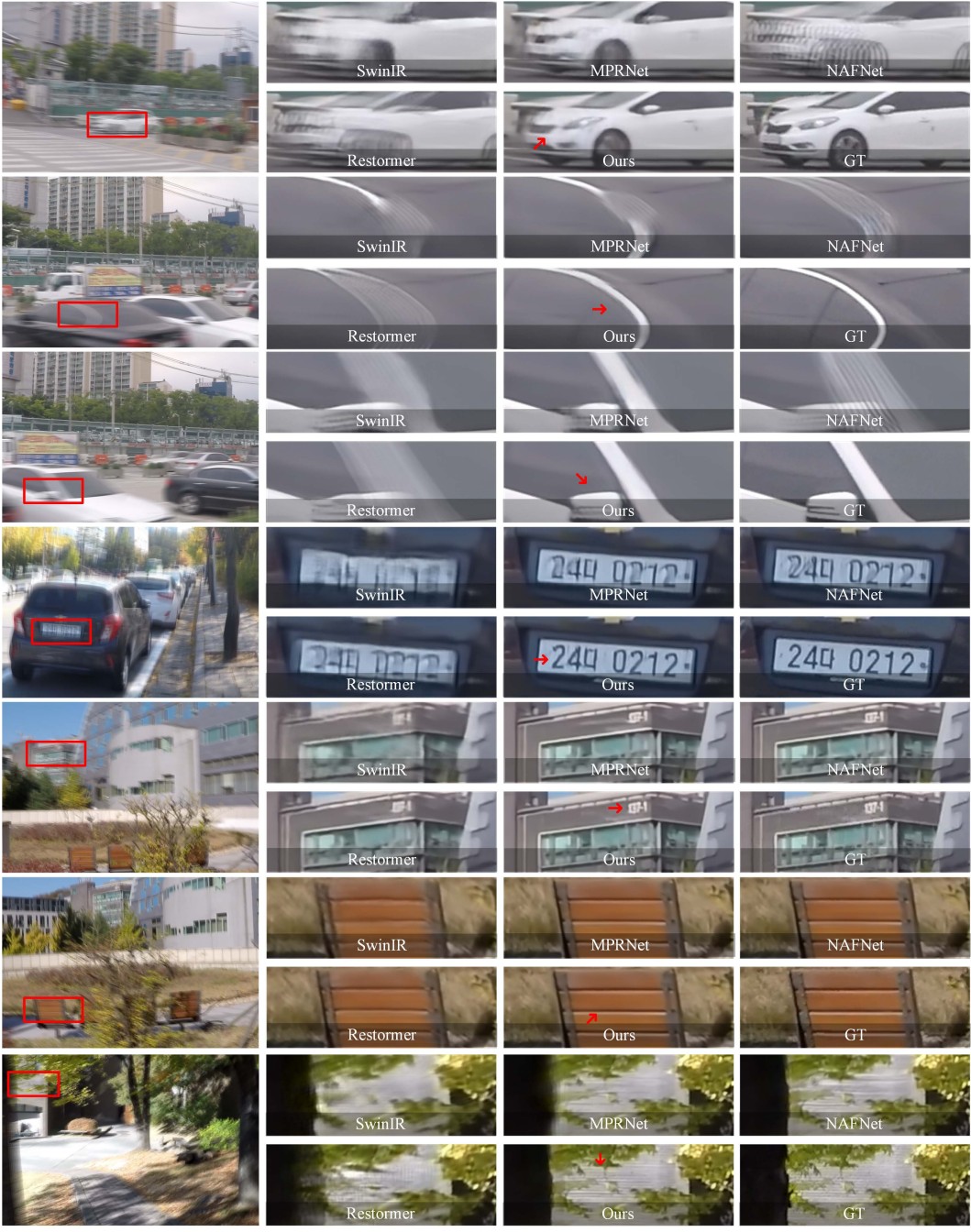

Figure O: Visual comparison of results with specific models (motion-deblurring).

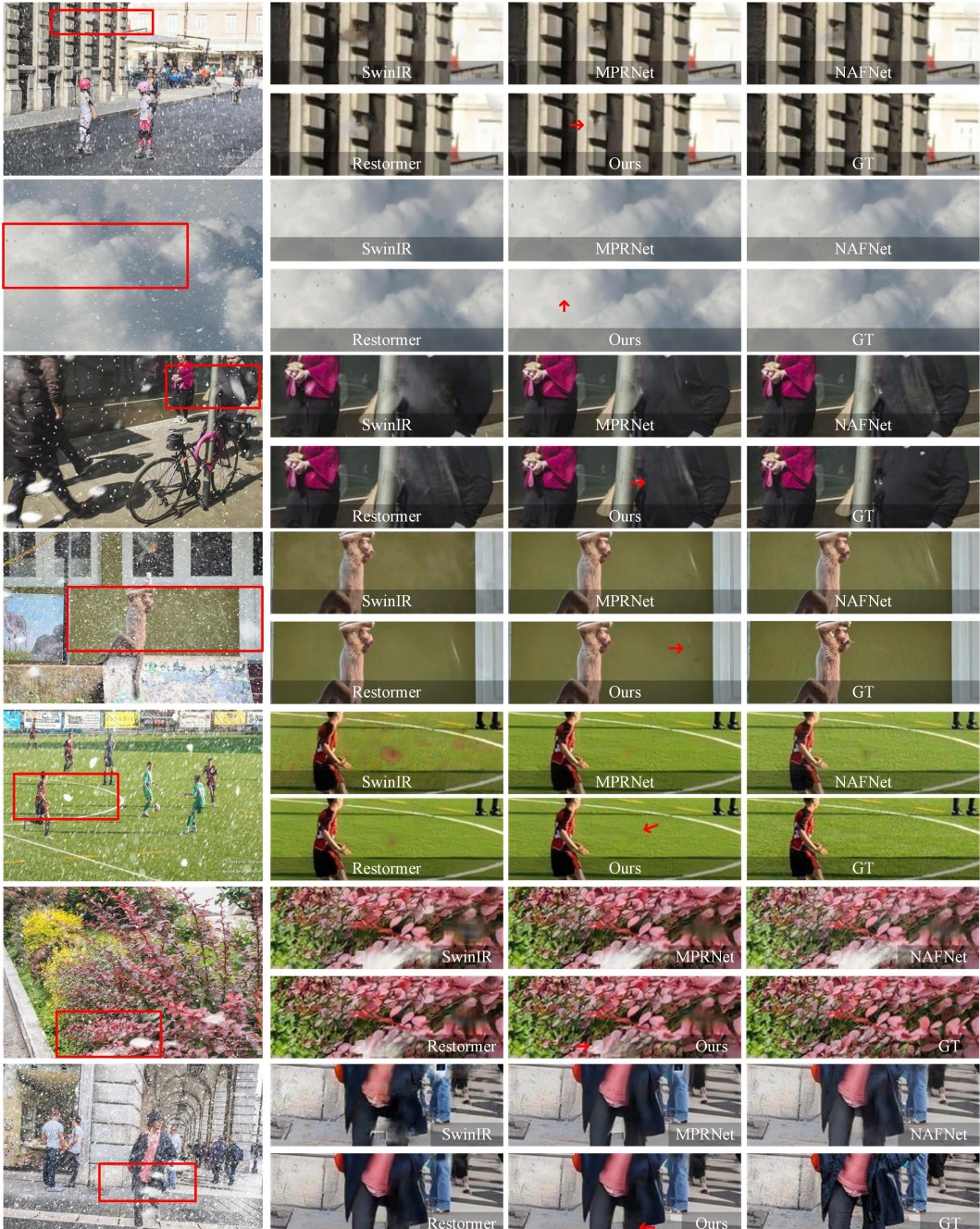

Figure P: Visual comparison of results with specific models (desnowing).

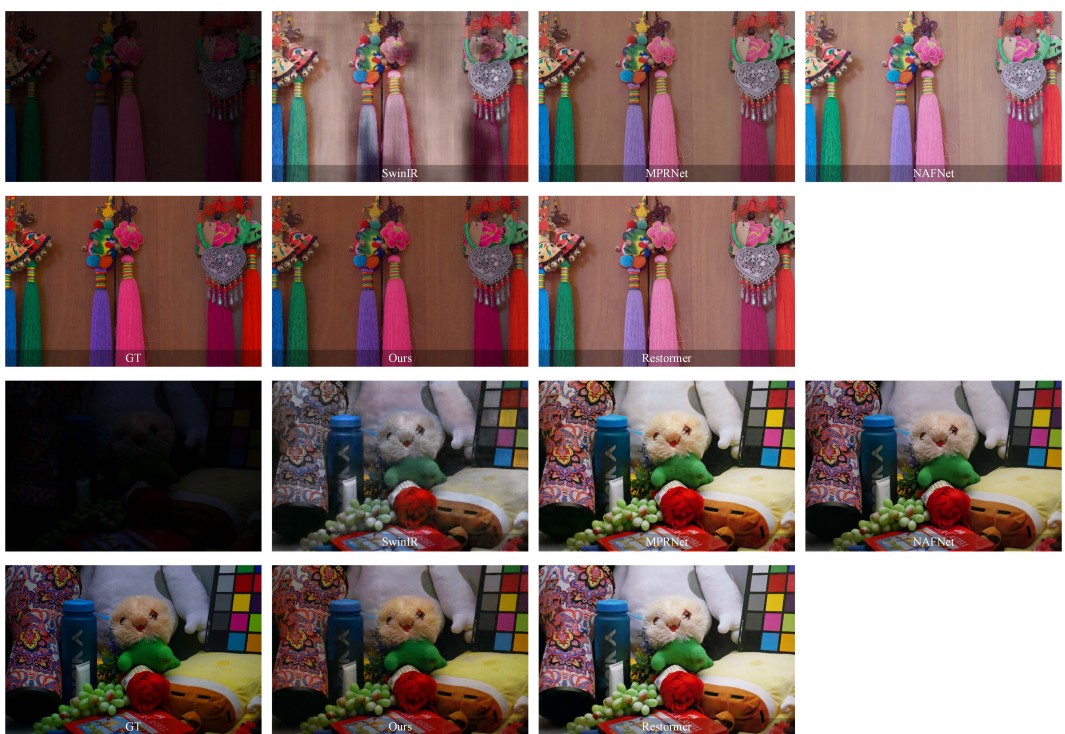

Figure Q: Visual comparison of results with specific models (lowlight enhancement).

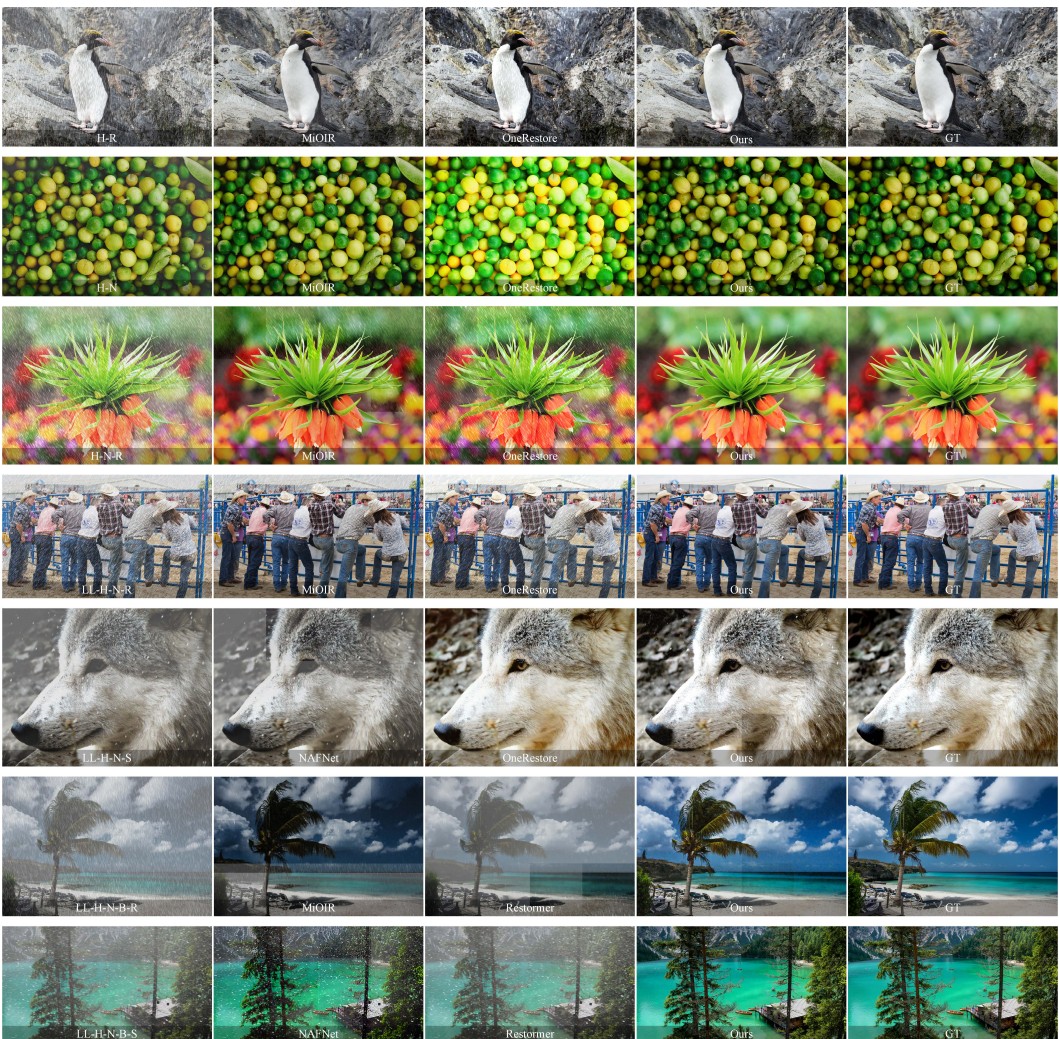

Figure R: Visual comparison of results on All-in-One image restoration (mixed-degradation, *in-dist*). Zoom in for more details.

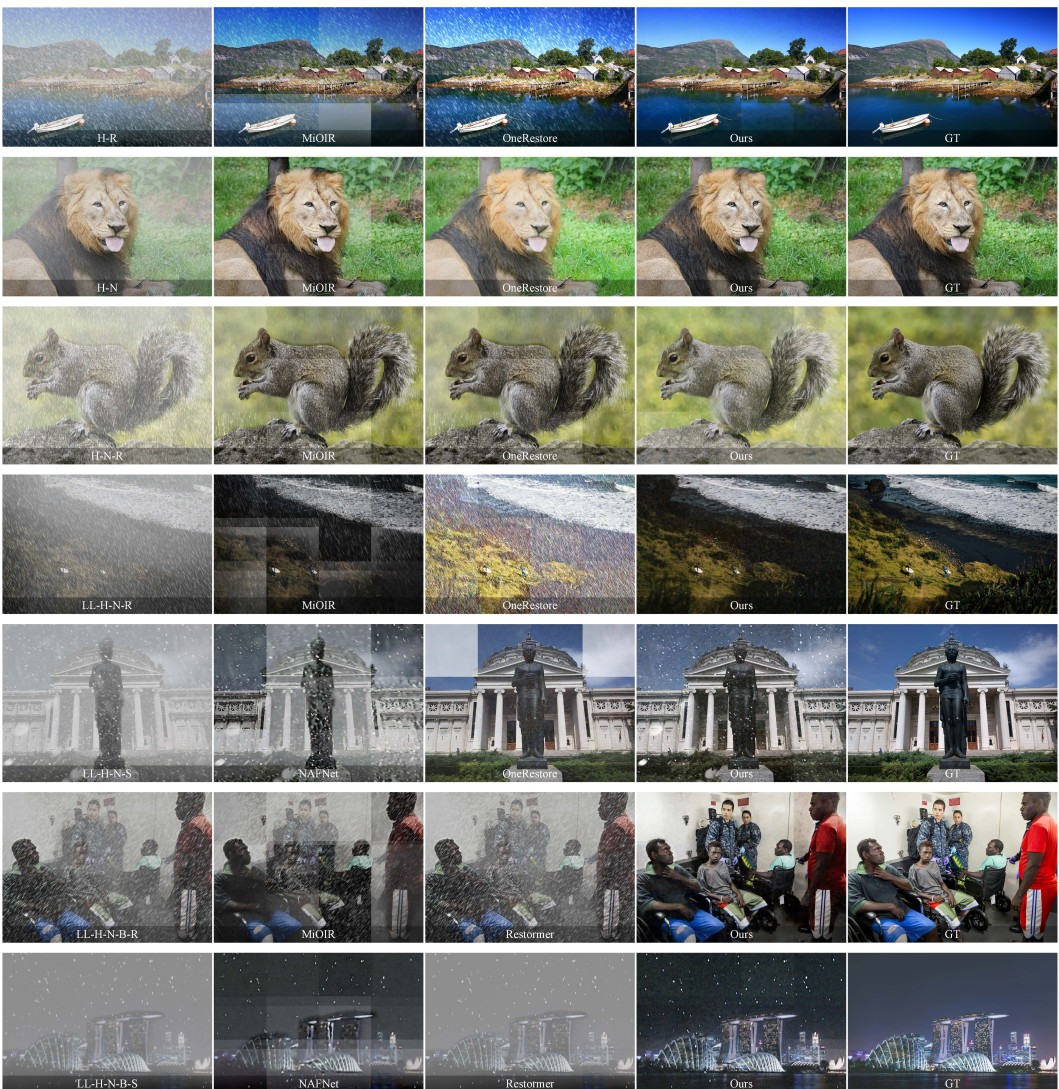

Figure S: Visual comparison of results on All-in-One image restoration (mixed-degradation, *out-dist*). Zoom in for more details.

