# OpenReview forum: "UniRestorer: Universal Image Restoration via Adaptively Estimating Image Degradation at Proper Granularity"
_ICLR.cc/2026/Conference — ICLR 2026 Poster_

### Official Review · Reviewer_Vp55 · 2025-10-23

**Soundness:** 3
**Presentation:** 3
**Contribution:** 2
**Rating:** 6
**Confidence:** 4

**Summary:**

UniRestorer presents a strong universal image restoration framework that achieves impressive performance across single, mixed, OOD, and even real-world degradations. Its combination of multi-granularity degradation representations, hierarchical expert specialization, and uncertainty-aware routing enables both robustness and precision, outperforming prior all-in-one baselines and approaching single-task SOTA results.
However, the core mechanism raises conceptual concerns: the training pipeline is overly complex, and the single-expert activation strategy appears suboptimal and parameter-inefficient. Clarification on these design choices—especially how the MoE routing behaves under mixed degradations—would be crucial to fully assess the method’s contribution.

**Strengths:**

1. The paper’s fine-grained degradation representation learning is a clear strength. By retraining a DA-CLIP–based extractor with fine-grained textual labels (e.g., light/medium/heavy noise or haze) and contrastive supervision, the authors enable the model to capture not only degradation types but also intensity levels.  Supplementary t-SNE visualizations results show that these representations are separable at both coarse and fine granularity, providing a solid foundation for the subsequent hierarchical clustering and multi-granularity expert design.
2. Demonstrates strong cross-distribution generalization, maintaining stable PSNR/SSIM across single, mixed, OOD, and real-world degradations, with zero-shot gains on unseen types.
3. Uncertainty-aware hierarchical routing adaptively selects coarse or fine experts for robustness or precision, reducing routing ambiguity and representation conflict while matching or surpassing single-task performance.
4. Comprehensive evidence beyond parameter scaling: broad baselines and ablations show gains arise from division of labor and routing, with the LoRA variant retaining near full-model performance.

**Weaknesses:**

1. The inference scheme activates only a single expert at a time, which limits the expressive power of the MoE and introduces substantial parameter redundancy, as many experts remain unused for each input.
2. When the MoE system encounters mixed degradations, how frequently does the router fall back to the 0-th level (coarse) expert?
If this fallback occurs in most cases, it is unclear why the model significantly outperforms a single Restormer trained directly on mixed degradations. Conversely, if the router instead activates a fine-level expert (e.g., a “rain” or “haze” expert) for a mixed input such as rain-haze, it would contradict the intended degradation clustering principle and could potentially degrade performance. Clarification on the router’s behavior and expert selection under mixed-degradation inputs is needed.
3. The overall training pipeline is overly complex and resource-intensive. It requires first training a degradation extractor, then performing hierarchical clustering, followed by separate training for multiple experts and an additional router stage.
4. The paper does not clarify how data sufficiency is ensured for the fine-granularity experts. Since the hierarchical clustering process recursively divides the training set into smaller subsets, some fine-level clusters may contain only a limited number of samples. It is unclear whether the authors applied any method to avoid underfitting or data imbalance across experts.

**Questions:**

The second weakness is the main issue that confuses me the most. I would appreciate a clear explanation, and I may consider raising my rating if it is addressed convincingly.

---

> ### Author Response · Authors · 2025-11-21
>
> > Q1: The inference scheme activates only a single expert at a time, which limits the expressive power of the MoE and introduces substantial parameter redundancy, as many experts remain unused for each input.
>
> Thanks for your comment. In this paper, we adopt the top-1 routing strategy for the following reasons: (1) Our model with top-1 routing strategy achieves inference costs comparable to other methods. (2) Using additional routers, such as the top-2 routing strategy, would double the inference cost. (3) This sparse activation strategy has proven effective in current LLM/MLLM. For example, DeepSeek-R1[1] has a total of 671B parameters, but only 37B parameters ($\sim$ 5.5\%) are activated in a single inference, significantly reducing computational costs. We believe this strategy is also reasonable and acceptable in image restoration tasks.
>
>
>
> > Q2: When the MoE system encounters mixed degradations, how frequently does the router fall back to the 0-th level (coarse) expert? If this fallback occurs in most cases, it is unclear why the model significantly outperforms a single Restormer trained directly on mixed degradations. Conversely, if the router instead activates a fine-level expert (e.g., a “rain” or “haze” expert) for a mixed input such as rain-haze, it would contradict the intended degradation clustering principle and could potentially degrade performance. Clarification on the router’s behavior and expert selection under mixed-degradation inputs is needed.
>
>
> Thanks for your comment. Firstly, our method is trained separately in single-degradation and mixed-degradation setup, respectively. In the mixed-degradation setup, the hierarchical clustering and expert training are both conducted on the CDD-11 dataset with mixed degradation types. We found that the clusters in the fine-grained level do not correspond to one concrete degradation type. Instead, the clusters in the fine-grained level correspond to one dominated mixed-degradation set. Specifically,  the dominated degradation set of clusters in the finest-grained level is:
>
> (1) \{Haze+Rain (H+R)\},
>
> (2) \{Rain (R)\},
>
> (3) \{Lowlight+Rain (L+R), Lowlight+Haze+Rain (L+H+R)\},
>
> (4) \{Lowlight+Haze (L+H)\},
>
> (5) \{Haze+Snow (H+S), Snow (S)\},
>
> (6) \{Haze (H)\},
>
> (7) \{Lowlight+Haze+Snow (L+H+S), Lowlight+Snow (L+S)\},
>
> (8) \{Lowlight (L)\}.
>
> The routing results in CDD-11 dataset are visualized in Fig. D.
> Taking the 'rain+haze' scenario mentioned by the reviewer as an example, the degradation-based routing assigns 58% of the samples to 'rain+haze' cluster, with the next most frequent assignments falling into the 'haze' or 'rain' clusters. Among these samples, 59% activate the expert at the finest level, while the remaining samples select coarser experts.
> Therefore, when processing an image with mixed degradation, the degradation-based routing first finds the most relevant degradation set at the finest-grained level, then granularity-based routing further selects the most appropriate expert for inference.  The results in Fig. D also demonstrate that the finest-level experts are mainly adopted when degradation is easy to estimate, and coarser-level experts are preferred when degradation is relatively difficult to estimate.
> We have added the analysis of routers' behavior in the revised version.
>
>
> [1] DeepSeek-R1: Incentivizing Reasoning Capability in LLMs via Reinforcement Learning

---

> > ### Author Response · Authors · 2025-11-21
> >
> > > Q3: The overall training pipeline is overly complex and resource-intensive. It requires first training a degradation extractor, then performing hierarchical clustering, followed by separate training for multiple experts and an additional router stage.
> >
> >
> > Thanks for your comments. Though we adopt a multi-stage training pipeline, the inference step is end-to-end, and its computational cost is comparable to a single network. The additional training cost mainly comes from the training of degradation extractor and the separate training of experts. We conducted experiments to demonstrate their necessity.
> >
> > (1) Degradation extractor: We conducted an ablation study on the degradation extractor in Table I of the appendix. The results demonstrate that current pre-trained extractors typically focus on learning task-relevant representations, which are not suitable for extracting finer-grained representations needed for our proposed hierarchical degradation space construction. Therefore, we retrained a new degradation extractor specifically designed for fine-grained degradations.
> >
> > (2) Separate training: We initially attempted to train experts and router networks in an end-to-end manner. However, in early stage, the poor restoration ability of experts makes training unstable. To address this, we adopt a separate training strategy, where experts are learned in advance and are not affected by the routers, allowing the stability of the whole model.
> >
> > Furthermore, our method is flexible, the training cost can be adjusted by changing restoration backbone. For example, we replace the used Restormer to lightweight NAFNet (named LiteNAF.) and lightweight Restormer (named LiteRes.) by reducing their channels, the performance and training cost comparison are reported in 3-task all-in-one image restoration task. Results in Table R4-1 show that we can change the restoration backbone to fit the requirement of training cost in practice.
> >
> > Table R4-1: Comparison to PromptIR with our lightweight variants.
> >
> > | Methods            | Rain  | Haze  | Noise | Time (h) |
> > |--------------------|-------|-------|-------|----------|
> > | PromptIR  | 36.37 | 30.58 | 31.12 | 201      |
> > | Ours               | 41.03 | 36.44 | 31.43 | 578      |
> > | Ours-LiteRes.      | 39.43 | 35.38 | 31.35 | 442      |
> > | Ours-LiteNAF.      | 37.76 | 31.69 | 31.28 | 169      |
> >
> >
> >
> > > Q4: It is unclear whether the authors applied any method to avoid underfitting or data imbalance across experts.
> >
> >
> >
> > Thanks for your comments. We also found the mentioned problems in the clustering step: data imbalancing, underfitting of data when depth is large. To alleviate the first problem, we adopt balanced K-Means as [2], which takes data balance into account in clustering step. In the second problem, we restrain the number of samples to not lower than 200 in clustering step, and finally, the depth of 3 is adopted. We statistic the number of samples in each cluster on the finest-grained level in Table R4-2.
> >
> > Table R4-2: The number of samples in each cluster on the finest-grained level.
> >
> > | Cluster ID | 0    | 1    | 2    | 3    | 4    | 5    | 6    | 7    |
> > |------------|------|------|------|------|------|------|------|------|
> > | # Samples  | 9.3% | 9.8% | 15.5% | 11.2% | 15.6% | 11.9% | 16.8% | 9.4% |
> >
> >
> > [2] MoDE, CLIP Data Experts via Clustering

---

### Official Review · Reviewer_tpRc · 2025-10-31

**Soundness:** 3
**Presentation:** 3
**Contribution:** 2
**Rating:** 4
**Confidence:** 4

**Summary:**

The paper proposes UniRestorer, a universal all-in-one image restoration framework that bridges degradation-agnostic and degradation-aware paradigms. It performs hierarchical clustering on degradation representations to build a multi-granularity mixture-of-experts (MoE) model. Through degradation and granularity estimation, the router adaptively selects fine- or coarse-grained experts, achieving robustness against estimation errors. Experiments on single- and mixed-degradation benchmarks show that UniRestorer surpasses prior all-in-one methods and nearly matches single-task performance

**Strengths:**

1. This paper introduces a multi-granularity degradation representation that unifies coarse- and fine-grained experts. The idea of granularity estimation to quantify degradation uncertainty and guide routing is novel and intuitive.
2. Comprehensive experiments: covers 7 single-degradation and 11 mixed-degradation settings, plus real-world and unseen tasks.
3. Paper is well-organized and technically detailed with intuitive figures (especially Fig. 3 illustrating routing).
4. The hierarchical MoE design could inspire cross-domain generalization and adaptive restoration architectures.

**Weaknesses:**

1. While granularity estimation is conceptually convincing, there’s no formal uncertainty-theoretic or probabilistic analysis of its behavior.
2. Each granularity level adds parameters and routing complexity; actual FLOPs and latency comparisons are limited. The LoRA variant helps, but trade-offs between full and LoRA experts could be better quantified.
3. The K-means–based clustering assumes a consistent degradation embedding space; sensitivity to clustering hyperparameters (number of clusters, feature normalization) is not reported.
4. Evaluation primarily uses synthetic degradations; more real-world degradation diversity (e.g., motion blur, ISP artifacts) would strengthen the claim of universality.
5. The effect of routing noise or errors in granularity estimation itself is underexplored; visualization of routing confidence would add interpretability.

**Questions:**

1. How sensitive is performance to the number of levels (l = 3) or cluster count per level? Could adaptive clustering improve robustness?
2. Can you visualize which experts are selected for different degradations and how granularity affects routing under ambiguous inputs?
3. Could the same hierarchical expert tree generalize to unseen degradation combinations(e.g., low-light + blur + noise) without retraining?
4. In hybrid usage, can user-provided degradation cues override granularity routing? Would it further close the gap with single-task models?

---

> ### Author Response · Authors · 2025-11-21
>
> > Q1: While granularity estimation is conceptually convincing, there’s no formal uncertainty-theoretic or probabilistic analysis of its behavior.
>
> Thanks for your comments.
> Due to the markdown compile issue, we provide the uncertainty-theoretic analysis in Section A of the Appendix.
>
>
> > Q2: Each granularity level adds parameters and routing complexity; actual FLOPs and latency comparisons are limited. The LoRA variant helps, but trade-offs between full and LoRA experts could be better quantified.
>
> Thanks for your comments. We have included comparisons on the parameter budget in Tables F and H of the Appendix. Although our method introduces an additional parameter budget, it maintains comparable computational costs to the single Restormer model during the inference phase.
>
> To ensure a fair comparison, we also scale up the parameters of other baselines, including Restormer, PromptIR, and MoCE-IR, to match the scale of ours and train them from scratch using the same degradation protocol. As shown in Table H of the Appendix, the restoration of our model still outperforms other baselines. This indicates that the improvement of our model is not simply due to an increase in parameters.
>
> Additionally, to evaluate the gap between the full-parameter model and the LoRA variants, we provide an analysis comparing both their performance and parameter counts. In our experiments, we generally adopt a rank of 8 for different tasks. As demonstrated in Table H, the full-parameter model achieves better restoration performance, with approximately 1 dB PSNR gain, while requiring 3.6 times more parameters than the LoRA variants. To some extent, the full-parameter model can be regarded as an upper bound for the LoRA variants. In practice, the rank size of LoRA can be adjusted based on the device's storage limitations.
>
>
> > Q3: The K-means–based clustering assumes a consistent degradation embedding space; sensitivity to clustering hyperparameters (number of clusters, feature normalization) is not reported.
>
> Thanks for your comments. In Table 6 of our manuscript, we provide a sensitivity analysis on the number of clusters (#DRs). The results show that increasing the number of DR groups from 1 to 8 leads to significant improvement in restoration performance. This is because a finer partition allows the degradation patterns within each group to be more consistent. However, when the number of DR groups is further increased from 8 to 16, the improvement becomes minimal. This suggests that overly fine-grained DR groups may introduce redundant partitions, increasing the difficulty of degradation estimation and impairing restoration performance. Additionally, we provide a quantitative analysis on the clustering quality of different numbers of clusters in Table R1-1 (Q1 from Reviewer xgD4), which demonstrates that the used hyperparameter is relatively optimal.
>
> We apply L2 normalization before K-means clustering. It is noted that feature normalization is not a tunable hyperparameter, but rather a standard practice, as it prevents the magnitude of features from distorting the clustering process. Therefore, we adopt feature normalization as a default operation before K-Means clustering.
>
> > Q4: Evaluation primarily uses synthetic degradations; more real-world degradation diversity (e.g., motion blur, ISP artifacts) would strengthen the claim of universality.
>
> Thank you for your comment. Tables 3 and 4 of our manuscript show that, on both real-world and unseen degradation datasets, our model outperforms other state-of-the-art all-in-one methods. Additionally, we conduct experiments on scenarios involving real motion blur and ISP artifacts removal. As demonstrated in Table R3-1, our method exhibits better generalization capability compared to other all-in-one methods. We have included comparisons with more real-world degradation datasets in our revised manuscript.
>
> Table R3-1: Comparisons to SOTA methods on more real-world degradations.
>
> | Methods          | ISP Artifact | RealBlur-J | RealBlur-R |
> |------------------|--------------|------------|------------|
> | PromptIR    | 12.81       | 25.48      | 33.34      |
> | MiOIR         | 13.79       | 24.51      | 16.79      |
> | MoCE-IR-3T     | 12.97       | 25.60      | 31.39      |
> | MoCE-IR-5T   | 12.88       | 14.30      | 12.20      |
> | **Ours**              | **16.26**   | **26.10**  | **34.06**  |

---

> ### Author Response · Authors · 2025-11-21
>
> > Q5: The effect of routing noise or errors in granularity estimation itself is underexplored; visualization of routing confidence would add interpretability.
>
> Thank you for your comments. We first define the `GT` granularity as the granularity level at which the model achieves the best restoration performance, compared to all other granularity levels. To assess the effect of routing noise and errors, we compare the restoration performance of our model using the `GT` granularity and the estimated granularity (`actual`). Table R3-2 illustrates the number of loads for different granularity levels and the corresponding restoration performance on *in-dist.* data. Table R3-3 shows the same analysis on *out-dist.* data. The restoration performance gap between the estimated granularity and `GT` granularity is minimal. This suggests that granularity estimation has only minor errors, making a slight impact on restoration performance.
> In addition, we visualize the routing confidence for each sample in *in-dist.* and *out-dist.* scenarios in Fig. C.
> We have added the analysis and visualization in the revision.
>
> Table R3-2: Effect of errors in granularity estimation, *in-dist*.
>
> | Methods       | 0-th Level | 1-st Level | 2nd Level | PSNR  |
> |---------------|------------|------------|-----------|-------|
> | Ours (`actual`) | 5          | 17         | 78        | 24.46 |
> | Ours (`GT`)     | 7          | 13         | 80        | 24.53 |
>
> Table R3-3: Effect of errors in granularity estimation, *out-dist*.
>
> | Methods       | 0-th Level | 1-st Level | 2nd Level | PSNR  |
> |---------------|------------|------------|-----------|-------|
> | Ours (`actual`) | 34         | 50         | 16        | 19.45 |
> | Ours (`GT`)     | 29         | 46         | 25        | 19.66 |
>
> > Q6: Could adaptive clustering improve robustness?
>
> Thanks for your comment. We conduct experiments on adaptive clustering. Similar to GRIDS[1], we offer a candidate list of [2, 3, 4, 5, 6] for cluster count $k$ and use the binary search strategy to search for the best $k$ in splitting each cluster. During splitting, we adopt the silhouette score as the metric to decide whether to split and to evaluate if the selected $k$ is appropriate for the current split. Additionally, we ensure that the minimum number of samples in each cluster is no lower than 200. This process results in a three-level tree with hyperparameter $[1, 6, 13]$.
> To evaluate the effectiveness of the results of adaptive clustering, we train our method in the newly formed degradation space and evaluate the performance on CDD-11 dataset. The results in Table R3-4 show that adaptive clustering indeed further improves the performance of our method.  We have added this in the revision.
>
>
> Table R3-4:  Performance comparison of adaptive clustering on CDD-11 dataset.
>
> | Methods  | L     | H     | R     | S     | L-H   | L-R   | L-S   | H-R   | H-S   | L-H-R | L-H-S | Avg.  |
> |----------|-------|-------|-------|-------|-------|-------|-------|-------|-------|-------|-------|-------|
> | Adaptive | 27.76 | 34.71 | 35.76 | 37.35 | 27.32 | 27.05 | 26.90 | 31.86 | 31.02 | 26.14 | 26.38 | 30.20 |
> | Ours     | 27.71 | 34.53 | 36.07 | 36.84 | 26.88 | 26.93 | 26.73 | 32.02 | 30.82 | 25.78 | 25.87 | 30.02 |
>
>
>
>
>
> > Q7: Can you visualize which experts are selected for different degradations and how granularity affects routing under ambiguous inputs?
>
> Thanks for your comments.
> We provide the clustering results of the finest level in the CDD-11 dataset. The dominant degradations of each cluster are as follows :
>
> (1) \{Haze+Rain (H+R)\},
>
> (2) \{Rain (R)\},
>
> (3) \{Lowlight+Rain (L+R), Lowlight+Haze+Rain (L+H+R)\},
>
> (4) \{Lowlight+Haze (L+H)\},
>
> (5) \{Haze+Snow (H+S), Snow (S)\},
>
> (6) \{Haze (H)\},
>
> (7) \{Lowlight+Haze+Snow (L+H+S), Lowlight+Snow (L+S)\},
>
> (8) \{Lowlight (L)\}.
>
> Fig. D in our revised manuscript shows the routing results in CDD-11 dataset. The results indicate that when degradation is ambiguous, the accurate degradation estimation becomes difficult, our method prefers coarser-grained granularity levels to guarantee the expert network can handle the input. In contrast, when degradation is easy to estimate, our method typically selects the finest-grained experts for superior performance.
> We have added the visualization of routing confidence in our revised version.
>
>
>
> [1] GRIDS: Grouped Multiple-Degradation Restoration with Image Degradation Similarity.

---

> ### Author Response · Authors · 2025-11-21
>
> > Q8: Could the same hierarchical expert tree generalize to unseen degradation combinations(e.g., low-light + blur + noise) without retraining?
>
> Thanks for your comment. As shown in Table R3-5, we evaluate the performance of our model and other all-in-one methods under unseen degradation combinations. The results demonstrate that our model exhibits better generalization ability.
>
> Table R3-5 Restoration results on combined degradation.
> | Degradation        | OneRestore | MoCE-IR-CDD11 | Ours     |
> |--------------------|------------|---------------|----------|
> | Lowlight+Blur+Noise | 12.35/0.489 | 15.91/0.614   | 17.64/0.631 |
>
>
>
>
> > Q9: In hybrid usage, can user-provided degradation cues override granularity routing? Would it further close the gap with single-task models?
>
> Thanks for your comment.
> When users provide a perceived degradation type and degree of an image, they can also provide a confidence level (e.g., 0, 1, 2, higher values indicate higher confidence) for the current degradation, which can serve as a granularity routing cue. We randomly synthesize rain streaks of varying intensities on clean images from the Urban100 dataset. We then evaluate the performance when both degradation type, degree, and confidence are provided (referred to as `Deg. & Gran.`). Specifically, we compare the `Deg. & Gran.` with two other conditions: `Deg.` (which only provides degradation type, as described in our main paper) and `single-task` (Restormer trained on deraining task). Results in Table R3-6 show that it has potential in further improving the performance and closing the gap with single-task models when granularity cues are provided.
>
>
> Table R3-6 Comparisons of different hybrid usage.
> | Method        | `Deg.`    | `Deg. & Gran.` | `Single-task` |
> |---------------|-----------|----------------|---------------|
> | Deraining     | 32.18/0.973 | 32.42/0.975    | 32.62/0.980   |

---

> ### Comment · Reviewer_tpRc · 2025-11-28
> **Response of Reviewer tpRc**
>
> After reading the authors' rebuttal and other reviewers' comments, I decided to raise my score to 6.
> The authors have provided thorough additional experiments and analyses, which address most of my concerns. Overall, I am satisfied with the quality of the paper, and I recommend that the relevant supplementary content be incorporated into the final version.

---

> > ### Author Response · Authors · 2025-11-28
> >
> > Thank you for your comments and valuable suggestions. We will incorporate the analysis and additional experiments presented in the rebuttal into the final version of our paper. Specifically, we will include the uncertainty-theoretic analysis, provide more comprehensive parameter quantification, analyze the sensitivity to clustering hyperparameters and add experiments on adaptive clustering. We will also add more comparisons on additional real-world datasets, add visualizations of routing confidence and qualitative results, and further explore the use of user-provided degradation cues.

---

### Official Review · Reviewer_TgnW · 2025-10-31

**Soundness:** 3
**Presentation:** 3
**Contribution:** 3
**Rating:** 8
**Confidence:** 3

**Summary:**

UniRestorer proposes a multi-granularity MoE framework for all-in-one image restoration. It (1) hierarchically clusters degradation space, (2) trains multi-level MoE experts, and (3) uses joint degradation + granularity estimation to route inputs to the most suitable expert. Claims large gains over SOTA all-in-one models and narrows gap to single-task models. Code and models will be released.

**Strengths:**

Granularity estimation elegantly handles degradation estimation noise — robust and practical.
Hierarchical clustering + MoE scales well across 15+ degradation types.
Large, consistent gains (e.g., +2.1 dB PSNR on mixed test sets).
Comprehensive ablations (granularity levels, routing loss, expert count).
Clean figures: t-SNE of degradation space, expert activation heatmaps.

**Weaknesses:**

Clustering is offline and static — no online adaptation to new/unseen degradations.
Granularity estimator adds overhead — no inference latency reported (vs. PromptIR, AirNet).
No theoretical justification for hierarchical clustering choice (e.g., why 3 levels?).
Evaluation limited to synthetic degradations — no real-world camera pipeline (e.g., RAW → ISP).
MoE training unstable? No mention of load balancing loss or expert collapse.

**Questions:**

Report inference FPS on RTX 3090 for 512×512 input — how much slower than PromptIR?
Test on real-world degradations (e.g., DND, SIDD, GoPro real blur).
Ablate dynamic clustering — can granularity be learned end-to-end without offline K-means?

---

> ### Author Response · Authors · 2025-11-21
>
> > Q1: No online adaptation to new/unseen degradations. No inference latency reported (vs. PromptIR, AirNet). No theoretical justification for hierarchical clustering choice (e.g., why 3 levels?). No real-world camera pipeline (e.g., RAW → ISP). MoE training unstable? No mention of load balancing loss or expert collapse.
>
>
>  Thank you for your suggestion.
>
> - Online adaptation: Ideally, if enough degradation types are included, the clustering step should be conducted offline, the new/unseen degradations could be an inter point in the constructed degradation space, and can be well processed by the corresponding expert.
> However, the comparison of existing all-in-one methods is limited to benchmark datasets.
> If the constructed degradation space cannot cover the new/unseen degradations, the model will find the closest point and use the corresponding expert, which may lead to limited performance.
> To evaluate the generalization ability for new degradations, we have conducted experiments on the real-world (Table 3 of the main paper) and unseen (Table 4 of the main paper) datasets. The results show that our model demonstrates better generalization compared to other baselines.
> In addition, following your suggestions, here we try updating the clustering results in a momentum way and finetuning the corresponding expert model. We conduct the experiments on unseen defocus blur. The results in Table R2-1 show that it achieves the best generalization performance, as adaptation ability is improved by updating the clustering.
>
> Table R2-1 Restoration performance on DPDD.
>
> | **Methods**                | **Defocus Deblur** |
> |----------------------------|-----------------------|
> | **PromptIR**   | 21.76                 |
> | **MioIR**   | 20.41                 |
> | **MoCE-IR-3T**    | 21.23                 |
> | **MoCE-IR-5T**   | 15.66                 |
> | **Ours**                     | 24.00                 |
> | **Ours (finetune)**          | 25.11                 |
>
>
>
>
> - Overhead: Table F of the Appendix in our manuscript reports the FLOPs and inference latency for all methods. The FLOPs and latency are calculated using a 512 × 512 × 3 input on a single NVIDIA A6000 GPU. We include the new baseline AirNet in Table F of our revised manuscript.
> As shown in Table F, though we have an additional degradation extractor and routers, the introduced computational cost is minimal. Thus, the FLOPs and latency of our method are comparable to our backbone network, Restormer, demonstrating the efficiency of our model.
>
>
>
> - Theoretical justification: Naturally, degradation exhibits hierarchical characteristics. For example, two images (e.g., one with light blur while the other with heavy blur) may be grouped together only based on the presence of blur, but can be further categorized into different groups when considering blur strength. The reasons for adopting three hierarchical levels are as follows: (1) as the ablation study shown in Table 6, the three-level setting provides a better balance between performance and training cost. (2) The quantitative analysis on clustering effectiveness in Table R1-1 (*see Q1 for Reviewer xgD4*) suggests 3-level hierarchy is more suitable.
>
>
>
> - Real-world camera: In Table 3 and Table 4 of our manuscript, we evaluate the restoration performance on real-world degradations. The results show that our model demonstrates better generalization on unseen degradations compared to other methods. Additionally, we conduct experiments on removing ISP artifacts (on dataset [1]) to further show the generalization of our model.
> As shown in Table R2-2, our model outperforms state-of-the-art all-in-one methods by 3-4 dB, highlighting the effectiveness of our approach.
>
> Table R2-2 Restoration performance on ISP Artifacts.
> | Methods                     | ISP Artifacts |
> |-----------------------------|---------------|
> | PromptIR    | 12.81         |
> | MiOIR         | 13.79         |
> | MoCE-IR-3T     | 12.97         |
> | MoCE-IR-5T     | 12.88         |
> | **Ours**                     | 16.26         |
>
>
>
>
> - Stable training: Vanilla MoE model often faces the challenge of expert collapse, where most samples are assigned to just one or a small fraction of the experts.
> In this paper, we alleviate this issue by firstly training each expert independently and then freezing the experts in training routers. The rationale behind this is: it allows each expert to focus on specific degradations, thus simplifying the learning task and minimizing the competition between experts.
> However, we found a collapse in granularity routing. To alleviate this problem, we adopt a load balancing loss in granularity routing, as shown in Eq. (10). The ablation study in Table J of the Appendix demonstrates that the load balancing loss improves both *in-dist.* and *out-dist.* performance.
>
> [1] Learning RAW-to-sRGB Mappings with Inaccurately Aligned Supervision.

---

> ### Author Response · Authors · 2025-11-21
>
> > Q2: Report inference FPS on RTX 3090 for 512×512 input — how much slower than PromptIR? Test on real-world degradations (e.g., DND, SIDD, GoPro real blur). Ablate dynamic clustering — can granularity be learned end-to-end without offline K-means?
>
>
> Thank you for your suggestion.
>
> - Inference FPS:  We report inference FPS on RTX 3090 for 512$\times$512 input as shown in Table R2-3.
> Our method, with an FPS of 2.431, is approximately 26.7\% slower than the PromptIR, which achieves an FPS of 3.317. However, our method is flexible and compatible with various expert networks to meet the efficiency targets. To demonstrate this, we conduct experiments by replacing our backbone, Restormer, with two lightweight backbones: lightweight NAFNet (LiteNAF.) and lightweight Restormer (LiteRes.). As shown in Table R2-3, Ours-LiteNAF. achieves FPS of 7.547, which is about twice as fast as PromptIR, and its restoration performance outperforms PromptIR in the 3-task. This demonstrates that the variant of our method not only improves efficiency but also maintains superior restoration performance, highlighting its potential for real-time applications.
>
> Table R2-3 Inference FPS.
> | Methods           | Rain  | Haze  | Noise | # FLOPs | Latency (s) | FPS   |
> |-------------------|-------|-------|-------|---------|-------------|-------|
> | PromptIR  | 36.37 | 30.58 | 31.12 | 1266.2  | 0.3014      | 3.317 |
> | Ours              | 41.03 | 36.44 | 31.43 | 1155.8  | 0.4112      | 2.431 |
> | Ours-LiteRes.     | 39.43 | 35.38 | 31.35 | 672.7   | 0.3355      | 2.981 |
> | Ours-LiteNAF.     | 37.76 | 31.69 | 31.28 | 154.7   | 0.1325      | 7.547 |
>
>
>
>
> - Real-world degradations: As illustrated in Table R2-4, we report results on real-world degradations, including SIDD, DND, RealBlur-J, and RealBlur-R. It is worth noting that since motion blur in GoPro is synthesized, we evaluate the restoration performance of motion blur on two real-world datasets: RealBlur-J and RealBlur-R. The results show that our method outperforms other all-in-one baselines, highlighting the effectiveness of our method. We have included the above analyses in our revised manuscript.
>
> Table R2-4 Results on real-world degradations.
> | Methods                | SIDD  | DND   | RealBlur-J | RealBlur-R |
> |------------------------|-------|-------|------------|------------|
> | PromptIR | 24.32 | 25.19 | 25.48      | 33.34      |
> | MiOIR      | 22.87 | 23.67 | 24.51      | 16.79      |
> | MoCE-IR-3T  | 23.91 | 24.04 | 25.60      | 31.39      |
> | MoCE-IR-5T  | 15.90 | 21.16 | 14.30      | 12.20      |
> | **Ours**                | **25.10** | **26.47** | **26.10**  | **34.06**  |
>
>
> - Dynamic Clustering: We initially attempted to train the clustering, granularity, routers, and experts in an end-to-end manner. However, in the early step, we found that the poor performance of expert networks makes training unstable, causing the learning of granularity and the routing process to collapse. Therefore, we adopt a multi-stage training strategy, where clustering, expert, and router networks are constructed independently, which is also similar to some MoE-based all-in-one image restoration works [1, 2].
> This disentanglement allows each part to specialize in its respective functions, ensuring the stability of the overall framework.
>
> [1] Lora-ir: Taming low-rank experts for efficient all-in-one image restoration.
>
> [2] All-in-one image restoration for unknown degradations using adaptive discriminative filters for specific degradations.

---

### Official Review · Reviewer_xgD4 · 2025-11-01

**Soundness:** 3
**Presentation:** 2
**Contribution:** 2
**Rating:** 6
**Confidence:** 5

**Summary:**

Existing all-in-one restoration schemes are either degradation-agnostic or degradation-aware, leaving a clear performance gap to single-task experts. This paper proposes UniRestorer, which first hierarchically clusters the degradation space and trains a multi-granularity mixture-of-experts (MoE) network. At inference it jointly estimates degradation type and granularity to activate the most suitable expert. Extensive experiments show that UniRestorer significantly surpasses state-of-the-art all-in-one competitors and narrows the gap to dedicated single-task models.

**Strengths:**

1. The paper proposes UniRestorer, the first framework that simultaneously exploits degradation and granularity estimation to overcome the inherent limitations of both degradation-agnostic and degradation-aware restoration methods.
2. Extensive quantitative and qualitative experiments convincingly demonstrate the superiority of UniRestorer over existing all-in-one baselines and its competitive performance against task-specific models.
3. The idea of granularity-aware expert selection is clearly articulated and technically grounded; it offers a fresh insight that could inspire future work on robust universal image restoration.

**Weaknesses:**

1. The paper lacks a quantitative analysis of the proposed hierarchical degradation-clustering step.
2. No comparison or discussion is provided against alternative clustering strategies (e.g., the spectral clustering adopted in SEAL).
3. It is unclear whether the Restormer baseline in Table 1 was re-trained under exactly the same degradation protocol and parameter budget; an ablation that removes both degradation and granularity estimation while keeping the backbone capacity fixed would better isolate the gain of the proposed method.

**Questions:**

Please check the weakness

---

> ### Author Response · Authors · 2025-11-21
>
> > Q1： Quantitative analysis on hierarchical degradation-clustering step.
>
>
> Thanks for your comments. We follow SEAL using silhouette score [1] to quantitatively evaluate the quality of the clustering, which is defined as follows:
>
> $
> S = \frac{1}{N} \sum_{i=1}^N s_i, \quad  s_i=\frac{b_i-a_i}{\max(a_i,b_i)}\in [-1,1],
> $
>
>
> where $a_i$ denotes the average intra-cluster distance of $i$-th sample, and $b_i$ denotes the minimum average distance between $i$-th sample and all points in the nearest neighboring cluster. As indicated by the above equation, a higher silhouette score reflects a more compact and better-separated clustering structure.
> We compute the silhouette score of our hierarchical degradation-clustering results on the CDD-11 dataset.
>
> Table R1-1 Silhouette score on K-means
> | **$\{ \# DRs \}$** | **1st-level** | **2nd-level** | **3rd-level** |
> |-------------------|---------------|---------------|---------------|
> | $\{1, 2\}$        | 0.3103        | --            | --            |
> | $\{1, 4\}$        | 0.4926        | --            | --            |
> | $\{1, 8\}$        | 0.6605        | --            | --            |
> | $\{1, 16\}$       | 0.5932        | --            | --            |
> | $\{1, 4, 8\}$     | 0.4926        | 0.6686        | --            |
> | $\{1, 4, 8, 16\}$ | 0.4926        | 0.6686        | 0.5525        |
>
>
> As shown in Table R1-1, when the number of DR groups increases from 2 to 8, the silhouette score improves. This is because a finer partition allows degradation patterns in each group to be more consistent.
> However, further increasing the number of DR groups from 8 to 16 does not yield additional improvement. This suggests that overly fine-grained DR groups may introduce redundant or noisy partitions, leading to less meaningful separation and consequently limiting the clustering quality.
> Besides, the setting of $\{1, 4, 8\}$ produces a higher silhouette score than that of $\{1, 8\}$. This indicates that introducing an intermediate degradation level helps the model better capture the continuous transition among degradation patterns, leading to more compact intra-group structures and clearer inter-group separations. In addition to silhouette score, the ablation study in Table 6 of the main paper also indicates that $\{1, 4, 8\}$ is a better trade-off between performance and training cost.
> Thus, in this paper, we adopt the hyperparameter setting of $\{1, 4, 8\}$.
> We have incorporated the above analysis into our revised manuscript.
>
> > Q2: Discussion on alternative clustering strategies.
>
>
> Thank you for your suggestion.
> Spectral clustering aims to partition data into clusters by leveraging the eigenvalues and eigenvectors of the similarity matrix. Spectral clustering captures complex, non-convex cluster shapes but is computationally expensive, while K-means clustering is simpler and more efficient, making it suitable for large datasets with spherical clusters.
> We also conduct additional experiments with a variant of our proposed model, denoted as $Ours_{spc}$, where the K-Means clustering is replaced with spectral clustering.
> We report both clustering quality (i.e., silhouette score) and restoration performance (i.e., PSNR) of the $Ours_{spc}$, as shown in Table R1-2 and Table R1-3, respectively.
>
> Table R1-2 Silhouette score on spectral clustering
> | **$\{ \# DRs \}$** | **1st-level** | **2nd-level** | **3rd-level** |
> |-------------------|---------------|---------------|---------------|
> | $\{1, 2\}$        | 0.3824        | --            | --            |
> | $\{1, 4\}$        | 0.4630        | --            | --            |
> | $\{1, 8\}$        | 0.5895        | --            | --            |
> | $\{1, 16\}$       | 0.2779        | --            | --            |
> | $\{1, 4, 8\}$     | 0.4630        | 0.6668        | --            |
> | $\{1, 4, 8, 16\}$ | 0.4630        | 0.6668        | 0.4889        |
>
>
> Table R1-3 Restoration performance
> | **Methods**        | **L**  | **H**  | **R**  | **S**  | **L-H** | **L-R** | **L-S** | **H-R** | **H-S** | **L-H-R** | **L-H-S** | **Avg.** |
> |--------------------|--------|--------|--------|--------|--------|--------|--------|--------|--------|----------|----------|----------|
> | **$Ours_{spc}$** | 26.91  | 27.41  | 32.77  | 32.84  | 25.62  | 25.86  | 24.79  | 26.25  | 24.19  | 23.87    | 23.77    | 26.75    |
> | **$Ours_{kms}$** | 27.10  | 27.36  | 33.33  | 32.55  | 25.53  | 25.74  | 25.16  | 26.42  | 24.21  | 24.03    | 24.17    | 26.87    |
>
>
> Note that due the limitation of time, Table R1-3 only includes results for two hierarchical layers, i.e., \#DRs=$\{1,4\}$. The results demonstrate that $Ours_{spc}$ achieves similar performance to $Ours_{kms}$ in both the silhouette score and PSNR. This suggests our model is robust to alternative clustering methods. We have added the discussion of alternative clustering methods in our revision.
>
> [1] Silhouettes: a graphical aid to the interpretation and validation of cluster analysis.

---

> > ### Author Response · Authors · 2025-11-21
> >
> > > Q3: It is unclear whether the Restormer baseline in Table 1 was re-trained under exactly the same degradation protocol and parameter budget;
> >
> >
> > Thanks for your comments.
> > The baselines (i.e., MPRNet, SwinIR, NAFNet, and Restormer) in Table 1 of the main paper are all re-trained under exactly the same degradation protocol as ours, but with parameter budgets the same as their original works.
> > In Table H of the Appendix, we also conduct an ablation study on Restormer and other all-in-one baselines by scaling them up. The results show that our method outperforms these baseline methods, demonstrating that the performance improvements of our model stem from the proposed routing based on degradation and granularity estimation techniques, rather than only from an increased parameter budget. We have clarified these points in the revised version of our manuscript.

---

### Author Response · Authors · 2025-11-21

We sincerely thank all reviewers for all the precious comments and valuable advice. In view of these suggestions, we conducted several additional experiments and revised our article accordingly. The major changes are summarized as follows.

- We added a **quantitative analysis** of the proposed hierarchical degradation **clustering** in Appendix C.

- We provided additional **comparisons and discussions** of our model using alternative clustering strategies, including **spectral clustering and adaptive clustering** in Appendix E.

- We conducted **additional experiments** of our model on real-world degradations, such as **real-world denosing, real-world deblurring, and ISP artifacts removal** in Appendix D.

- We included a comprehensive computational cost analysis, including **inference latency, FPS, parameters, training time, and inference time** in Appendix D.

- We **visualized the granularity routing** and evaluated the **effect of routing errors** in granularity estimation of our method in Appendix F.

- We provided **a theoretical justification** for our choice of hierarchical clustering in Appendix C and conducted an **uncertainty-theoretic analysis** for granularity estimation in Appendix A.

- We added an analysis of our model with **user-provided degradation cues** in Appendix E.

---

### Author Response · Authors · 2025-12-03

Dear Area Chairs,

We sincerely thank the reviewers and area chairs for their valuable time and constructive feedback. Below is a brief summary of this paper.

This paper proposes UniRestorer, where we perform hierarchical degradation clustering and train a multi-granularity MoE image restoration model. By estimating degradation and routing on proper granularity, our method activates the most suitable expert in inference. Extensive experiments demonstrate UniRestorer outperforms previous all-in-one methods across various benchmark datasets and closes the performance gap to single-task models.

In the discussion phase, only Reviewer tpRc (rating: 4) provided a follow-up response. The summary is as follows:

Reviewer xgD4 (rating: 6) raised questions on quantitative analysis of hierarchical clustering, alternative clustering strategies, and baseline protocols. We supplemented quantitative evaluations with silhouette scores (in Table B of the Appendix), added spectral clustering experiments showing similar performance (in Tables M and N of the Appendix), and clarified that all baselines were retrained under the same degradation settings. The comparisons of the same parameter budget were included (in Table H of the Appendix).

Reviewer TgnW (rating: 8) focused on adaptation ability, efficiency, hierarchical design, performance on real-world datasets, and training stability. We conducted experiments on online adaptation to unseen degradation (in Q1 of the Reviewer TgnW), showing that the performance further improves. We additionally reported FPS and supplemented the overhead comparison on RTX 3090 (in Q2 of the Reviewer TgnW). We justified the three-level hierarchy with both quantitative analysis (in Table B of the Appendix) and ablation studies (in Table 6). In additional evaluations on real-world datasets and ISP artifacts (in Table E of the Appendix), our method shows clear advantages. We explained how the used load-balancing loss ensures stable optimization. Finally, we clarified the advantages and necessity of our static clustering compared with dynamic clustering.

Reviewer tpRc (rating: 4) confirmed that all concerns were addressed and stated that the rating would be raised to 6. The main concerns of the reviewer are theoretical justification, model complexity, sensitivity to clustering hyperparameters, generalization ability on real-world/unseen degradations, routing behaviors, and hybrid usage. In response, we added an uncertainty-theoretic analysis (in Section A of the Appendix), provided full comparisons of parameter budgets, FLOPs, and latency (in Table F of the Appendix), and demonstrated that performance gains are not simply due to the increased parameters (in Table H of the Appendix). We conducted experiments on clustering hyperparameters (in Table 6) and the adaptive clustering method (in Table O of the Appendix) to analyze the sensitivity and the robustness. We added more real-world/unseen degradation evaluations (in Table E of the Appendix), showing clear improvements over existing methods. We analyzed routing behaviors by visualizing the routing confidence and routing results (in Section F of the Appendix). We further explored the hybrid usage when more user-provided cues are incorporated (in Table P of the Appendix).

The main concern of Reviewer Vp55 (rating: 6) is routing behavior in mixed degradation, and the reviewer commented that the rating would be raised if the concern is addressed. To this end, we first visualized the routing results (in Fig. D of the Appendix) showing our method can activate fine- or coarse-level experts appropriately. Taking the example mentioned by the reviewer, we further analyzed the routing behavior based on the routing results (detailed information can be seen in Q2 of Reviewer Vp55). The other concerns of Reviewer Vp55 are expert activation, training complexity, and data imbalance. We clarified that the use of top-1 routing aims to maintain comparable inference cost with existing methods, and the used sparse-activation MoE design is widely adopted in existing LLM/VLM. We provided training time statistics (in Q3 of the Reviewer Vp55), explained the advantages and necessity of our training pipeline. We finally declared that we adopted balanced K-Means and controlled cluster sizes to alleviate the data imbalance issue.

We sincerely appreciate the efforts of the reviewers and area chairs. We believe the revisions address all concerns and further highlight the significance of our contributions.

Best regards,

Authors of Paper 8500

---

### Meta-Review · Area_Chair_zXCT · 2026-01-07

**Summary:**

This paper proposes UniRestorer, an all-in-one image restoration framework. The key idea is to hierarchically cluster a learned degradation embedding space and train a complex multi-granularity MoE architecture with routing guided by degradation-type and granularity estimation.

Reviewers generally thought the idea well-motivated and the empirical performance strong, with consistent gains over all-in-one baselines and competitive results against single-task models.

The main issues raised by the reviewers include
- whether the hierarchical clustering and routing choices are quantitatively justified and fairly compared (e.g., clustering analysis, alternative clustering, baseline retraining and parameter budget control),
- whether the method is practical, related to latency/overhead, training complexity, MoE stability/load balancing,
- whether claims of universality/generalization are supported beyond synthetic settings.

The rebuttal added experiments for the raised questions and resolved most of the technical questions asked by reviewers.

The AC remains concerned about the gap between synthetic benchmark data and real-world applications—given the often unrealistic nature of commonly used datasets in this area. The AC is also not fully convinced by the justification for the benefits of the complex design and the underlying reasons for the reported gains (i.e., principled architecture/learning improvements vs. effectively ensembling more parameters via multiple LoRA modules).

As a key design component, the multi-granularity MoE routing mechanism and the role of experts are not yet comprehensively studied. For instance: How accurate is the router? Do different experts genuinely learn distinct functionalities, or do they behave more like an ensemble via implicit mixture or task-specific selection? How diverse are the learned experts? If the experts correspond closely to specific degradations in the training data, defined by pre-set synthesis hyperparameters (types and degrees), can this architecture reliably translate to realistic applications where degradations are more complex, continuous, and harder to categorize?

After the rebuttal, reviewers generally considered their technical issues addressed for the scope of the submission and acknowledged the paper’s technical contributions to the relevant community, while some high-level concerns on the justification of the design and generalization to real data remain (AC also partially share similar concerns with overlapping and different perspectives).  Overall, the paper appears accept-worthy, even though broader open problems and deeper mechanistic understanding remain.

**Reviewer Concerns:**

The rebuttal addresses most reviewer concerns related to implementation details, empirical validation, and practicality, including
- quantitative clustering analysis and alternative strategies (choice of the clustering method) (xgD4),
- baseline retraining (whether compared mdoels are retrained) and parameter-budget clarification (xgD4),
- inference latency and efficiency  (TgnW),
- additional benchmarks (TgnW, tpRc),
- routing confidence or error analysis and visualization (tpRc),
- adaptive clustering and sensitivity studies (tpRc),
- explanations related to MoE, e.g., stability, load balancing, and data imbalance across experts (TgnW, Vp55).

The rebuttal resolved many of these technical and empirical questions.

However, several higher-level concerns remain partially outstanding, primarily raised by Vp55 and tpRc: the mechanistic justification of the multi-granularity MoE design, and the generalization gap between synthetic benchmarks and realistic real-world degradations. It also remains unclear whether experts that align well with synthetically defined degradation types and degrees can robustly transfer to complex, uncategorized real-world conditions, or whether gains stem mainly from increased parameterization (e.g., multiple LoRA experts). The AC is also concerned on related aspects from overlapped and different perspectives, as discussed above.

These issues suggest the need for clearer statement of the scope and deeper analysis in the main paper, even though reviewers generally acknowledge the paper’s technical contributions.

**Reviewer Scores:**

- xgD4. Likely stays 6. Their technical concerns (clustering quantification, alternative clustering, baseline fairness) are substantially addressed, though some comparisons remain partial.


- TgnW. Likely stays 8. Their concerns were directly answered with latency/FPS, stability/load balancing, real-world tests, and limited adaptation experiments.


- tpRc (initial 4). Raised to 6 in the discussion. This aligns with the rebuttal addressing most concerns and recommending that supplementary/rebuttal content be integrated into the final version.


- Vp55. Likely stays 6 or possibly 7. Depending on how convincing they find the mixed-degradation routing clarification and the data-balance measures.

---

### Decision · Program_Chairs · 2026-01-26

Accept (Poster)